# 🧔 SSR: Socratic Self-Refine for Large Language Model Reasoning

## Abstract

Large Language Models (LLMs) have demonstrated remarkable reasoning abilities, yet existing test-time frameworks often rely on coarse self-verification and self-correction, limiting their effectiveness on complex tasks. In this paper, we propose **S**ocratic **S**elf-**R**efine (**SSR**), a novel framework for fine-grained evaluation and precise refinement of LLM reasoning. Our proposed SSR decomposes model responses into verifiable (sub-question, sub-answer) pairs, enabling step-level confidence estimation through controlled re-solving and self-consistency checks. By pinpointing unreliable steps and iteratively refining them, SSR produces more accurate and interpretable reasoning chains. Empirical results across five reasoning benchmarks and three LLMs show that SSR consistently outperforms state-of-the-art iterative self-refinement baselines. Beyond performance gains, SSR provides a principled black-box approach for evaluating and understanding the internal reasoning processes of LLMs.

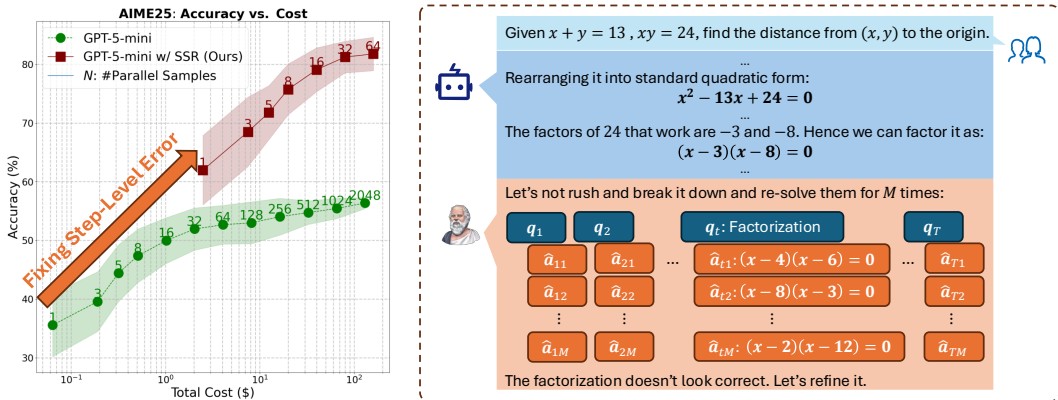

Figure 1: **Test-Time Parallel Scaling Performance (Left)** and **Conceptual Overview (Right)** of our proposed **S**ocratic **S**elf-**R**efine (**SSR**). By decomposing responses into Socratic steps, re-evaluating intermediate results through self-consistency, and refining specific step-level errors, SSR achieves substantially higher initial accuracy (∼**67.57% relative improvement**) and continues to scale effectively even when standard Chain-of-Thought (CoT) begins to saturate. Notably, this performance advantage holds under comparable computational cost. Experiments are conducted with `GPT-5-mini` in low-reasoning, low-verbosity mode.

## 1 Introduction

Large Language Models (LLMs) have rapidly advanced the frontier of machine reasoning, demonstrating impressive performance across domains ranging from mathematical problem solving to complex logical inference (Wei et al., 2022a; Wang et al., 2022; Chung et al., 2024; Guo et al., 2025; Ke et al., 2025). Central to these capabilities is the paradigm of reasoning with explicit intermediate steps, often instantiated through chain-of-thought (CoT) prompting (Wei et al., 2022b). By externalizing reasoning traces, CoT enables models to articulate their latent decision-making process, offering both interpretability and opportunities for iterative improvement (Madaan et al., 2023). Despite these strengths, the reasoning traces generated by LLMs remain prone to cascading errors: a single flawed

step can propagate downstream, leading to incorrect or incoherent final answers (Wu et al., 2025; You et al., 2025). This vulnerability raises pressing questions about how to reliably evaluate, refine, and searching for better multi-step reasoning at test time.

Existing frameworks have sought to address these challenges largely fall into two paradigms: sample selection with self-verification and self-refinement. Sample selection with self-verification, aims to assess response reliability by assigning confidence scores to completed reasoning traces either by LLM-as-a-Judge (Gu et al., 2024), or a specific ranking model (Snell et al., 2024), and then through multiple sampling and selection improves the final answer reliability (Zheng et al., 2023b; Chen et al., 2025). While these approaches can identify low-quality outputs, they often operate at a coarse granularity, overlooking subtle step-level errors embedded within long derivations (Fang et al., 2025). Self-refinement methods, by contrast, encourage LLMs to iteratively critique and revise their own responses (Madaan et al., 2023; Zhang et al., 2024; Bi et al., 2024). Although such frameworks have yielded measurable gains, their reliance on holistic self-feedback frequently limits their ability to pinpoint and correct specific erroneous steps. As a result, both paradigms struggle to provide robust and interpretable error correction in complex reasoning tasks.

In this paper, we propose **S**ocratic **S**elf-**R**efine (`SSR`), a novel framework designed to overcome these limitations by introducing fine-grained, step-level evaluation and targeted refinement of LLM reasoning. `SSR` reformulates the reasoning process into a sequence of verifiable (sub-question, sub-answer) pairs, which we refer to as Socratic steps. This decomposition enables precise confidence estimation through controlled re-solving and self-consistency checks at the step level. Unreliable steps are selectively refined, allowing the model to fix errors without depending on vague feedback. By iteratively applying this process, `SSR` improves both the accuracy and interpretability of LLM reasoning, offering a principled black-box approach to evaluating and refining model behavior.

Empirical results across 5 reasoning tasks (3 mathematical and 2 logical) and multiple state-of-the-art LLMs demonstrate that `SSR` consistently outperforms baseline self-refinement methods. Beyond raw accuracy gains, our analysis shows that `SSR` yields more reliable refinement trajectories, particularly when combined with plan-level adjustments or adaptive gating mechanisms. These findings highlight the importance of explicit step-level verification in building trustworthy LLM reasoning systems. More broadly, `SSR` represents a step toward interpretable and controllable test-time reasoning, bridging the gap between coarse-grained judgment and fine-grained error correction. To summarize, our contributions are:

- We propose a novel framework, Socratic Self-Refine (`SSR`), that allows more fine-grained confidence estimation and precise error control over decomposed reasoning steps. By formulating reasoning as a sequence of (sub-question, sub-answer) pairs, `SSR` overcomes the limitations of existing holistic self-refinement methods.
- We empirically validate `SSR` on 5 reasoning tasks using two state-of-the-art models, demonstrating that it consistently outperforms existing self-refine-based baselines.
- Our `SSR` introduces a mechanism for eliciting the model's step-level confidence, by having the LLM re-solve each sub-question multiple times with explicit context control. Leveraging self-consistency as a reliable confidence estimate for each step, `SSR` provides a pioneering effort in evaluating and interpreting the internal reasoning processes of LLMs.

## 2 RELATED WORK

**Self-Evaluation and Refinement of LLMs.** Recent work has introduced both *intrinsic* and *generative* approaches for LLM self-evaluation. On the intrinsic side, uncertainty-based methods estimate correctness either through consistency, by comparing multiple independently generated outputs (Kuhn et al., 2023; Manakul et al., 2023), or through statistics derived from the model's output distribution (Kang et al., 2025; Fu et al., 2025; Zhang et al., 2025a). On the generative side, the *LLM-as-a-Judge* paradigm directly prompts models to evaluate responses, often achieving strong alignment with human preferences and supporting test-time strategies like abstaining from low-quality responses or selecting among candidates (Zheng et al., 2023b; Gu et al., 2024; Zhou et al., 2025b; Ren et al., 2023; Chen et al., 2025; Huang et al., 2025; Zhong et al., 2025; Zhou et al., 2025a). While limitations such as positional bias (Zheng et al., 2023a; Shi et al., 2024) and a preference for longer responses (Hu et al., 2024) do exist, both uncertainty-based and judge-based methods remain effective and have proven valuable for evaluating LLM outputs. Building on these evaluation techniques, a

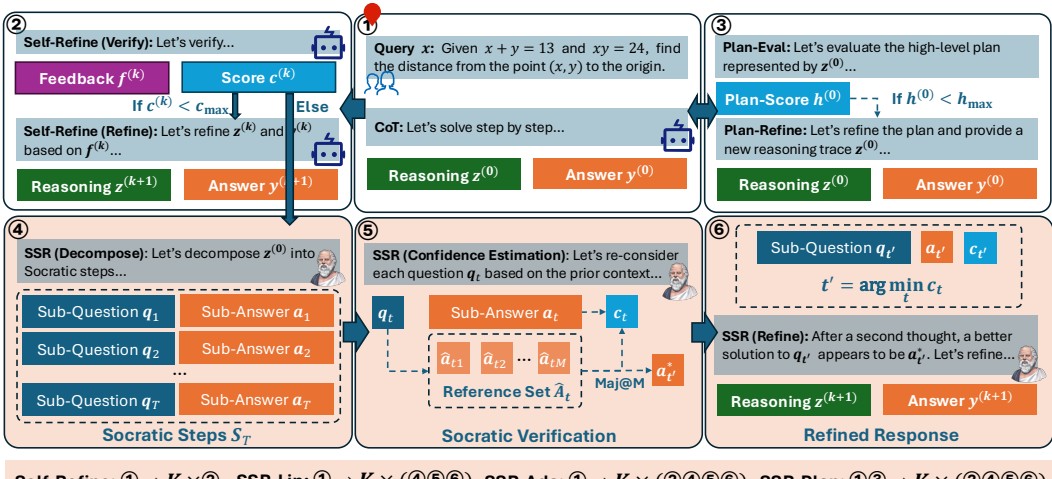

Figure 2: Overview of **S**ocratic **S**elf-**R**efine (**SSR**). **Block ①:** Chain-of-Thought (CoT) reasoning, serves as the starting point for the iterative refinement methods; **Block ②:** Simple Self-Refine, generates feedback and then refines the original response based on the feedback; **Block ③:** Plan refinement, summarizes the high-level plan of a reasoning trace, and refines the plan and the trace if necessary; **Block ④-⑥:** Three building blocks of our SSR, includes Socratic decomposition, Socratic verification, and Socratic refinement. **SSR-Lin**: Linear SSR, faithfully applies three blocks (④-⑥) for $K$ iterations; **SSR-Ada**: Adaptive SSR, only carries out Socratic blocks (④-⑥) when the normal Self-Refine cannot identify any mistakes ($c = c_{\max}$); **SSR-Plan**: Adaptive SSR with Plan Refinement, adds an additional plan refinement round (③) before the full iterative refinement algorithm (④-⑥).

growing body of work extends beyond verification to self-refinement, where LLMs not only diagnose weaknesses in their outputs but also iteratively improve them (Madaan et al., 2023). Early efforts explored direct self-correction based on feedback, while subsequent methods introduced structured search (Zhang et al., 2024), parallel sampling to enrich candidate diversity (Bi et al., 2024; Chen et al., 2025), and reformulation strategies that generate improved sub-questions by incorporating contextual preconditions (Teng et al., 2025). More recent work trains generative verifiers to guide the refinement process (Zhong et al., 2025). Collectively, these approaches demonstrate that refinement transforms passive evaluation into an active mechanism for improving reliability, making it a key step toward controllable and trustworthy reasoning in LLMs.

**Process Evaluation of LLMs.** Verifying only the final outcome of an LLM is insufficient; ensuring reliability requires mechanisms that also evaluate the reasoning process itself. Beyond using human annotations to train process reward models (Lightman et al., 2023; He et al., 2024; Zhang et al., 2025b), the rapid advancement of model capabilities has motivated a growing set of test-time methods for step-level verification. These approaches typically decompose the reasoning trace and assess the correctness of each step to localize errors more accurately (Ling et al., 2023; Miao et al., 2024; Zhao et al., 2025; Mukherjee et al., 2025; Fang et al., 2025). Compared to existing work of process evaluation, our SSR framework adopts a Socratic formulation of reasoning, representing the process as a sequence of question-answer pairs (details in Sec. 3). This structure makes the steps straightforward to re-execute and enables reliable confidence estimation. Crucially, SSR goes beyond verification by producing informative signals that directly support subsequent refinement.

## 3 SOCRATIC SELF-REFINE (SSR)

This section introduces our Socratic Self-Refine (SSR). Sec. 3.1 introduces the fundamental assumption that natural-language reasoning can be described as a Socratic process. Sec. 3.2 presents the core of SSR, including the decomposition into Socratic steps, their verification, and reasoning refinement guided by Socratic confidence scores. Finally, Sec. 3.3 discusses two techniques for practical deployment of SSR: plan refinement and adaptive iteration refinement. **For details of the prompt templates introduced in this section, please refer to Appendix C.3.**

**Notation.** In this paper, scalars are denoted by lowercase letters ($x$), vectors (or token/word sequences) by bold lowercase letters ($\boldsymbol{x}$), random vectors by boldface lowercase letters ($\mathbf{x}$), and matrices (or sets of tokens, words, or phrases) by bold uppercase letters ($\boldsymbol{X}$). We denote by $[m] = 1, 2, \ldots, m$ the set of consecutive integers from 1 to $m$. For consistency, $K$ denotes the total number of refinement iterations, **while** $(k)$ **indicates the current iteration; when unambiguous, we omit** $(k)$ **to reduce clutter.** Finally, $N$ is the number of parallel runs used for test-time scaling.

### 3.1 LLM Reasoning as Socratic Process

**Preliminary of LLM Reasoning.** For problems with short-form ground-truth answers, LLM reasoning can be modeled as marginalization over intermediate natural language reasoning traces $\boldsymbol{z}$ (a sequence of tokens/words) to produce the final answer $\boldsymbol{y}$ (Chen et al., 2024):

$$\pi_{\boldsymbol{\theta}}(\boldsymbol{y} \mid \boldsymbol{x}) = \int \pi_{\boldsymbol{\theta}}(\boldsymbol{y} \mid \boldsymbol{z}, \boldsymbol{x}) \pi_{\boldsymbol{\theta}}(\boldsymbol{z} \mid \boldsymbol{x}) d\boldsymbol{z} \tag{1}$$

Chain-of-Thought (CoT) reasoning (Wei et al., 2022b) approximates this integral with a single sample: the model first generates a reasoning trace $\boldsymbol{z} \sim \pi_{\boldsymbol{\theta}}(\cdot \mid \boldsymbol{x})$ and then derives the final answer $\boldsymbol{y} \sim \pi_{\boldsymbol{\theta}}(\cdot \mid \boldsymbol{z}, \boldsymbol{x})$. Empirically, allocating more computation to approximate Eqn. 1 improves performance. A common strategy is Majority Voting (Maj@N), which averages over multiple sampled reasoning traces (Wang et al., 2022):

$$\pi_{\boldsymbol{\theta}}(\boldsymbol{y} \mid \boldsymbol{x}) \approx \frac{1}{N} \sum_{n=1}^{N} \pi_{\boldsymbol{\theta}}(\boldsymbol{y} \mid \boldsymbol{z}_n, \boldsymbol{x}), \quad \boldsymbol{z}_n \sim \pi_{\boldsymbol{\theta}}(\boldsymbol{z} \mid \boldsymbol{x}). \tag{2}$$

**Reasoning as Socratic Process.** In this paper, we posit that the reasoning process is implicitly modeled as a sequence of goal-setting and problem-solving steps (Simon & Newell, 1971; Russell et al., 1995; Kahneman, 2011; Gandhi et al., 2025); that is, the natural-language reasoning trace $\boldsymbol{z}$ can be viewed as semantically equivalent to a sequence of question-answer pairs. Formally, given a query $\boldsymbol{x}$, we assume that for any reasoning–answer pair $(\boldsymbol{z}, \boldsymbol{y})$, there exists a ground-truth decomposition $\boldsymbol{S}_T \equiv (\boldsymbol{z}, \boldsymbol{y})$ such that [1]

$$\boldsymbol{S}_T = \{\boldsymbol{s}_t \triangleq (\boldsymbol{q}_t, \boldsymbol{a}_t)\}_{t \in [T]}, \tag{3}$$

where each $\boldsymbol{s}_t$ is a *Socratic step*, $\boldsymbol{a}_T = \boldsymbol{y}$ denotes the final answer, and the equivalence $\boldsymbol{S}_T \equiv (\boldsymbol{z}, \boldsymbol{y})$ implies that the oracle probability model $p$ satisfies

$$p(\boldsymbol{z}, \boldsymbol{y} \mid \boldsymbol{x}) = p(\{(\boldsymbol{q}_t, \boldsymbol{a}_t)\}_{t \in [T]} \mid \boldsymbol{x}). \tag{4}$$

Compared with the purely natural-language reasoning process $\boldsymbol{z}$, the explicit sequence of Socratic steps offers clear advantages, most notably, finer-grained modeling and potential control of the reasoning process, enabling verification and intervention. This explicit modeling lies at the heart of our proposed method, Socratic Self-Refine (SSR), which we detail in Sec. 3.2.

### 3.2 Socratic Self-Refine (SSR): Decomposition, Verification, and Refinement

**From Entangled Reasoning to Explicit Socratic Process.** Under the assumption of Eqn. 4, our goal is to recover the full Socratic process $\boldsymbol{S}_T$ from the natural-language reasoning trace $\boldsymbol{z}$. Since no prior work explicitly models this process, and the oracle posterior $p(\boldsymbol{S}_T \mid \boldsymbol{x}, \boldsymbol{y}, \boldsymbol{z})$ is unavailable, we adopt a zero-shot prompting approach with LLMs to decompose $\boldsymbol{z}$ into the Socratic process $\boldsymbol{S}_T$:

$$\boldsymbol{S}_T \sim \pi_{\boldsymbol{\theta}}(\cdot \mid \boldsymbol{x}, \boldsymbol{y}, \boldsymbol{z}, \boldsymbol{x}_{\text{dec}}) \approx p(\cdot \mid \boldsymbol{x}, \boldsymbol{y}, \boldsymbol{z}), \tag{5}$$

where $\boldsymbol{x}_{\text{dec}}$ denotes a decomposition query that prompts the LLM to extract a sequence of sub-questions and their corresponding sub-answers. Leveraging prior work on LLM-based summarization and information extraction (Van Veen et al., 2024), this decomposition can be performed reliably with relatively little overhead.

---

[1]Note that (i) the ground-truth decomposition may not be unique. E.g., $\{\boldsymbol{s}_t\}_{t=1}^{T}$ and $\{\boldsymbol{s}_t\}_{t=2}^{T}$ are both valid decompositions, with the latter representing a coarser process; and (ii) the true structure of the decomposition can be non-linear (Teng et al., 2025), though it can be mapped to a linear form in CoT reasoning.

**LLM Self-Verification on Socratic Steps.** We now leverage the reformulation of the original reasoning trace $\boldsymbol{z}$ into the Socratic process $\boldsymbol{S}_T$ to enable LLM self-verification. The joint probability distribution of $\boldsymbol{S}_T$ can be factorized into a product of conditional probabilities:

$$\pi_{\boldsymbol{\theta}}(\boldsymbol{S}_T \mid \boldsymbol{x}) = \pi_{\boldsymbol{\theta}}(\{(\boldsymbol{q}_t, \boldsymbol{a}_t)\}_{t \in [T]} \mid \boldsymbol{x}) = \prod_{t=1}^{T} \underbrace{\pi_{\boldsymbol{\theta}}(\boldsymbol{q}_t \mid \{\boldsymbol{s}_i\}_{i<t}, \boldsymbol{x})}_{t\text{-th step planning}} \cdot \underbrace{\pi_{\boldsymbol{\theta}}(\boldsymbol{a}_t \mid \boldsymbol{q}_t, \{\boldsymbol{s}_i\}_{i<t}, \boldsymbol{x})}_{t\text{-th step execution}}, \quad (6)$$

where $\{\boldsymbol{s}_i\}_{i<1} \triangleq \emptyset$. This factorization captures our core perspective on LLM reasoning: at each step, the model first plans by formulating the next sub-question, and then executes by generating the corresponding sub-answer. Such a sequential formulation naturally lends itself to Monte Carlo search over possible reasoning trajectories, where the two types of actions are sub-question generation ($\boldsymbol{q}$) and sub-answer generation ($\boldsymbol{a}$) (Qi et al., 2024; Acuna et al., 2025). However, as the modern LLMs typically do not undergo the training of explicitly proposing and answering the next probable sub-questions, this approach might be less effective.

SSR evaluates the confidence of each sub-answer $\boldsymbol{a}_t$ given the current sub-question $\boldsymbol{q}_t$, the original query $\boldsymbol{x}$, and the context of the previous Socratic steps $(\boldsymbol{q}_i, \boldsymbol{a}_i)_{i<t}$. Specifically, we encode all relevant information into the context and ask the LLM to solve each sub-question independently $M$ times. The resulting answers form a reference set

$$\widehat{\boldsymbol{A}}_t = \{\widehat{\boldsymbol{a}}_{ti}\}_{i \in [M]}, \quad \widehat{\boldsymbol{a}}_{ti} \sim \pi_{\boldsymbol{\theta}}(\cdot \mid \boldsymbol{q}_t, \{\boldsymbol{s}_i\}_{i<t}, \boldsymbol{x}). \quad (7)$$

We then compare the original $t$-th step sub-answer $\boldsymbol{a}_t$ with $\widehat{\boldsymbol{A}}_t$ and estimate the confidence score as

$$c_t = \frac{1}{M} \sum_{i=1}^{M} \mathbb{1}_{\boldsymbol{a}_t = \widehat{\boldsymbol{a}}_{ti}}, \quad \forall t \in [T]. \quad (8)$$

For mathematical problems, intermediate sub-answers can be restricted to mathematical expressions through simple prompting, allowing for deterministic equivalence checking. In practice, however, we find that this restriction does not consistently constrain LLM outputs We therefore resort to LLM self-evaluation, producing confidence scores directly with a context-free confidence estimation prompt $\boldsymbol{x}_{\text{conf}}$:

$$\boldsymbol{C}_T = \{c_t \sim \pi_{\boldsymbol{\theta}}(\cdot \mid \boldsymbol{a}_t, \widehat{\boldsymbol{A}}_t, \boldsymbol{x}_{\text{conf}})\}_{t \in [T]}. \quad (9)$$

These confidence scores are then used to guide refinement of the current-round reasoning and can also be aggregated to reflect overall response quality, which supports sample selection in our test-time scaling experiments (Sec. 4.5). It is worth noting that we enforce strict context management during confidence estimation: the prompt includes only the candidate sub-answer and the reference answer set, with no additional information. This design has two motivations. First, we assume that judging equivalence between expressions can be done in a context-free manner, i.e., with only the expressions. Second, isolating the context helps control the computation budget.

**LLM Self-Refinement with Socratic Steps.** Once the confidence scores of all Socratic steps are estimated, we use them to guide reasoning refinement. In SSR, we first identify the step $t' = \arg\min_t \{c_t\}_{t \in [T]}$ with the lowest confidence score $c_{t'}$. We then apply majority voting over its reference answer set $\widehat{\boldsymbol{A}}_{t'}$ to obtain a refined sub-answer:

$$\boldsymbol{a}_{t'}^* = \arg\max_{\boldsymbol{a}} \pi_{\boldsymbol{\theta}}(\boldsymbol{a} \mid \boldsymbol{q}_{t'}, \{\boldsymbol{s}_i\}_{i<t'}, \boldsymbol{x}) \approx \text{maj\_vote}(\widehat{\boldsymbol{A}}_{t'}). \quad (10)$$

This refined sub-answer is injected into the iteration-$k$ refinement of $(\boldsymbol{z}^{(k)}, \boldsymbol{y}^{(k)})$, producing the next iteration:

$$(\boldsymbol{z}^{(k+1)}, \boldsymbol{y}^{(k+1)}) \sim \pi_{\boldsymbol{\theta}}(\cdot \mid \boldsymbol{x}, \boldsymbol{y}^{(k)}, \boldsymbol{z}^{(k)}, \underbrace{\boldsymbol{q}_{t'}^{(k)}, \boldsymbol{a}_{t'}^{(k)}, \boldsymbol{a}_{t'}^{*(k)}}_{\text{Socratic Feedback}}, \boldsymbol{x}_{\text{ref}}), \quad (11)$$

where the triplet $(\boldsymbol{q}_{t'}^{(k)}, \boldsymbol{a}_{t'}^{(k)}, \boldsymbol{a}_{t'}^{*(k)})$ is called Socratic Feedback, the template of which can be found in Appendix C.3, and $\boldsymbol{x}_{\text{ref}}$ is the refinement query that prompts the LLM to revise for a new reasoning trace $\boldsymbol{z}^{(k+1)}$ that leads to $\boldsymbol{a}_{t'}^{*(k)}$. Because most modern LLMs are trained with instruction tuning (Wei et al., 2021) and preference tuning (Ouyang et al., 2022), both relying on chain-of-thought–like structures, the direct injection of the Socratic process in unnatural formats (e.g., JSON) might disrupt reasoning. Our design principle in SSR is therefore to minimize format disruption and to inject only the necessary information into the refinement context. For a detailed analysis of this issue, please refer to Sec. 4.4. *We refer to the variant that directly combines the three steps described above as Linear SSR (SSR-Lin).*

### 3.3 SSR Deployment: Better Efficiency and Beyond Step-Level Refinement

**Improving the Efficiency of SSR with Gating Self-Refine.** Applying fine-grained, step-level SSR at every refinement step can be costly compared to other iterative refinement frameworks (Madaan et al., 2023; Teng et al., 2025). To balance efficiency and accuracy, we adopt a gating mechanism that combines Self-Refine (Madaan et al., 2023) with SSR. In deployment, Self-Refine serves as the default refinement method, while SSR is invoked only when Self-Refine fails to identify mistakes in the reasoning trace or when the response is already correct. Because these two situations cannot be distinguished in advance, applying SSR in the latter case incurs only a minor additional cost, while in the former case it provides an extra layer of safety. Compared to SSR only, this approach reduces overhead while preserving the advantages of SSR's ability of fine-grained step-level verification. *We denote SSR with this adaptive gating mechanism as Adaptive SSR (SSR-Ada).*

**SSR Planning Refinement.** Our current SSR relies on two implicit assumptions about reasoning planning: (i) response quality evaluation is independent of high-level planning, and (ii) refinement focuses only on execution accuracy. These restrictions may limit the performance of SSR. By assuming probabilistic independence between each sub-question $q_t$ and the preceding answers $\{a_i\}_{i<t}$, the factorization [2] in Eqn. 6 can be simplified as

$$\pi_{\boldsymbol{\theta}}(\boldsymbol{S}_T \mid \boldsymbol{x}) = \underbrace{\pi_{\boldsymbol{\theta}}(\{\boldsymbol{q}_t\}_{t\in[T]} \mid \boldsymbol{x})}_{\text{high-level planning}} \cdot \underbrace{\prod_{t=1}^{T} \pi_{\boldsymbol{\theta}}(\boldsymbol{a}_t \mid \boldsymbol{q}_t, \{\boldsymbol{s}_i\}_{i<t}, \boldsymbol{x})}_{\text{sequential execution}}. \tag{12}$$

To ensure the reliability of high-level planning before applying step-level SSR, while keeping the overhead modest compared to other baselines, we perform only one round of plan refinement. Unlike our main SSR procedure, we do not repeatedly sample rollouts or evaluate their quality. Instead, we directly prompt the LLM to judge whether the high-level plan (a sequence of sub-questions or their natural-language description) is sufficiently sound for the subsequent execution. *We denote SSR-Ada plus this plan refinement as SSR-Plan.* **For the detailed algorithmic description of our SSR, please refer to Algorithm 1 in the Appendix.**

## 4 Experiments

We evaluate our SSR's effectiveness through comprehensive experiments, covering experimental setup (Sec. 4.1), main results on the mathematical and logical reasoning benchmarks (Sec. 4.2), , ablation studies on the choice of incorporating Socratic content into refinement (Sec. 4.4), and test-time scaling effect of our SSR (Sec. 4.5). **For additional results, please refer to Appendix D.**

### 4.1 Settings

**Models.** We use the latest `GPT-4.1-nano` (general-purpose) and `GPT-5-mini` (reasoning) models from OpenAI as our LLM backbones, chosen for their balanced capabilities in instruction following and reasoning. We additionally include the results of the strong LLM `Gemini-2.5-Flash` (Comanici et al., 2025) in Appendix D.1.

**Datasets.** We benchmark the reasoning frameworks on two categories of datasets: **mathematical reasoning** and **logical reasoning**. For mathematical reasoning, we adopt the challenging Level-5 subset of the MATH dataset (**MATH-Level-5**) with numerical answers (Hendrycks et al., 2021), American Invitational Mathematics Examination (**AIME**) from 2024 and 2025 (AIME-Team, 2025), and the math subset of Humanity's Last Exam (HLE) (Phan et al., 2025).

**Evaluation.** We adopt the library of Math-Verify (Hugging Face, 2024) for matching the candidate and ground-truth answer (except for the non-numerical subset of HLE). For logical reasoning, we use the synthetic reasoning-gym environment (Stojanovski et al., 2025) to generate sub-tasks including the Zebra-Puzzle and Mini-Sudoku, where we use exact string matching and rule-based verifier as the evaluation, respectively.

**Baselines.** We compare our SSR against several iterative refinement–based test-time LLM reasoning frameworks. **Self-Refine** (Madaan et al., 2023) iteratively generates feedback for a given response

---

[2]Under this assumption, we posit that the LLM establishes an overall plan before generating the actual response (Ye et al., 2024; Lindsey et al., 2025).

Table 1: **Last-Round Performance of Iterative Refinement-Based Reasoning Methods. LR-Acc:** Last-round refinement's accuracy, yielded by 10 repeated experiments; **LR-Maj@5:** Last-round refinement's accuracy of majority voting with 5 samples in parallel, yielded by 50 repeated experiments. **Boldface** and underlining denote the best and the second-best performance, respectively.

| Method | MATH-Level-5 | | AIME24 | | AIME25 | | Zebra-Puzzle | | Mini-Sudoku | |
|---|---|---|---|---|---|---|---|---|---|---|
| | LR-Acc | LR-Maj@5 | LR-Acc | LR-Maj@5 | LR-Acc | LR-Maj@5 | LR-Acc | LR-Maj@5 | LR-Acc | LR-Maj@5 |
| GPT-4.1-nano | | | | | | | | | | |
| CoT | 74.88$_{\pm1.35}$ | 82.32$_{\pm1.11}$ | 27.00$_{\pm4.58}$ | 32.80$_{\pm2.15}$ | 23.00$_{\pm3.48}$ | 26.93$_{\pm2.97}$ | 55.20$_{\pm3.28}$ | 56.56$_{\pm2.44}$ | 47.40$_{\pm3.35}$ | 66.04$_{\pm2.69}$ |
| Self-Refine | 68.69$_{\pm1.15}$ | 79.81$_{\pm0.75}$ | 28.00$_{\pm4.99}$ | 34.33$_{\pm3.00}$ | 22.67$_{\pm2.91}$ | 28.33$_{\pm3.42}$ | 53.50$_{\pm1.96}$ | 56.08$_{\pm1.93}$ | 53.60$_{\pm4.59}$ | 73.04$_{\pm3.21}$ |
| Debate | **79.28$_{\pm0.86}$** | **84.08$_{\pm0.76}$** | 27.00$_{\pm4.82}$ | 32.40$_{\pm3.13}$ | **26.67$_{\pm2.58}$** | 27.60$_{\pm2.75}$ | 54.70$_{\pm3.29}$ | 57.16$_{\pm2.66}$ | **60.80$_{\pm4.81}$** | **78.38$_{\pm2.75}$** |
| MCTSr | 74.02$_{\pm1.12}$ | 83.01$_{\pm0.81}$ | 23.67$_{\pm4.33}$ | 30.47$_{\pm3.13}$ | 20.00$_{\pm4.94}$ | 25.73$_{\pm4.22}$ | 54.90$_{\pm2.47}$ | 54.88$_{\pm2.45}$ | 53.33$_{\pm1.63}$ | 73.84$_{\pm2.43}$ |
| AoT | 75.15$_{\pm1.00}$ | 82.83$_{\pm0.83}$ | 21.11$_{\pm4.97}$ | 25.67$_{\pm3.61}$ | 21.33$_{\pm3.06}$ | 25.53$_{\pm3.75}$ | 29.33$_{\pm3.16}$ | 43.60$_{\pm2.65}$ | 42.80$_{\pm2.96}$ | 65.08$_{\pm2.26}$ |
| SSR-Lin (Ours) | 77.06$_{\pm0.93}$ | 83.64$_{\pm0.69}$ | **32.67$_{\pm3.59}$** | **39.93$_{\pm3.23}$** | 24.00$_{\pm4.67}$ | 27.33$_{\pm4.06}$ | 54.60$_{\pm2.20}$ | 54.10$_{\pm2.09}$ | 53.10$_{\pm2.47}$ | 72.76$_{\pm2.55}$ |
| SSR-Ada (Ours) | 75.70$_{\pm1.31}$ | 82.71$_{\pm0.90}$ | 29.67$_{\pm6.74}$ | 37.47$_{\pm4.25}$ | 24.67$_{\pm3.06}$ | **28.80$_{\pm3.38}$** | 54.30$_{\pm1.90}$ | 55.14$_{\pm1.71}$ | 51.50$_{\pm4.41}$ | 73.22$_{\pm3.37}$ |
| SSR-Plan (Ours) | 76.01$_{\pm0.57}$ | 83.75$_{\pm0.74}$ | 27.33$_{\pm5.73}$ | 35.80$_{\pm3.39}$ | 22.33$_{\pm3.67}$ | 27.53$_{\pm4.46}$ | **56.90$_{\pm3.11}$** | **57.30$_{\pm2.39}$** | 47.70$_{\pm4.22}$ | 66.46$_{\pm4.61}$ |
| GPT-5-mini | | | | | | | | | | |
| CoT | 82.95$_{\pm1.02}$ | 90.05$_{\pm0.54}$ | 50.67$_{\pm4.67}$ | 60.87$_{\pm3.93}$ | 37.00$_{\pm6.57}$ | 49.80$_{\pm4.19}$ | 82.80$_{\pm2.71}$ | 91.00$_{\pm1.30}$ | 42.40$_{\pm2.42}$ | 61.96$_{\pm3.19}$ |
| Self-Refine | 87.02$_{\pm1.40}$ | 94.11$_{\pm0.47}$ | 63.33$_{\pm4.94}$ | 74.40$_{\pm3.74}$ | 53.67$_{\pm6.23}$ | 68.33$_{\pm3.48}$ | 82.00$_{\pm2.61}$ | 92.64$_{\pm1.61}$ | 63.60$_{\pm3.35}$ | 93.82$_{\pm1.35}$ |
| Debate | 90.62$_{\pm0.94}$ | 93.47$_{\pm0.46}$ | 63.67$_{\pm3.79}$ | 74.13$_{\pm3.44}$ | 53.33$_{\pm3.33}$ | 61.87$_{\pm3.21}$ | **91.20$_{\pm1.72}$** | **93.74$_{\pm1.07}$** | 90.40$_{\pm3.95}$ | 98.54$_{\pm1.31}$ |
| MCTSr | 87.42$_{\pm0.89}$ | 92.91$_{\pm0.71}$ | 57.00$_{\pm5.67}$ | 68.87$_{\pm4.35}$ | 46.97$_{\pm6.11}$ | 55.40$_{\pm4.76}$ | 83.00$_{\pm1.90}$ | 89.82$_{\pm1.49}$ | 61.40$_{\pm6.17}$ | 89.68$_{\pm2.56}$ |
| AoT | 80.56$_{\pm0.63}$ | 88.84$_{\pm0.60}$ | 46.67$_{\pm5.16}$ | 57.00$_{\pm3.21}$ | 33.00$_{\pm6.05}$ | 43.60$_{\pm3.82}$ | 65.30$_{\pm3.07}$ | 74.78$_{\pm2.07}$ | 61.70$_{\pm3.72}$ | 82.72$_{\pm2.75}$ |
| SSR-Lin (Ours) | 88.36$_{\pm1.06}$ | 93.01$_{\pm0.63}$ | 64.00$_{\pm5.12}$ | 74.60$_{\pm4.10}$ | 55.67$_{\pm4.48}$ | 65.47$_{\pm3.76}$ | 87.70$_{\pm2.97}$ | 93.70$_{\pm1.76}$ | 93.60$_{\pm1.69}$ | 99.70$_{\pm0.54}$ |
| SSR-Ada (Ours) | 91.57$_{\pm0.51}$ | 95.62$_{\pm0.35}$ | 68.67$_{\pm4.52}$ | 75.93$_{\pm3.08}$ | 60.33$_{\pm4.58}$ | 70.13$_{\pm3.46}$ | 87.30$_{\pm2.53}$ | 93.00$_{\pm1.69}$ | **96.10$_{\pm2.07}$** | 99.98$_{\pm0.14}$ |
| SSR-Plan (Ours) | **92.16$_{\pm0.67}$** | **95.93$_{\pm0.30}$** | **69.67$_{\pm4.82}$** | **79.00$_{\pm3.48}$** | **62.00$_{\pm6.18}$** | **71.53$_{\pm5.26}$** | 88.00$_{\pm1.55}$ | 93.20$_{\pm1.08}$ | 94.80$_{\pm2.48}$ | **100.00$_{\pm0.00}$** |

and updates the response based on this self-feedback. **Debate** (Du et al., 2023) employs a multi-agent framework in which each agent iteratively refines or defends its response by engaging with the responses of peer agents. Monte Carlo Tree Self-Refine (**MCTSr**) (Zhang et al., 2024) treats the full generation as a node and the self-refine step as an edge, applying Monte Carlo Tree Search (MCTS) to search for the best response. Atom of Thoughts (**AoT**) (Teng et al., 2025) incrementally constructs a Directed Acyclic Graph (DAG) of reasoning, contracts intermediate results into improved sub-questions, and solves them step by step. We do not include parallel sampling–based baselines such as Forest of Thoughts (**FoT**) (Bi et al., 2024), since these approaches are complementary to iterative refinement methods. Their benefits are instead reflected through the **Maj@5** metric in Table 1.

**Implementation of SSR.** We implement and evaluate three variants of SSR in Sec. 3.3. Linear SSR (**SSR-Lin**) applies Socratic self-refine at every iteration, making it the most costly but also the most thorough approach to step-level fine-grained refinement. Adaptive SSR (**SSR-Ada**) first applies the basic Self-Refine; if the feedback reveals clear and critical errors, the feedback is directly adopted, while if no errors are detected, the method falls back to Socratic self-refine. SSR with plan refinement (**SSR-Plan**) adds an initial round of plan refinement before the step-level Socratic self-refine, thereby equipping SSR with high-level refinement capabilities. **For more details, please refer to Appendix C.2.**

## 4.2 SSR'S STEP-LEVEL VERIFICATION LEADS TO CONSISTENT PERFORMANCE GAINS

**Table 1 and Table 2 show results on comprehensive metrics for various methods.**

Overall, the proposed SSR variants bring substantial improvements when powered by the strong GPT-5-mini. Across all tasks, SSR consistently surpasses competitive baselines, yielding clear gains in both LR-Acc and LR-Maj@5. Notably, SSR-Plan achieves the best or second-best results in nearly every setting, with particularly large margins on challenging mathematical reasoning benchmarks like AIME. This highlights that structured preliminary planning amplifies the benefits of iterative refinement, even when starting from already strong GPT-5-mini reasoning capabilities. Our framework also demonstrates effectiveness on the weaker GPT-4.1-nano backbone. Despite its limited reasoning capacity, all three SSR variants in general improve performance over baselines, underscoring that our refinement strategies generalize across model scales. This implies a viable path of adopting our SSR to boost smaller, resource-efficient models.

Second, the results in Table 2 show that SSR maintains superiority under upper-bound evaluation metrics. Both BoK-Acc and Pass@K demonstrate that SSR variants yield higher-quality and diverse refinement trajectories compared to baselines. Again, SSR-Plan often achieves the best results, while

Table 2: **Upper-Bound Performance of Iterative Refinement-Based Reasoning Methods. BoK-Acc:** Best-of-K refinements' accuracy, yielded by prompting LLM-as-a-Judge (Gu et al., 2024) for selecting the best answer out of K iterations of refinement; **Pass@K:** Pass-at-K refinements' accuracy (at lease one of K iterations gets the answer correct). Both experiments are repeated for 10 times. **Boldface** and underlining denote the best and the second-best performance, respectively.

| Method | MATH-Level-5 | | AIME24 | | AIME25 | | Zebra-Puzzle | | Mini-Sudoku | |
|---|---|---|---|---|---|---|---|---|---|---|
| | *BoK-Acc* | *Pass@K* | *BoK-Acc* | *Pass@K* | *BoK-Acc* | *Pass@K* | *BoK-Acc* | *Pass@K* | *BoK-Acc* | *Pass@K* |
| GPT-4.1-nano | | | | | | | | | | |
| CoT | $74.88_{\pm1.35}$ | - | $27.00_{\pm4.58}$ | - | $23.00_{\pm3.48}$ | - | $55.20_{\pm3.28}$ | - | $47.40_{\pm3.35}$ | - |
| Self-Refine | $76.48_{\pm0.95}$ | $81.60_{\pm0.82}$ | $30.67_{\pm5.54}$ | $31.67_{\pm5.00}$ | $23.67_{\pm4.07}$ | $26.00_{\pm4.90}$ | $55.60_{\pm3.77}$ | $59.60_{\pm2.37}$ | $56.90_{\pm5.84}$ | $65.70_{\pm3.55}$ |
| Debate | $\mathbf{79.62_{\pm0.79}}$ | $84.51_{\pm1.01}$ | $29.00_{\pm3.00}$ | $35.33_{\pm3.40}$ | $\underline{26.00_{\pm3.89}}$ | $31.00_{\pm3.67}$ | $\mathbf{56.80_{\pm2.79}}$ | $\mathbf{68.50_{\pm4.06}}$ | $\mathbf{63.50_{\pm3.96}}$ | $70.70_{\pm3.44}$ |
| AoT | $\underline{79.37_{\pm1.54}}$ | $\mathbf{87.28_{\pm0.64}}$ | $23.33_{\pm5.21}$ | $33.70_{\pm3.99}$ | $24.33_{\pm4.48}$ | $29.33_{\pm5.33}$ | $37.33_{\pm3.20}$ | $63.22_{\pm3.64}$ | $50.20_{\pm5.08}$ | $\mathbf{76.00_{\pm3.26}}$ |
| SSR-Lin (Ours) | $78.03_{\pm1.00}$ | $82.97_{\pm0.98}$ | $\mathbf{33.33_{\pm4.22}}$ | $\mathbf{38.33_{\pm5.63}}$ | $\mathbf{26.67_{\pm3.94}}$ | $32.00_{\pm4.00}$ | $55.90_{\pm2.74}$ | $65.40_{\pm1.96}$ | $58.20_{\pm3.71}$ | $\underline{75.40_{\pm3.38}}$ |
| SSR-Ada (Ours) | $78.05_{\pm1.37}$ | $85.14_{\pm0.56}$ | $\underline{31.67_{\pm5.82}}$ | $\underline{36.33_{\pm5.67}}$ | $25.67_{\pm4.48}$ | $32.00_{\pm3.40}$ | $55.30_{\pm1.19}$ | $62.80_{\pm2.04}$ | $56.70_{\pm3.44}$ | $74.20_{\pm4.94}$ |
| SSR-Plan (Ours) | $78.40_{\pm1.10}$ | $\underline{85.27_{\pm0.47}}$ | $31.33_{\pm5.42}$ | $35.67_{\pm4.23}$ | $24.33_{\pm3.67}$ | $\mathbf{34.33_{\pm5.17}}$ | $\underline{56.60_{\pm3.58}}$ | $\underline{64.60_{\pm3.01}}$ | $56.40_{\pm4.05}$ | $73.70_{\pm2.37}$ |
| GPT-5-mini | | | | | | | | | | |
| CoT | $82.95_{\pm1.02}$ | - | $50.67_{\pm4.67}$ | - | $37.00_{\pm6.57}$ | - | $82.80_{\pm2.71}$ | - | $42.40_{\pm2.42}$ | - |
| Self-Refine | $89.40_{\pm1.00}$ | $91.59_{\pm0.83}$ | $61.33_{\pm4.00}$ | $68.00_{\pm3.71}$ | $51.67_{\pm6.87}$ | $56.67_{\pm6.67}$ | $90.90_{\pm2.21}$ | $91.30_{\pm1.79}$ | $85.70_{\pm3.23}$ | $83.30_{\pm2.19}$ |
| Debate | $90.43_{\pm0.88}$ | $91.70_{\pm0.79}$ | $64.00_{\pm4.16}$ | $64.67_{\pm4.27}$ | $53.00_{\pm2.77}$ | $55.00_{\pm2.69}$ | $91.70_{\pm1.62}$ | $93.70_{\pm1.35}$ | $90.20_{\pm3.54}$ | $91.80_{\pm3.57}$ |
| AoT | $85.87_{\pm0.49}$ | $91.38_{\pm0.80}$ | $56.67_{\pm6.15}$ | $61.67_{\pm5.82}$ | $39.33_{\pm3.27}$ | $49.00_{\pm5.39}$ | $88.80_{\pm1.94}$ | $\mathbf{93.50_{\pm1.43}}$ | $93.70_{\pm1.73}$ | $90.70_{\pm2.15}$ |
| SSR-Lin (Ours) | $88.16_{\pm1.31}$ | $89.54_{\pm1.25}$ | $65.33_{\pm5.42}$ | $67.00_{\pm3.79}$ | $55.33_{\pm7.02}$ | $59.00_{\pm5.17}$ | $\underline{92.20_{\pm2.23}}$ | $93.20_{\pm2.60}$ | $95.30_{\pm1.19}$ | $95.50_{\pm1.57}$ |
| SSR-Ada (Ours) | $\underline{93.14_{\pm0.52}}$ | $\underline{94.63_{\pm0.36}}$ | $\mathbf{71.67_{\pm4.28}}$ | $\mathbf{74.00_{\pm4.90}}$ | $\underline{61.00_{\pm4.73}}$ | $66.00_{\pm3.89}$ | $91.80_{\pm1.89}$ | $93.00_{\pm1.84}$ | $\underline{98.20_{\pm1.25}}$ | $\underline{98.10_{\pm1.45}}$ |
| SSR-Plan (Ours) | $\mathbf{93.48_{\pm0.52}}$ | $\mathbf{95.05_{\pm0.34}}$ | $\underline{71.00_{\pm4.48}}$ | $\underline{73.67_{\pm4.07}}$ | $\mathbf{65.67_{\pm6.16}}$ | $\mathbf{69.67_{\pm5.26}}$ | $\mathbf{92.30_{\pm1.62}}$ | $\underline{93.30_{\pm1.79}}$ | $\mathbf{98.70_{\pm1.00}}$ | $\mathbf{98.30_{\pm1.19}}$ |

Table 3: Accuracies (%) of iterative refinement-based reasoning methods on the 915-question text-only math subset of Humanity's Last Exam (**HLE**) (Phan et al., 2025), with `GPT-5-mini` and `GPT-5` (medium reasoning, medium verbosity).

| Model | CoT | Self-Refine | SSR-Plan (Ours) |
|---|---|---|---|
| GPT-5-mini | 16.18 | 18.58 (+2.40) | **21.53 (+5.35)** |
| GPT-5 | 27.98 | 26.57 (-1.41) | **29.61 (+1.63)** |

`SSR`-Ada provides a favorable trade-off between efficiency and accuracy, confirming the value of adaptively combining Self-Refine with Socratic refinement.

Finally, the comparison across reasoning categories highlights complementary strengths. In mathematical reasoning, `SSR` gains from explicit verification and refinement of sub-answers, which reduces cascading errors in long derivations. In logical reasoning tasks such as Zebra-Puzzle and Mini-Sudoku, where execution accuracy dominates, step-level Socratic verification also proves highly effective, often yielding substantial improvements over baselines.

Overall, the experiments confirm that the explicit modeling and verification of Socratic steps in `SSR` provides more reliable and controllable refinement than existing iterative approaches, with `SSR`-Plan standing out as the most robust variant.

### 4.3 WHEN SELF-REFINE BREAKS, SSR THRIVES: EXTENDING SSR TO CHALLENGING TASKS

In this section, we evaluate the effectiveness of `SSR` using more recent and stronger models, which require more challenging tasks to avoid performance saturation. Specifically, we employ the full `GPT-5` model in medium reasoning and medium verbosity modes, *without tool calling or web searching*, and conduct experiments on Humanity's Last Exam (**HLE**) (Phan et al., 2025). Due to budget constraints, we restrict our evaluation to the 915-question text-only math subset of HLE, where all questions are purely textual. We further divide this subset into two partitions based on whether the ground-truth answers are numerical. For the 478-example numerical partition, we follow the Math-Verify (Hugging Face, 2024) evaluation protocol described above, while for the 437-example non-numerical partition, we adopt the official LLM-as-a-Judge evaluation protocol with `GPT-5`. The remaining settings are kept identical to those described earlier. **See Appendix D.2 for details.**

**The results are reported in Table 3.** Our `SSR` framework consistently outperforms both Chain-of-Thought (CoT) and Self-Refine baselines across model scales. With `GPT-5-mini`, `SSR` achieves 21.53% accuracy, surpassing CoT by 5.35 points and Self-Refine by 2.95 points, indicating that our two-level refinement reasoning framework is particularly beneficial for smaller models with limited reasoning capability. When scaled to the full `GPT-5`, `SSR` still yields a gain of 3.04 points over Self-Refine and 1.63 over CoT, suggesting that our approach complements intrinsic reasoning

abilities rather than relying on model size alone. Notably, it remains effective even for `GPT-5` where vanilla Self-Refine fails to generalize. These results confirm that `SSR` effectively enhances iterative reasoning robustness for stronger frontier models like `GPT-5` even in challenging tasks such as HLE.

## 4.4 ANALYSIS: SSR CONTEXT MANAGEMENT

As discussed in Sec. 3.2, representing a natural language reasoning trace $z$ as a Socratic process $S_T$ requires careful consideration, since it introduces a distributional shift between the model's training data and our artificially structured context. In this subsection, we explore alternative ways of integrating the Socratic process $S_T$ into reasoning refinement. Specifically, we focus on two key aspects:

Table 4: **Ablation Study on `SSR` Context Management,** evaluated on `GPT-5-mini`.

| Method | Refinement | Context | Dataset | |
|---|---|---|---|---|
| | | | AIME24 | AIME25 |
| CoT | - | - | $50.67_{\pm4.67}$ | $37.00_{\pm6.57}$ |
| Self-Refine | Reflection | Natural | $63.33_{\pm4.94}$ | $53.67_{\pm6.23}$ |
| | Reflection | Natural | $\mathbf{69.67_{\pm4.82}}$ | $\mathbf{62.00_{\pm6.18}}$ |
| `SSR`-Plan | Reflection | Socratic | $67.67_{\pm4.48}$ | $60.33_{\pm4.82}$ |
| (Ours) | Intervention | Natural | $54.67_{\pm4.76}$ | $42.67_{\pm7.12}$ |
| | Intervention | Socratic | $57.00_{\pm8.09}$ | $52.00_{\pm5.62}$ |

- **Context Format** *(Natural / Socratic)*: Iterative refinement can be performed using only the Socratic steps $S_T$ *(Socratic)*, discarding the original natural language reasoning trace $z$; or conversely, using only $z$ without the Socratic decomposition *(Natural)*.
- **Context Completeness** *(Reflection / Intervention)*: Since LLM chain-of-thought reasoning assumes linear dependencies, once the first problematic step $s_{t'}$ is identified, later steps can be discarded. Refinement may then intervene directly at the error location *(Intervention)*, avoiding unnecessary tokens, unlike `SSR` which refines after the full reasoning is completed *(Reflection)*.

**The results are reported in Table 4.** From the table, we observe that our implementation adopted in the main experiments (*reflection + natural context*) yields the strongest results (69.67 on AIME24 and 62.00 on AIME25), outperforming both Self-Refine and other variants of `SSR`. This suggests that *preserving the original reasoning trace while applying reflection-based precise step-level refinement provides the model with richer contextual cues for error correction.*

Under reflection, replacing the natural context with the Socratic context yields slightly weaker but still competitive results, suggesting that while Socratic decomposition supports step-level analysis, it may miss some nuances of natural language reasoning. In contrast, intervention-based refinement consistently underperforms, as prematurely truncating the reasoning trace discards useful contextual information and leads to weaker refinements.

## 4.5 ANALYSIS: TEST-TIME SCALING OF SSR

In this subsection, we investigate whether the performance gains of `SSR` can be sustained under increased test-time compute. Test-time scaling for iterative refinement generally follows two orthogonal approaches: *(i) sequential scaling*, which increases the number of refinement iterations, and *(ii) parallel scaling*, which runs multiple refinements in parallel and aggregates the outputs.

In our study, sequential scaling extends the number of iterations by $3\times$, with performance reported as Last-Round Accuracy (LR-Acc). Parallel scaling increases the number of parallel samples to 64, also reporting aggregated LR-Acc. Experiments are conducted on AIME25 with the `GPT-5-mini` backbone (low-reasoning, low-verbosity). As baselines, we include basic CoT and Self-Refine. For Self-Refine and `SSR`, we perform an additional self-evaluation on the final

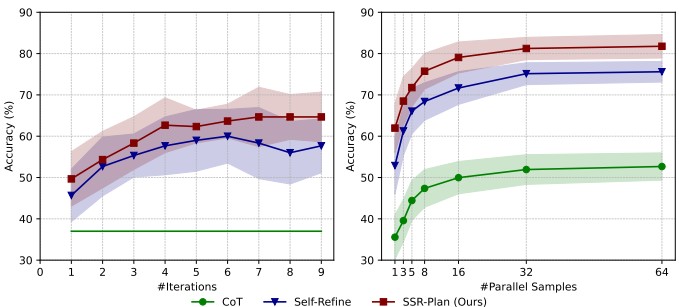

Figure 3: **Performance of Sequential (Left) and Parallel (Right) Test-Time Scaling,** evaluated on AIME25 (AIME-Team, 2025) with GPT-5-mini low-reasoning low-verbosity mode.

reasoning trace and use the resulting 0-5 score for weighted best-of-$N$ (WBoN). For CoT, we apply majority voting (Maj@$N$).

**The results are shown in Fig. 3.** *On the sequential scaling side (left),* SSR consistently outperforms Self-Refine across all iteration counts. Accuracy improves steadily as the number of refinement iterations increases, with SSR showing stronger gains and greater stability than Self-Refine. In contrast, Self-Refine benefits from additional iterations but plateaus at a lower accuracy, confirming that iterative refinement is essential for improvement. *On the parallel scaling side (right),* all methods improve as the number of parallel samples increases, but SSR maintains a clear margin over Self-Refine and CoT. Notably, SSR reaches higher accuracy levels more quickly, suggesting that its Socratic step-level verification yields more consistent refinements, which aggregate effectively under parallel sampling. Self-Refine shows moderate improvements with larger sample sizes, while CoT lags behind, highlighting the importance of structured refinement.

In both parallel and sequential scaling, SSR consistently outperforms Self-Refine and vanilla CoT, even when the baselines are given additional compute and cost, as shown in Fig. 1 and Appendix D.3. This demonstrates that SSR makes more efficient use of available resources. Unlike Self-Refine, whose improvements plateau quickly, SSR continues to gain with further iterations, indicating that confidence-aware step refinement enables more robust and scalable performance under larger budgets.

## 5 CONCLUSION

In this paper, we introduced Socratic Self-Refine (SSR), a novel iterative refinement framework that leverages step-level Socratic decomposition to evaluate and improve LLM reasoning. By explicitly modeling reasoning as a sequence of sub-questions and sub-answers, SSR provides fine-grained confidence estimation and enables targeted refinements where errors are most likely to occur. Across both mathematical and logical reasoning benchmarks, SSR consistently outperforms existing iterative refinement baselines, with the plan-refinement variant achieving the most robust gains. Beyond empirical performance, SSR highlights the importance of moving from outcome-level to process-level evaluation. By treating reasoning as a verifiable sequence of interpretable steps, our framework makes LLM outputs more transparent and opens the door to interventions that are more systematic than ad hoc self-correction. We believe our SSR offers a valuable mechanism for controlling the reasoning trajectory, mitigating biases, and aligning model behavior more closely with human expectations.

**Limitations.** Despite its advantages, SSR has several limitations. First, SSR is not intended as a universal solution for every task type. In problems where the solution path is inherently shallow (e.g., one or few steps questions) or where performance is dominated by factual retrieval rather than inference, the benefits of SSR is naturally limited. Second, the computational cost of fine-grained verification is substantially higher than that of standard iterative refinement, limiting scalability to large datasets or long reasoning chains. Finally, our evaluation focuses primarily on mathematical and logical reasoning tasks; the generalizability of SSR to open-ended domains such as commonsense or multi-modal reasoning remains to be validated.

**Future Work.** In future work, we aim to extend SSR to more diverse reasoning domains, including scientific and multimodal tasks, and explore tighter integration with training-time objectives. Another promising direction is developing more efficient confidence estimation to further reduce cost, as well as investigating human-in-the-loop settings where SSR can enhance interpretability and reliability.

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

# APPENDIX

In Appendix A, we describe the role of LLMs in our work. In Appendix B, we present the full algorithmic description of our proposed SSR. In Appendix C, we provide our implementation details of the experiments, including:

- **dataset details** (Appendix C.1),
- **baseline details** (Appendix C.2),
- **prompt templates** used in LLM reasoning (Appendix C.3),

Finally, in Appendix D, we present additional experimental results, including:

- results on **Gemini-2.5-Flash** (Appendix D.1),
- detailed results of Humanity's Last Exam (HLE) (Appendix D.2),
- results on *sequential* and *parallel* **test-time scaling** (Appendix D.3),
- analysis on the effect of the **granularity of Socratic steps** in SSR (Appendix D.4),
- results validating the **consistency of Socratic decomposition** (Appendix D.5),
- results showing the **effectiveness of SSR's confidence estimation** (Appendix D.6),
- results validating SSR-Plan's **plan-level refinement** (Appendix D.7),
- demonstrations of SSR behaviors (Appendix D.8),
- results on **SSR-as-a-Judge** (Appendix D.9),
- and **a qualitative analysis** on our SSR refinement (Appendix D.10).

## A    LLM USAGE DISCLOSURE

Large language models (LLMs) were used exclusively to help polish the writing of this paper by improving grammar, clarity, and readability. They did not contribute to research ideation, experimental design, data analysis, or the generation of scientific content. All technical contributions, claims, and conclusions are solely those of the authors.

## B    ALGORITHM

---

**Algorithm 1** Socratic Self-Refine (**SSR**)

---

**input** $\{x, x_{\text{dec}}, x_{\text{conf}}, x_{\text{ref}}\}$: prompt for original query, reasoning decomposition, confidence estimation, and refinement;
   $\pi_{\boldsymbol{\theta}}$: LLM policy;
   $(z^{(0)}, y^{(0)})$: initial CoT reasoning and answer;
   $K$: maximum refinement rounds;
   $M$: number of parallel solves per sub-question for confidence;
   $C_{\max}$: maximum value of the confidence.
1: **(Optional)** $\{q_t\}_{t \in [T]} \sim \pi_{\boldsymbol{\theta}}(\cdot \mid x, y^{(0)}, z^{(0)}, x_{\text{dec}})$. Prompt $\pi_{\boldsymbol{\theta}}$ to judge plan adequacy. If
   inadequate, refine once and update $(z^{(0)}, y^{(0)})$.                     ▷ Optional plan check (Eqn. 12).
2: **for** $k = 1, \ldots, K$ **do**
3:    $(z^{(k+1)}, y^{(k+1)}, C^{(k)}) \leftarrow \text{Self-Refine}(z^{(k)}, y^{(k)})$.                     ▷ Self-Refine Gating.
4:    **if** $C^{(k)} = C_{\max}$ **then**
5:       $S_T = \{(q_t, a_t)\}_{t \in [T]} \sim \pi_{\boldsymbol{\theta}}(\cdot \mid x, y^{(k)}, z^{(k)}, x_{\text{dec}})$.             ▷ SSR Decomposition (Eqn. 4).
6:       **for** $t = 1$ to $T$ in parallel **do**
7:          $\widehat{A}_t = \{\widehat{a}_{ti}\}_{i \in [M]}, \widehat{a}_{ti} \sim \pi_{\boldsymbol{\theta}}(\cdot \mid q_t, \{s_i\}_{i<t}, x)$.                     ▷ Reference Set Sampling.
8:          $c_t \sim \pi_{\boldsymbol{\theta}}(\cdot \mid a_t, \widehat{A}_t, x_{\text{conf}})$.                 ▷ Confidence Estimation (Eqn. 8).
9:       **end for**
10:      $t' \leftarrow \arg\min_{t \in [T]} c_t$.                     ▷ Pick weakest step
11:      $a^*_{t'} \leftarrow \text{maj\_vote}(\widehat{A}_{t'})$.                     ▷ Majority vote sub-answer
12:      $(z^{(k+1)}, y^{(k+1)}) \sim \pi_{\boldsymbol{\theta}}(\cdot \mid x, y^{(k)}, z^{(k)}, q^{(k)}_{t'}, a^{(k)}_{t'}, a^{*(k)}_{t'}, x_{\text{ref}})$.                     ▷ Round-$k$ SSR.
13:   **end if**
14: **end for**
**output** $(z^{(K)}, y^{(K)})$: refined reasoning and answer.

---

# C  IMPLEMENTATION DETAILS

Appendix C.1 introduces the basic charactaristics of the adopted datasets; Appendix C.2 introduces the implementation details of the state-of-the-art iterative refinement baselines and our SSR. Appendix C.3 lists the prompt template we use for different methods.

## C.1  DATASETS

Table 5 shows the statistics of datasets in our experiments. These datasets span two different types of reasoning tasks and different difficulty levels, from moderate to extremely challenging, covering both grade-school-level numerical reasoning and advanced symbolic mathematical tasks. This diversity in problem domains and difficulty ensures a comprehensive and representative assessment of the model's capabilities across varied reasoning scenarios.

Table 5: Dataset Statistics.

| Dataset | #Examples | Split | Task Type | Language | Level |
|---|---|---|---|---|---|
| MATH-Level-5 (Hendrycks et al., 2021) | 681 | Numerical-Answer Test Subset | Mathematical | English | Moderate |
| AIME24 (AIME-Team, 2025) | 30 | Full Set | Mathematical | English | Highly Challenging |
| AIME25 (AIME-Team, 2025) | 30 | Full Set | Mathematical | English | Highly Challenging |
| HLE (Phan et al., 2025) | 915 | Text-Only Math Subset | Mathematical | English | Extremely Challenging |
| Zebra-Puzzle (Stojanovski et al., 2025) | 100 | Randomly Synthesized | Logical | English | Moderate |
| Mini-Sudoku (Stojanovski et al., 2025) | 100 | Randomly Synthesized | Logical | English | Moderate |

## C.2  BASELINES AND OUR SSR

We compare our proposed **S**ocratic **S**elf-**R**efine (SSR) against several state-of-the-art iterative refinement reasoning frameworks. The detailed prompt templates are provided in the next section.

- **Self-Refine** (Madaan et al., 2023): We follow the prompt template defined in LLM-as-a-Judge (Zhou et al., 2025a), which produces feedback and scores for the model's own response; the feedback is then used for refinement. We perform three refinement iterations, with each iteration independent of previous ones for conciseness.
- **Debate** (Du et al., 2023): We adopt the official LLM-Debate code with two modifications: (i) using the unified CoT prompt for initial thought generation, as in this paper, and (ii) explicitly instructing each agent to **refine** its response based on the peer agent's response. We run two agents for three iterations of debate, and for fair comparison, randomly select one of the final-round answers as the output.
- **Monte Carlo Tree Self-Refine (MCTSr)** (Zhang et al., 2024): We adopt the released code for reproducibility. Since the original prompt was designed for smaller open-source LLMs (Touvron et al., 2023; Dubey et al., 2024) with format mismatches to our setting, we adapt the template while retaining the same verification prompt (as Self-Refine) and faithfully preserving the Monte Carlo Tree construction and exploration. The maximum number of iterations is set to four, following the original paper.
- **Atom-of-Thoughts (AoT)** (Teng et al., 2025): We mainly follow the released implementation. However, as the original decomposition restricts intermediate answers to purely numerical forms, which is limiting for challenging mathematical and logical reasoning, we slightly relax this constraint. For fair comparison, we set the maximum number of atoms to three, omit the final "Ensemble" step, and report only the last-iteration performance in Table 1. Results with the ensemble step are reported separately in Column "BoK-Acc" of Table 2.
- **Forest-of-Thought (FoT)** (Bi et al., 2024): As a parallel scaling variant of MCTSr (ignoring early stopping), FoT is not directly evaluated. Nevertheless, MCTSr's results in the "LR-Maj@5" column can be treated as an approximate proxy for FoT performance with tree size 5 and majority voting aggregation.
- **Linear SSR (SSR-Lin, Ours)**: Each iteration proceeds as follows: (i) decompose the given CoT into Socratic steps; (ii) re-answer each sub-question multiple times, assuming prior steps are correct; (iii) identify the step with the lowest confidence score and refine based on the majority-voted sub-answer. We set the number of iterations to three for fairness.
- **Adaptive SSR (SSR-Ada, Ours)**: At the beginning of each round, SSR-Ada first applies Self-Refine. If unreliable steps are identified with non-perfect scores, refinement proceeds via this efficient route. Otherwise (if Self-Refine fails or is overconfident), the method falls back to the full Socratic refinement.

- **SSR with Plan Refinement (SSR-Plan, Ours)**: Extends `SSR`-Ada by adding a preliminary plan refinement stage before iterative refinement.

**Shared LLM Configuration.** For `GPT-4.1-nano`, we set the maximum token length to 16,384 and temperature to 0.6. For `GPT-5-mini`, we set the maximum completion length to 16,384 and temperature to 1.0. For `Gemini-2.5-Flash`, we set the maximum completion length to 32,768 and temperature to 0.6.

## C.3 PROMPT TEMPLATES

This subsection presents the prompt templates used for the baselines and our `SSR`. The templates are identical for both mathematical and logical reasoning, except for a role specification: "you are a precise math problem solver" versus "you are a precise logical reasoning problem solver."

- **CoT:** uses *Chain-of-Thought*;
- **Self-Refine:** uses *Verification* and *Refine (Normal)*;
- **MCTSr:** uses *Verification* and *Refine (Normal)*;
- **AoT:** uses *Decompose (AoT)*, *Contract (AoT)*, and *Ensemble*;
- **SSR:** uses *Decompose (SSR, Ours)*, *Solve Sub-Question (SSR, Ours)*, *Confidence Estimate (SSR, Ours)*, *Reflection (SSR, Ours)*, and *Refine (SSR, Ours)*.

---

**Chain-of-Thought**

You are a precise math problem solver. Solve the given math problem step by step:

QUESTION: {question}

Please extend your chain of thought as much as possible; the longer the chain of thought, the better.

You can freely reason in your response, but please enclose the final answer within <answer></answer> tags (pure number without units and explanations).

---

**Verification**

Please act as an impartial judge and evaluate the correctness of the response provided by an AI assistant to the user prompt displayed below. You will be given the assistant's response.

When evaluating the assistant's response, identify any mistakes or inaccurate information. Be as objective as possible. Avoid any biases, such as order of responses, length, or stylistic elements like formatting.

Before providing an your final verdict, think through the judging process and output your thoughts as an explanation.

After providing your explanation, you must output a score of scale 0 to 5, where 0 represents you are completely certain that the response is incorrect and 5 represents you are completelycertain that the response is correct. Please enclose your score in <answer> and </answer> tags.
<|User Prompt|>
{question}
<|The Start of Assistant's Answer|>
{response}
<|The End of Assistant's Answer|>

**Refine (Normal)**

You are a precise math problem solver. Refine the provided solution to the given math problem, step-by-step, by meticulously addressing the judge's feedback (whose score is enclosed within <answer></answer> tags).

QUESTION: {question}
ORIGINAL SOLUTION: {original_cot_response}
JUDGE RESPONSE: {judge_response}

Your task is to re-evaluate the original reasoning, identify where it went wrong based on the judge's comments, which should be enclosed within <evaluation></evaluation> tags; after that, construct a new, corrected chain of thought. Explain each step thoroughly. The more detailed and explicit your reasoning, the better.

You can freely reason in your response, but please enclose the final, numerical answer within <answer></answer> tags (pure number only, without units or explanations).

**Decompose (AoT)**

You are tasked with breaking down a math problem's reasoning process into a series of sub-questions.

Original Question: {question}
Complete Reasoning Process: {trajectory}

Instructions:
- Break down the reasoning process into a series of sub-questions.
- Each sub-question should:
  - Be written in a clear, interrogative form.
  - Be precise, unambiguous, and directly answerable from the provided reasoning or prior sub-question answers.
  - Have a clear, **exact expression** as its answer (e.g., use fractions like '1/3', symbolic representations like 'pi', or precise numerical values such as '1.0'). **Crucially, avoid approximations or rounding** unless the original question explicitly requires it.
  - List the 0-based indexes of other sub-questions it depends on. This list can be empty if no prior sub-question answers are needed.
- Dependencies are defined as information necessary to answer the current sub-question that:
  - Does NOT come directly from the original question.
  - MUST come from the answers of previous sub-questions.
- **Stop generating sub-questions once the final answer to the Original Question has been fully derived from the reasoning process.** Do not include any subsequent or irrelevant steps that do not directly contribute to reaching the final answer.

Format your response as the following JSON object:

```
{{
    "sub-questions": [
        {{
            "description": "<clear, precise interrogative question>",
            "answer": <exact expression of the answer>,
            "depend": [<indices of prerequisite sub-questions>]
        }},
        ...
    ],
    "answer": {answer}
}}
```

**Contract (AoT)**

You are a math problem solver specializing in optimizing step-by-step reasoning processes. Your task is to optimize the existing reasoning trajectory into a more efficient, single self-contained question.

For the original question: {question}

Here are step-by-step reasoning process:
{response}

{sub_questions}
Here are explanations of key concepts:

- self-contained: The optimized question must be solvable independently, without relying on any external information
- efficient: The optimized question must be simpler than the original, requiring fewer reasoning steps (these steps are reduced because some solved independent sub-problems become known conditions in the optimized question or are excluded as incorrect explorations)

You can freely reason in your response, but please enclose the your optimized question within <question></question> tags.

**Decompose (SSR, Ours)**

You are tasked with breaking down a math problem's reasoning process into a series of **atomic** sub-questions.

Original Question: {question}
Complete Reasoning Process: {trajectory}

Instructions:
- Break down the reasoning process into a series of sub-questions.
- Each sub-question should:
  - Be written in a clear, interrogative form.
  - Be precise, unambiguous, and directly answerable from the provided reasoning or prior sub-question answers.
  - Have a clear, **exact expression** as its answer (e.g., use fractions like '1/3', symbolic representations like 'pi', or precise numerical values such as '1.0'). **Crucially, avoid approximations or rounding** unless the original question explicitly requires it.
  - List the 0-based indexes of other sub-questions it depends on. This list can be empty if no prior sub-question answers are needed.
- **Stop generating sub-questions once the final answer to the Original Question has been fully derived from the reasoning process.** Do not include any subsequent or irrelevant steps that do not directly contribute to reaching the final answer.
- The sub-question, sub-answer pairs should perfectly represent the reasoning process of the solution.
Format your response as the following JSON object:

```
{{
    "sub-questions": [
        {{
            "description": "<clear, precise interrogative question>",
            "answer": <exact expression of the answer>,
        }},
        ...
    ],
    "answer": {answer}
```

```
}}
```

**Solve Sub-Question (SSR, Ours)**

You are a precise math problem solver. Given the original question and the series of sub-questions and their answers which perfectly represent the reasoning process of the solution, think step by step and answer the next sub-question. Do not extend the reasoning process beyond this sub-question and enclose the answer within <answer></answer> tags.

Original question:
{question}

The series of sub-questions and their answers:
{socratic_reasoning_trajectory}

The next sub-question to be answered:
{next_sub_question}

**Confidence Estimate (SSR, Ours)**

You are a math expert. Given the a math expression as the prediction and a list of reference answers, determine the confidence of the prediction.

The prediction is:
{prediction}

The reference answers are:
{answers}

Please answer with a number of scale 0 to 5 that represents the confidence of the prediction. 0 means the prediction does not match any of the reference answers. 5 means the prediction matches the reference answers perfectly. If you cannot determine the confidence, please answer with -1. Enclose the answer within <answer></answer> tags.

**Reflection (SSR, Ours)**

Wait, in the sub-step of "{wrong_question}", the answer is "{wrong_answer}", but after careful re-evaluating the process, I think that the actual answer to this sub-question should be "{revised_answer}".

**Refine (SSR, Ours)**

{cot_instruction}

{cot_reasoning_trace}

{reflection}

Let's re-evaluate the reasoning process based on your reflection. Enclose it within <evaluation></evaluation> tags. After that, let's reasoning step by step again to solve the original question. This time, you should address the specific issue identified in your own re-evaluation. Finally, enclose the final answer within <answer></answer> tags."

---

**Ensemble**

You are a precise math problem solver. Compare then synthesize the best answer from multiple solutions to solve the following question.

QUESTION: {question}

SOLUTIONS:
{solutions}

Please extend your chain of thought as much as possible; the longer the chain of thought, the better.

You can freely reason in your response, but please enclose the final answer within <answer></answer> tags (pure number without units and explanations).

---

**LLM-as-a-Judge for Humanity's Last Exam (HLE) Evaluation**

Judge whether the following [candidate_answer] to [question] is correct or not based on the precise and unambiguous [correct_answer] below.

[question]: {question}
[correct_answer]: {correct_answer}
[candidate_answer]: {candidate_answer}

Your judgement must be in the format and criteria specified below:

reasoning: Explain why the [candidate_answer] is correct or incorrect based on [correct_answer], focusing only on if there are meaningful differences between [correct_answer] and the [candidate_answer]. Do not comment on any background to the problem, do not attempt to solve the problem, do not argue for any answer different than [correct_answer], focus only on whether the answers match.

correct: Answer '1' if [candidate_answer] matches the [correct_answer] given above, or is within a small margin of error for numerical problems. Answer '0' otherwise, i.e. if there if there is any inconsistency, ambiguity, non-equivalency, or if the extracted answer is incorrect.

Please enclose your reasoning within <reasoning></reasoning> tags, and your correct answer within <correct></correct> tags.

---

# D    ADDITIONAL EXPERIMENTAL RESULTS

Appendix D.1 reports additional results on the strong `Gemini-2.5-Flash` model. Appendix D.2 provides detailed experiments on Humanity's Last Exam (HLE) (Phan et al., 2025). Appendix D.3 includes further results on both *sequential* and *parallel* test-time scaling. Appendix D.4 analyzes how the granularity of Socratic steps influences the performance of `SSR`. Appendix D.5 validates the consistency of Socratic decomposition across model, prompt, and dataset variations. Appendix D.6 demonstrates the effectiveness of `SSR`'s confidence estimation. Appendix D.7 provides additional validation of `SSR`-Plan's plan-level refinement. Appendix D.8 presents demonstrations that further characterize the behavior of `SSR`. Appendix D.9 evaluates `SSR` as an LLM judge to shed light on its underlying mechanism. Finally, Appendix D.10 includes qualitative examples illustrating `SSR`'s refinement behavior in practice.

Table 6: **Performance of Iterative Refinement-Based Reasoning Methods. LR-Acc:** Last-round refinement's accuracy, yielded by 10 repeated experiments; **Pass@K:** Pass-at-K refinements' accuracy (at lease one of K iterations gets the answer correct). **LR-Maj@5:** Last-round refinement's accuracy of majority voting with 5 samples in parallel, yielded by 50 repeated experiments. **Boldface** and underlining denote the best and the second-best performance, respectively.

| Method | AIME24 | | | AIME25 | | | Zebra-Puzzle | | |
|---|---|---|---|---|---|---|---|---|---|
| | LR-Acc | Pass@K | LR-Maj@5 | LR-Acc | Pass@K | LR-Maj@5 | LR-Acc | Pass@K | LR-Maj@5 |
| Gemini-2.5-Flash | | | | | | | | | |
| CoT | $81.85_{\pm 2.77}$ | - | $85.60_{\pm 1.55}$ | $68.00_{\pm 4.52}$ | - | $72.47_{\pm 3.99}$ | $67.44_{\pm 1.89}$ | - | $76.12_{\pm 1.92}$ |
| Self-Refine | $82.96_{\pm 3.67}$ | $87.41_{\pm 3.05}$ | $88.87_{\pm 2.46}$ | $76.33_{\pm 7.06}$ | $81.00_{\pm 4.23}$ | $84.60_{\pm 2.48}$ | $75.25_{\pm 2.95}$ | $77.00_{\pm 3.32}$ | $88.98_{\pm 1.49}$ |
| MCTSr | $83.00_{\pm 4.07}$ | - | $86.67_{\pm 2.31}$ | $70.95_{\pm 7.50}$ | - | $77.73_{\pm 2.78}$ | $75.60_{\pm 2.94}$ | - | $85.68_{\pm 1.91}$ |
| AoT | $81.67_{\pm 1.67}$ | $85.33_{\pm 2.21}$ | $86.13_{\pm 2.86}$ | $70.74_{\pm 5.62}$ | $75.19_{\pm 6.50}$ | $78.40_{\pm 2.60}$ | $54.71_{\pm 3.49}$ | $86.14_{\pm 1.88}$ | $65.74_{\pm 2.39}$ |
| SSR-Lin (Ours) | $\mathbf{86.30_{\pm 3.99}}$ | $\mathbf{90.37_{\pm 4.29}}$ | $\mathbf{90.93_{\pm 2.98}}$ | $\mathbf{79.26_{\pm 4.66}}$ | $\underline{83.33_{\pm 4.16}}$ | $\mathbf{88.47_{\pm 3.14}}$ | $\mathbf{87.62_{\pm 2.18}}$ | $\underline{89.75_{\pm 2.54}}$ | $\mathbf{92.30_{\pm 1.36}}$ |
| SSR-Ada (Ours) | $82.50_{\pm 4.00}$ | $87.50_{\pm 3.23}$ | $88.33_{\pm 1.67}$ | $76.30_{\pm 6.37}$ | $\mathbf{84.44_{\pm 4.71}}$ | $87.27_{\pm 2.72}$ | $87.14_{\pm 1.96}$ | $89.00_{\pm 1.69}$ | $91.86_{\pm 1.30}$ |
| SSR-Plan (Ours) | $84.17_{\pm 4.00}$ | $\underline{89.17_{\pm 3.63}}$ | $89.67_{\pm 1.00}$ | $78.00_{\pm 6.00}$ | $84.00_{\pm 4.42}$ | $86.73_{\pm 3.16}$ | $86.50_{\pm 2.69}$ | $89.00_{\pm 2.50}$ | $\underline{92.06_{\pm 1.39}}$ |

Table 7: Accuracies (%) of iterative refinement-based reasoning methods on the 478-question challenging math subset (w/ numerical ground-truth answer) of Humanity's Last Exam (HLE) (Phan et al., 2025), with `GPT-5-mini` and `GPT-5` (medium reasoning, medium verbosity).

| Model | CoT | Self-Refine | SSR-Plan (Ours) |
|---|---|---|---|
| GPT-5-mini | 17.78 | 23.85 (+6.07) | **26.57 (+8.89)** |
| GPT-5 | 30.33 | 33.89 (+3.56) | **35.56 (+5.23)** |

## D.1 RESULTS OF GEMINI-2.5-FLASH

**We further report results of applying SSR to a stronger model, Gemini-2.5-Flash, from a different model family** (Comanici et al., 2025). Owing to its exceptionally strong mathematical and logical reasoning ability, two benchmarks used in the main body (MATH-Level-5 and Mini-Sudoku) are no longer suitable for differentiating framework performance, as naive CoT already solves nearly all questions correctly. Therefore, we report results only on the remaining three datasets, following the same evaluation protocols described in Sec. 4.

When applied to the stronger Gemini-2.5-Flash model, our SSR variants continue to demonstrate consistent improvements over baseline iterative refinement methods. On AIME24 and AIME25, SSR-Lin achieves the highest LR-Acc and LR-Maj@5, while SSR-Ada and SSR-Plan deliver highly competitive results, particularly in terms of Pass@K, reflecting their ability to exploit refinement opportunities even when the base model is already very strong. The gains are especially notable on AIME25, where SSR-Ada substantially outperforms all baselines in both LR-Acc and Pass@K, indicating the effectiveness of adaptively switching between efficient self-refinement and more costly Socratic refinement. On Zebra-Puzzle, all three variants of SSR surpass or match the best-performing baselines, with SSR-Lin again delivering the strongest overall results. These findings confirm that even for a state-of-the-art reasoning model like Gemini-2.5-Flash, our refinement strategies provide additional benefits, reinforcing their generality and scalability across model families and task types.

## D.2 DETAILED RESULTS OF HUMANITY'S LAST EXAM (HLE)

**Table 7 and Table 8 present a detailed breakdown of SSR performance on the numerical and non-numerical subsets of Humanity's Last Exam (HLE) (Phan et al., 2025).** On the numerical subset, SSR achieves substantial gains over both CoT and Self-Refine, improving accuracy by up to 8.89% with GPT-5-mini and 5.23% with the full GPT-5. In contrast, on the non-numerical subset, improvements are smaller or even negative, particularly for GPT-5, where Self-Refine and SSR both slightly underperform CoT. This disparity suggests that non-numerical problems, often involving abstract or conceptual reasoning, may benefit less from explicit step-level self-verification and refinement, as it can introduce semantic drift or over-justification. Overall, these results demonstrate that SSR is especially effective for precise, calculation-heavy reasoning but may require further adaptation for more open-ended or qualitative tasks.

Table 8: Accuracies (%) of iterative refinement-based reasoning methods on the 437-question challenging math subset (w/ non-numerical ground-truth answer) of Humanity's Last Exam (HLE) (Phan et al., 2025), with `GPT-5-mini` and `GPT-5` (medium reasoning, medium verbosity).

| Model | CoT | Self-Refine | SSR-Plan (Ours) |
|---|---|---|---|
| `GPT-5-mini` | 14.42 | 12.81 (-1.61) | **16.02** (+1.60) |
| `GPT-5` | **25.40** | 18.08 (-7.32) | 23.11 (-2.29) |

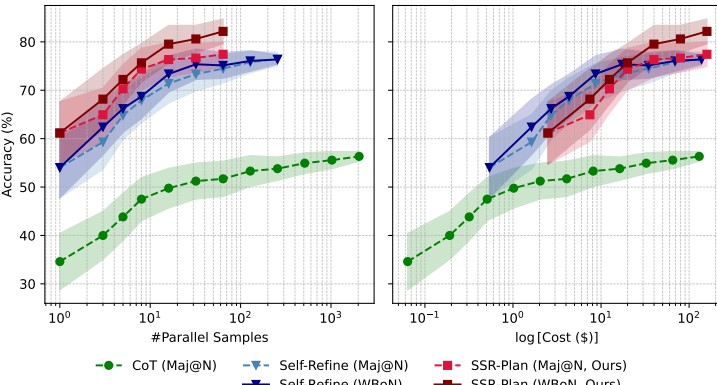

Figure 4: **Performance of Parallel Test-Time Scaling,** evaluated on AIME25 with `GPT-5-mini` low-reasoning low-verbosity mode. Iterative refinement (both Self-Refine (Madaan et al., 2023) and our SSR) holds non-trivial advantage against CoT (Wei et al., 2022b) in terms of absolute performance and budget control. Our SSR outperforms the baselines under the same budget, with SSR's confidence estimation playing a crucial role.

## D.3 TEST-TIME SCALING AT LARGER SCALE

Applying iterative refinement, even for a single round, inevitably increases computation and latency at test time. Thus, comparisons restricted to a fixed number of iterations, as in Sec. 4.5, may be unfair or incomplete. To more fairly assess efficiency, we examine the test-time scaling behavior of our SSR relative to baselines under comparable computational cost. **The results are presented in Fig. 4 (parallel scaling) and Fig. 5 (sequential scaling).**

In the parallel scaling setting (Fig. 4), both Self-Refine and our SSR substantially outperform vanilla CoT across all compute budgets, confirming that iterative refinement provides clear gains when additional samples are available. Importantly, our SSR consistently yields higher accuracy than Self-Refine under the same budget, demonstrating that confidence-aware step selection and plan refinement lead to more efficient use of compute. In the sequential scaling setting (Fig. 5), a similar trend emerges: while performance plateaus quickly for Self-Refine, SSR continues to improve steadily with additional iterations, particularly in the early- to mid-cost regime. This suggests that SSR better leverages iterative opportunities, correcting errors that Self-Refine either overlooks or misjudges. Taken together, these results demonstrate that SSR not only provides stronger single-iteration performance but also scales more effectively under increased compute, striking a favorable balance between accuracy and cost.

## D.4 ANALYSIS: GRANULARITY OF SOCRATIC STEPS IN SSR

In this section, we investigate the effect of explicitly controlling decomposition granularity by varying the maximum number of Socratic steps. This is implemented by modifying the decomposition prompt: instead of instructing *"Break down the reasoning process into a series of sub-questions,"* we use *"Identify the most important milestones of the reasoning process and break it down into a series of sub-questions, with the number of sub-questions less than or equal to {max_steps}."* We then report iteration accuracy as a function of the actual number of Socratic steps produced by decomposition. To isolate the effect of SSR, our main analysis is conducted with the Linear variant (SSR-Lin), without adaptive gating or plan refinement, while also including SSR-Plan for reference (retaining steps that undergo Socratic decomposition). Note that setting the maximum number of steps to 1 reduces SSR

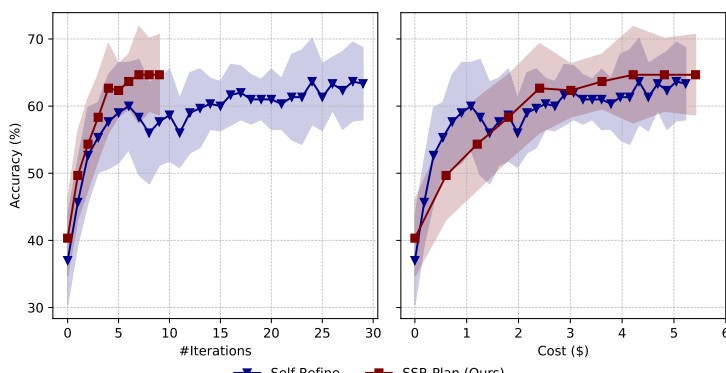

Figure 5: **Performance of Iterative Test-Time Scaling,** evaluated on AIME25 with `GPT-5-mini` low-reasoning low-verbosity mode.

to a final-answer evaluation via majority voting. Accordingly, we focus on the range of 3-10 steps in our experiments.

**The results are reported in Fig. 6.** For `SSR`-Lin, performance is relatively low and fluctuates with the number of Socratic steps, though a slight upward trend can be observed at higher step counts (e.g., 9-10 on AIME24). This suggests that finer-grained decomposition can sometimes help, but the effect is weak and unstable when refinement is applied without planning. In contrast, the Plan-refinement variant (`SSR`-Plan) consistently achieves higher accuracy across all settings (possibly due to the

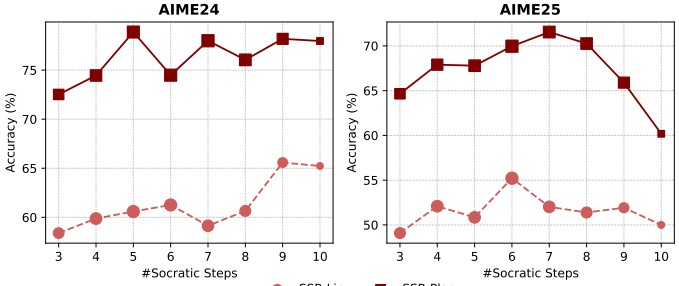

Figure 6: **Performance of our `SSR`, with explicit control of granularity,** evaluated on AIME24 and AIME25 (AIME-Team, 2025) with GPT-5-mini low-reasoning low-verbosity mode. The marker size of each data point is proportional to the support size.

gating mechanism of Self-Refine) and remains stable under varying levels of granularity. On AIME24, performance remains strong regardless of step count, while on AIME25, accuracy peaks around 6-7 steps and only drops when the decomposition becomes overly fine (10 steps). These results highlight that high-level plan refinement not only boosts overall accuracy but also makes `SSR` less sensitive to the specific choice of granularity, ensuring more reliable gains.

### D.5  ANALYSIS: `SSR` DECOMPOSITION CONSISTENCY

We conducted additional experiments to examine the consistency of Socratic-step decomposition across *(i) different runs, (ii) different base LLMs,* and *(iii) different versions of the decomposition prompt.* Our experimental setup is summarized as follows:

- **CoT responses to be decomposed:** We base our analysis on the chain-of-thought outputs generated by `GPT-4.1-nano`, as `GPT-5`'s responses are overly concise and do not reveal full reasoning traces due to OpenAI's policy restrictions.
- **Base Models:** We use `GPT-5` in low-reasoning mode as our main base model for its strong balance between reasoning and instruction following. To assess cross-model consistency, we also test `Gemini-2.5-Flash` from a different model family.
- **Datasets:** We evaluate on two datasets from our main experiments: **AIME25** (mathematical reasoning) and **Zebra Puzzle** (logical reasoning).
- **Evaluation Metrics:**
  - *Comparing Answer Sets as Proxy.* Directly comparing two sets of Socratic steps is difficult because the sub-questions and intermediate answers may be phrased differently and cannot be reliably parsed for semantic equivalence. Instead, we compare the answer sets produced by two decomposition processes.

Table 9: **SSR's Decomposition Consistency on Reasoning Data,** where V0/V1/V2 denotes different decomposition prompts. **OC:** Overlap Coefficient; **Jaccard:** Jaccard Similarity.

| Model-1 | Model-2 | AIME25 | | Zebra-Puzzle | |
|---------|---------|--------|--------|--------------|--------|
| | | OC | Jaccard | OC | Jaccard |
| GPT-5 (V0) | GPT-5 (V0) | $0.8358_{\pm 0.2067}$ | $0.6716_{\pm 0.2214}$ | $0.7338_{\pm 0.2041}$ | $0.5006_{\pm 0.2465}$ |
| GPT-5 (V0) | Gemini-2.5-Flash (V0) | $0.8122_{\pm 0.1739}$ | $0.4406_{\pm 0.2061}$ | $0.5832_{\pm 0.2309}$ | $0.2744_{\pm 0.1635}$ |
| GPT-5 (V0) | GPT-5 (V1) | $0.8545_{\pm 0.1415}$ | $0.6422_{\pm 0.1880}$ | $0.7199_{\pm 0.2497}$ | $0.4750_{\pm 0.2750}$ |
| GPT-5 (V0) | GPT-5 (V2) | $0.8282_{\pm 0.1540}$ | $0.6221_{\pm 0.2093}$ | $0.7745_{\pm 0.1861}$ | $0.5419_{\pm 0.2685}$ |
| GPT-5 (V1) | GPT-5 (V2) | $0.8717_{\pm 0.1912}$ | $0.6784_{\pm 0.2275}$ | $0.7472_{\pm 0.2107}$ | $0.5022_{\pm 0.2615}$ |

Table 10: **Match Rate (%) between SSR' max-confidence answer and Universal Self-Consistency (USC)-selected answer (Chen et al., 2023) on the reasoning datasets.**

| Model | AIME25 | Zebra-Puzzle |
|-------|--------|--------------|
| GPT-4.1-nano | 72.29 | 92.04 |
| GPT-5-mini | 93.04 | 98.05 |

- *Taking the Granularity into Account.* As noted in Footnote 1, the ground-truth decomposition may not be unique, and two decomposition results of slight difference in granularity with one covered by the other should consider two consistent-enough decomposition results. Hence we resort to Overlap Coefficient for evaluating the similarity between two decompositions. Specifically, for two sets $A$ and $B$, we report the following two metrics:
  * **Overlap Coefficient (OC)**:

$$OC(A, B) = \frac{|I(A,B)|}{\min\{|A|,|B|\}},\tag{13}$$

  which captures the proportion of shared steps. This is the most relevant measure for decomposition consistency, since randomness may produce different granularities, making direct set equivalence not fully suitable.
  * **Jaccard Similarity**:

$$\text{Jaccard}(A, B) = \frac{|I(A,B)|}{|U(A,B)|},\tag{14}$$

  reported for completeness as a secondary reference metric.

**The results are reported in Table 9.** As shown in the table, across tasks, models, and prompt variants, Socratic-step decomposition shows strong and reliable consistency. On AIME25, Overlap Coefficients remain high (0.83-0.87 within-model; 0.81 cross-model), and even on the more ambiguous Zebra Puzzles, consistency stays solid (0.58-0.77). Prompt variants exhibit similar agreement. These results indicate that the extracted steps are stable, largely model- and prompt-invariant, and capture a coherent underlying reasoning structure, supporting the validity of our decomposition approach.

### D.6   ANALYSIS: CONSISTENCY OF SSR'S CONFIDENCE ESTIMATION

While it is true that LLM-as-a-Judge can be unreliable for evaluating complex multi-step reasoning, in our SSR, estimating the confidence of a single answer given a small reference set is a much simpler and more stable task. As described in Section 3.2, our confidence estimation $c \sim \pi_{\theta}(a, \widehat{A}, x_{\text{conf}})$ is a direct and principled extension of Universal Self-Consistency (USC) (Chen et al., 2023), widely validated method where the LLM is given a set of candidate answers and asked to **select the most self-consistent one:**

$$a^* \sim \pi_{\theta}(\widehat{A}, x_{\text{USC}}).\tag{15}$$

To further demonstrate the reliability of SSR's confidence scores, we compare them against USC. Given a sampled reference set $\widehat{A} = \{\widehat{a}_1, \widehat{a}_2, \cdots, \widehat{a}_M\}$, where $M = 5$ is set in our case, we apply USC as in Eqn. 15 to select the most consistent answer $\widehat{a}^*$ and treat this selection as a proxy ground truth. We then evaluate how often **the answer with the highest SSR's confidence score,** $\arg\max_i \{c_i \sim \pi_{\theta}(\widehat{a}_i, \widehat{A}, x_{\text{conf}})\}_{i \in [M]}$, matches $\widehat{a}^*$.

**The agreement rates on both the mathematical (AIME25, 2,400 examples evaluated) and logical (Zebra-Puzzle, 11,516 examples evaluated) datasets are reported in Table 10.** These numbers

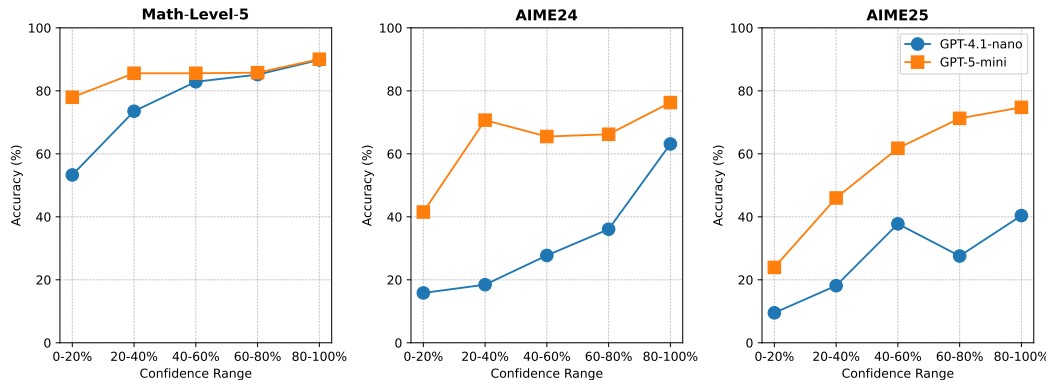

Figure 7: **Correlation between the SSR-Lin's Last-Round Aggregated Confidence Scores and the Final Answer Accuracy,** evaluated on three math reasoning datasets.

Table 11: **Last-Round Performance of SSR-Plan's Plan-Refinement Variants. LR-Acc:** Last-round refinement's accuracy, yielded by 10 repeated experiments; **LR-Maj@5:** Last-round refinement's accuracy of majority voting with 5 samples in parallel, yielded by 50 repeated experiments. **Boldface** denotes the best performance.

| Method | AIME24 | | AIME25 | | Zebra-Puzzle | | Mini-Sudoku | |
|---|---|---|---|---|---|---|---|---|
| | LR-Acc | LR-Maj@5 | LR-Acc | LR-Maj@5 | LR-Acc | LR-Maj@5 | LR-Acc | LR-Maj@5 |
| GPT-4.1-nano | | | | | | | | |
| 3×(Plan-Refine → SSR-Refine) | 27.33±4.42 | 32.53±3.50 | 23.33±3.65 | 26.47±2.62 | 55.50±2.42 | 55.46±2.00 | 42.10±4.16 | 58.44±2.98 |
| 3×Plan-Refine → 3×SSR-Refine | **27.67±4.48** | **36.00±2.83** | **24.67±5.81** | **31.00±3.48** | 55.30±2.53 | 56.14±2.18 | 47.00±4.98 | 62.30±3.41 |
| 1×Plan-Refine → 3×SSR-Refine (Ours) | 27.33±5.73 | 35.80±3.39 | 22.33±3.67 | 27.53±4.46 | **56.90±3.11** | **57.30±2.39** | **47.70±4.22** | **66.46±4.61** |
| GPT-5-mini | | | | | | | | |
| 3×(Plan-Refine → SSR-Refine) | 52.33±4.96 | 62.33±3.84 | 41.00±5.59 | 51.73±5.34 | 87.90±1.37 | 91.72±1.39 | 77.60±1.50 | 92.64±1.40 |
| 3×Plan-Refine → 3×SSR-Refine | 62.33±4.96 | 71.20±3.38 | 55.67±7.75 | 66.20±4.11 | 86.50±1.43 | 91.86±1.59 | 89.80±3.25 | 99.44±0.73 |
| 1×Plan-Refine → 3×SSR-Refine (Ours) | **69.67±4.82** | **79.00±3.48** | **62.00±6.18** | **71.53±5.26** | **88.00±1.55** | **93.20±1.08** | **94.80±2.48** | **100.00±0.00** |

demonstrate that SSR 's confidence-based selection aligns extremely well with the well-established USC method, providing a solid foundation for its adoption in our SSR.

To further show the confidence scores are well-grounded, Fig. 7 provide an additional correlation analysis between the SSR-Lin's last-round aggregated step-level confidence scores and the correctness of the final answers. The reason for choosing SSR-Lin for analysis instead of SSR-Ada or SSR-Plan is that adaptive gating mechanism adopted by the latter two variants might cause the last-round confidence scores with mixed sources (from SSR or naive Self-Refine). As shown in the figure, across multiple datasets and multiple models, the average confidence scores produced by SSR-Lin is highly correlated with the final answers' correctness.

## D.7 ABLATION STUDY: SSR'S CURRENT PLAN-REFINEMENT DESIGN

To better understand the role and effect of Plan-Refinement in SSR-Plan, we consider two variants that attempt to integrate Plan-Refinement into SSR:

- **Interleaving the plan-level and step-level refinement,** 3×(Plan-Refine → SSR-Refine);
- **Multiple rounds of plan-level refinement** before multi-round of step-level refinement, 3×Plan-Refine → 3×SSR-Refine.

**The results are reported in Table 11.** We observe an unexpected performance drop when plan-refinement is applied at every refinement round. This becomes intuitive when viewed alongside Fig. 5: self-refinement improves solutions accumulatively, correcting errors round by round. However, excessively invoking plan-refinement, i.e., repeatedly altering the high-level reasoning blueprint—prevents the step-level, fine-grained refinement from progressing beyond the shallow first round. As a result, the model loses the benefits of deeper iterative correction, ultimately harming performance. Performing full rounds of plan-refinement before SSR 's step-level refinements does not appear to provide any benefit. In contrast, our current SSR implementation, which applies plan-refinement only once at

Table 12: **Last-Round Refine Rate (%) of Iterative-Refinement Methods.**

| Initial CoT's Correctness | Math-Level-5 | | AIME24 | | AIME25 | |
|---|---|---|---|---|---|---|
| | *Self-Refine* | *SSR-Plan* | *Self-Refine* | *SSR-Plan* | *Self-Refine* | *SSR-Plan* |
| | | | GPT-4.1-nano | | | |
| Correct (↓) | 16.45 | **9.91** | **12.35** | 22.22 | **13.04** | 37.68 |
| Incorrect (↑) | 60.49 | **75.34** | 46.58 | **79.00** | 45.02 | **81.82** |
| | | | GPT-5-mini | | | |
| Correct (↓) | 4.82 | **2.61** | **1.97** | 3.29 | **1.80** | 4.50 |
| Incorrect (↑) | 69.51 | **84.41** | 79.73 | **92.57** | 73.02 | **88.89** |

Table 13: **Plan-Refinement Rate (PRR, %) and Use-SSR Rate (USR, %) of SSR-Plan.**

| Model | Math-Level-5 | | AIME24 | | AIME25 | | Zebra-Puzzle | | Mini-Sudoku | |
|---|---|---|---|---|---|---|---|---|---|---|
| | *PRR* | *USR* | *PRR* | *USR* | *PRR* | *USR* | *PRR* | *USR* | *PRR* | *USR* |
| GPT-4.1-nano | 13.00 | 78.39 | 35.33 | 57.80 | 31.67 | 49.60 | 56.80 | 44.98 | 81.30 | 76.62 |
| GPT-5-mini | 7.81 | 50.21 | 24.67 | 56.53 | 22.00 | 58.27 | 7.30 | 66.04 | 27.60 | 56.76 |

the beginning, achieves the strongest empirical performance. This result aligns well with our initial assumption as in defined in Eqn. 12.

## D.8 ADDITIONAL DEMONSTRATIONS OF SSR BEHAVIORS

**Last-Round Refine Rate (%) of SSR-Plan.** We study the correlation between the refine rate (the rate at which a model eventually changes the original final answer) and the initial correctness of a response, of our SSR-Plan. The results are based on the existing logs of our main experiments, as reported in Table 12. The results demonstrate that our approach tends to make refinements that actually change the final answers when they are incorrect significantly more often than the cases when the initial answers are correct, for which it is able to preserve the original answer. This robustness explains the origin of our SSR's improvement.

**Plan-Refinement Rate (%) and Use-SSR Rate (%) of SSR-Plan.** An interesting question to ask about our SSR-Plan involve *"how often does SSR-Plan refines the high-level plan of a response?"* and *"how often does SSR-Plan invokes the complicated SSR-Refine for step-level errors?"* Table 13 present the distribution of our gating mechanism (Use-SSR Rate, **USR**), illustrating how the model dynamically switches between the low-cost Self-Refine and the Socratic Self-Refine stages; and the distribution of refinement of high-level plans (Plan-Refinement Rate, **PRR**), illustrating the relationship between errors in the high-level plans versus those in the execution steps.

## D.9 ANALYSIS: SSR-AS-A-JUDGE

To better understand the strengths of SSR, we further assess its self-evaluation quality and compare it with the LLM-as-a-Judge framework (Gu et al., 2024). We evaluate the self-evaluation ability on the four datasets we use in the main body, and we further include the results on ProcessBench (Zhang et al., 2025b). For self-evaluation, due to the smaller dataset sizes of AIME24 and AIME25, we sample 100 parallel reasoning traces per question, yielding datasets of 3,000 examples each. For logical reasoning, we sample 10 traces per question, resulting in datasets of 1,000 examples each. In the LLM-as-a-Judge setting, the model is prompted to provide both feedback and a confidence score on a 0–5 scale. For SSR, we perform a single iteration of Socratic step decomposition and confidence estimation of each step. All experiments run with GPT-5-mini low-reasoning low-verbosity mode. Since SSR produces step-level confidence scores $C_T = \{c_t\}_{t \in [T]}$ for the Socratic steps $S_T = \{s_t\}_{t \in [T]}$, these must be aggregated into a single score to represent overall response quality. We show the result of (i) **Min** ($\min\{c_t\}_{t \in [T]}$), the weakest step confidence; (ii) **Mean-Log** ($\frac{1}{T} \sum_{t=1}^{T} \log c_t$), a length-normalized version inspired by confidence and uncertainty estimation in sequence modeling (Zhang et al., 2025a); and (iii) SSR-Ada with **Mean**.

We formulate the evaluation of a judge's ability as a problem of incorrect reasoning trace detection, where incorrect responses are labeled as positives. We report three correlation-based metrics: Area Under the Receiver Operating Characteristic Curve (**AUROC**), **Precision**[*] and **Recall**[*] at the optimal classification threshold (Hanley & McNeil, 1982; Boyd et al., 2013; Farquhar et al., 2024; Ye et al.,

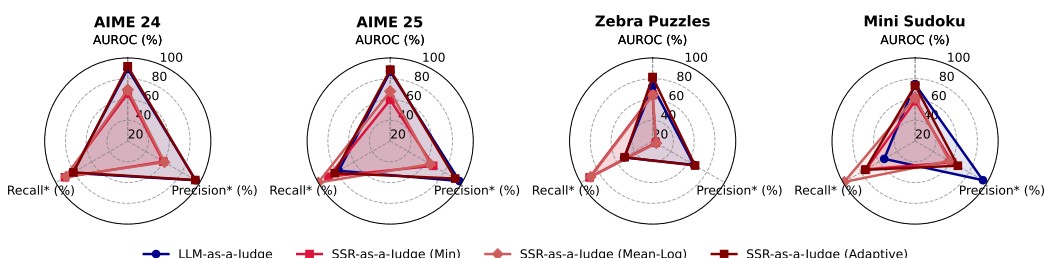

Figure 8: **Self-Evaluation Performance of SSR-as-a-Judge and LLM-as-a-Judge,** evaluated with `GPT-5-mini`.

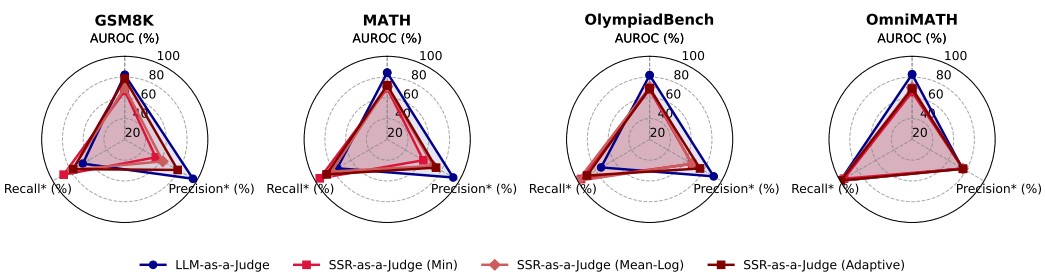

Figure 9: **Performance of SSR-as-a-Judge and LLM-as-a-Judge,** evaluated on Process-Bench (Zhang et al., 2025b), with `GPT-4.1-nano`.

2025; Zhang et al., 2025a), which together measure how well confidence scores distinguish between correct and incorrect responses.

**The results are shown in Fig. 8 and Fig. 9.** Somewhat unexpectedly, across most evaluation metrics, the judging ability of SSR does not surpass the basic LLM-as-a-Judge. This is evident in consistently lower AUROC, suggesting that the confidence scores produced by SSR contain more noise and thus yield less balanced evaluations. *Why, then, does SSR still outperform baselines as an iterative refinement framework?* As illustrated in Fig. 8, the key lies in its complementary role to Self-Refine. While SSR lags behind LLM-as-a-Judge in AUROC,, it consistently achieves much higher recall of incorrect reasoning traces, particularly on logical reasoning benchmarks such as Zebra Puzzle and Mini-Sudoku. This broader coverage allows SSR to catch errors that Self-Refine often misses, even if it introduces additional noise. The mechanism behind SSR-Ada can thus be understood as three cascading factors:

- **High precision of LLM-as-a-Judge:** when used in Self-Refine, it reliably identifies problematic reasoning traces, but often misses a large portion of incorrect ones.
- **High coverage of SSR:** it captures and provides useful signals for truly problematic steps in reasoning, though at the cost of introducing some unreliable feedback for feedback.
- **Inherent robustness of LLMs:** during refinement, LLMs can withstand noisy refinement feedback, serving as a safeguard that enables recovery and improvement despite occasional errors.

## D.10 QUALITATIVE ANALYSIS

In this section, we provide additional qualitative examples that further illustrate how SSR performs step-level verification and refinement in practice. Specifically, we include:

- an example where SSR successfully solves a challenging AIME25 problem (with details summarized as needed);
- an example highlighting a failure case of SSR-Plan's plan-level refinement (with details summarized as needed);
- a complete reasoning trace of SSR using `GPT-4.1-nano` as the backbone LLM.

**A Summarized SSR Example.**  We include an additional summarized example where SSR successfully corrects errors that naive Self-Refine fails to fix (spot mistakes).

- **Question:**

```
Let $A_1 A_2 A_3 \\ldots A_{11}$ be an $11$-sided non-convex
simple polygon with the following properties:

\\begin{itemize}
\\item For every integer $2 \\le i \\le 10$,  the area of
$\\triangle A_i A_{1} A_{i+1}$  is equal to $1$.
\\item For every integer $2 \\le i \\le 10$,
$\\cos(\\angle A_i A_{1} A_{i+1}) = \\frac{12}{13}$.
\\item The perimeter of the $11$-gon
$A_1 A_2 A_3 \\dots A_{11}$
is equal to $20$.
\\end{itemize}

Then $A_1 A_2 + A_1 A_{11} = \\frac{m \\sqrt{n} - p}{q}$ where
$m, n, p$, and $q$ are positive integers, $n$ is not divisible
by the square of any prime, and no prime divides all of $m, p$,
and $q$. Find $m + n + p + q$.
```

- **Ground-Truth Answer: 19;**
- **Original CoT Response (Summarized):**

```
The reasoning establishes that each triangle A_iA_1A_{i+1} has
area 1 and constant angle cosine 12/13, implying r_i r_{i+1}
= 26/5. This forces the radii from A_1 to alternate:
r_2=r_4=\cdots=a and r_3=r_5=\cdots=b, with ab=26/5.
Using the law of cosines, all nine interior edges A_2A_3
through A_{10}A_{11} share a common length s, which simplifies
to s = (a+b)^2 - 20. The perimeter condition then gives
a+b + 9s = 20. Letting S = a+b leads to the quadratic
9S^2 + S - 200 = 0, whose positive solution is
S = \frac{\sqrt{7201} - 1}{18}. Because 7201 is squarefree
(19 \cdot 379), the required parameters are
m=1, n=7201, p=1, q=18, giving the final value m+n+p+q = 7221.
```

- **Self-Refine Verification (Summarized):**

```
The verification confirms that every major step in the assistant's
solution is mathematically correct. The area constraint correctly
yields r_i r_{i+1}=26/5, establishing the period-2 pattern of
radii. The law of cosines is applied properly, and the symmetry
argument justifies that all nine interior edges share the same
length s. The algebraic simplification leading to s = S^2 - 20
is checked carefully and found valid. The perimeter equation
S + 9(S^2 - 20) = 20 is derived accurately, producing the
quadratic 9S^2 + S - 200 = 0. Solving it gives
S = (\sqrt{7201} - 1)/18, with all number-theoretic conditions
satisfied. Overall, both the reasoning and computations are sound,
and the final answer is correct.
```

- **Low-Confident Socratic Step:**
  - **Sub-Question:**

```
For the alternating radii a and b, what is the common interior
edge length s between A2...A11 expressed in a and b?
```

  - **Original Answer:**

```
    s = a^2 + b^2 - 2 * C * (12/13)
```

– **Reference Answers:**

```
[
    "Each such edge has length s = sqrt(a^2 + b^2 -
    2*(12/13)*C) = sqrt(a^2 + b^2 - 2*(12/13)*(26/5)) =
    sqrt(a^2 + b^2 - (624/65)).",
    "s = sqrt(a^2 + b^2 - 2*(26/5)*(12/13)) = sqrt(a^2
    + b^2 - (24*26)/(65)) = sqrt(a^2 + b^2 - 624/65)",
    "s = sqrt(a^2 + b^2 - 2*(12/13)*C) = sqrt(a^2 + b^2
    - 2*(12/13)*(26/5)) = sqrt(a^2 + b^2 - (624/65))",
    "s = sqrt(a^2 + b^2 - 2*(26/5)*(12/13)) = sqrt(a^2 +
    b^2 - (24*26)/(65)) = sqrt(a^2 + b^2 - 624/65)",
    "s = sqrt(a^2 + b^2 - 2*(12/13)*a*b)"
],
```

– **Confidence:** 0;

- **Socratic Refinement (Summarized):**

```
The refinement identifies and corrects a minor earlier mistake
regarding the interior edge length s, clarifying that the squared
form s^2 from the law of cosines should be handled consistently.
It re-derives the solution cleanly and rigorously: using the area
constraint and the angle data, it confirms the alternating radii
pattern a,b with ab=26/5. Applying the law of cosines yields
s^2 = S^2 - 20, where S = a + b. Using the perimeter condition
S + 9\sqrt{S^2 - 20} = 20, it solves for S, obtaining the positive
root S = (9\sqrt{5} - 1)/4. This matches the required form, giving
m=9, n=5, p=1, q=4, and hence the final value m+n+p+q = 19. The
revised solution is consistent, self-contained, and
mathematically correct.
```

**A Failing Case of SSR-Plan's Plan Refinement.** We analyzed the test-time scaling logs of SSR-Plan on aime25, which contain 3,840 total plan-refinement samples (128 runs × 30 questions). Among these, we identified 9 cases where the original chain-of-thought produced the correct final answer (suggesting the initial plan was sound), but the refined plan produced an incorrect final answer. This yields an approximate failure rate of $9/3840 \approx 0.234\%$, indicating that **plan-refinement errors are exceedingly rare.**

Below we present an example of such a failure, where the refined plan subtly altered the algebraic objective and caused the model to solve an incorrect polynomial (a spurious "biquadratic" in $x^2$) rather than the correct quartic:

- **Question:**

```
The parabola with equation $y=x^{2}-4$ is rotated $60^{\\circ}$
counterclockwise around the origin. The unique point in the
fourth quadrant where the original parabola and its image
intersect has $y$-coordinate $\\frac{a-\\sqrt{b}}{c}$, where
$a$, $b$, and $c$ are positive integers, and $a$ and $c$ are
relatively prime. Find $a+b+c$.
```

- **Ground-Truth Answer:** 62;

- **Original CoT Response (Summarized):**

```
The parabola y = x^2 - 4 is rotated 60^\circ counterclockwise,
and intersection points between the original and rotated curves
```

```
are found by applying the rotation matrix to (x,y) and
enforcing that the rotated point also satisfies the parabola
equation. Substituting y = x^2 − 4 into the rotated coordinates
produces a quartic equation in x. After simplification, the
quartic factors into two quadratics, and the one with real
solutions is x^2 + \sqrt{3}x − 4 = 0. Solving yields the
fourth-quadrant root x = (\sqrt{19} − \sqrt{3})/2. Substituting
back gives y = \frac{3 − \sqrt{57}}{2}, which is of the form
(a − \sqrt{b})/c  with a=3, b=57, c=2. Therefore, a+b+c = 62.
```

- **Plan Summarization:**

```
Set up the problem by representing the 60\u00b0 rotation as a
linear map (rotation matrix) and express the image (x',y') of
a general point (x,y) on the original parabola y = x^2 \u2212
4. Impose the intersection condition by requiring that the
rotated point also lie on the parabola, yielding an equation
in x alone after substituting y = x^2 \u2212 4. Convert that
condition to a polynomial equation (a quartic) in x, simplify,
and factor the quartic into two quadratic factors. Discard
nonreal roots, choose the root giving a point in the fourth
quadrant (x > 0, y < 0), compute the corresponding y = x^2
\u2212 4, express y in the form (a \u2212 \u221ab)/c with
integer parameters, and sum a + b + c.
```

- **Plan Score:** 4;

- **Refined Plan (Summarized):**

```
The strategy is to rotate a general point (x,y) from the
parabola y = x^2 − 4 by 60^\circ using the rotation matrix,
yielding coordinates (X,Y). Imposing the condition that the
rotated point also satisfies the parabola equation Y = X^2 − 4
leads, after substituting y = x^2 − 4, to an equation solely
in x. Clearing denominators and grouping terms produces a
quartic polynomial (or a pair of polynomial conditions
separated by coefficients of 1 and \sqrt{3}). The plan then
outlines factoring or reducing the quartic--checking for even
symmetry, attempting a substitution u = x^2, or matching
quadratic factors. After solving for the real roots, the
correct intersection is selected by enforcing the
fourth-quadrant condition x > 0,\, y < 0. Finally, the
resulting y = x^2 − 4 is simplified into the form
(a − \sqrt{b})/c, with a,b,c integers meeting the problem
constraints, and their sum a+b+c is computed.
```

- **New CoT Response based on the Refined Plan (Summarized):**

```
The solution rotates points on the parabola y = x^2 − 4 by
60^\circ using standard rotation formulas, then enforces that
the rotated point also satisfies the parabola equation.
Substituting y = x^2 − 4 into the rotation coordinates and
separating rational and \sqrt{3}-terms produces a quartic
equation in x, which reduces (via u = x^2) to the quadratic
u^2 − 8u + 12 = 0. This yields x^2 = 2 or 6. Checking quadrant
constraints and verifying consistency with the rotation shows
the valid fourth-quadrant intersection corresponds to x^2 = 6,
leading to the y-coordinate y = (7 − 4\sqrt{3})/4. Interpreted
in the required form (a − \sqrt{b})/c, this gives
a = 7, b = 48, c = 4, so the requested sum is a+b+c = 59.
```

**A Full-Trace `SSR` Example.** As shown in Fig. 10-16, the qualitative example illustrates how `SSR` intervenes within the reasoning process rather than only at the final output. By decomposing reasoning into Socratic steps, estimating step-level confidence, and selectively refining weak points, `SSR` prevents local errors from propagating through the reasoning trace. Compared to standard Self-Refine, which often produces brittle trajectories, `SSR` provides interpretable and controllable corrections that lead to more reliable outcomes. These case studies complement our quantitative results and highlight the practical value of `SSR` as a step-level refinement framework.

**[QUESTION]**
Exits on a highway are numbered consecutively from 1 to 50. The distance from exit 41 to exit 50 is 100 km. If each exit is at least 6 km from the next exit, what is the longest possible distance, in kilometers, between exit 47 and exit 48?

**[GROUND TRUTH]**
52

**[Chain-of-Thought] (1/3)**

Let's carefully analyze the problem step by step:

1. **Understanding the problem:**
   - The highway has exits numbered from 1 to 50.
   - The distance between exit 41 and exit 50 is 100 km.
   - Each exit is at least 6 km from the next exit.
   - We are asked to find the maximum possible distance between exit 47 and exit 48.

2. **Implications of the distance between exit 41 and exit 50:**
   - Since the exits are numbered consecutively, and the distance from exit 41 to exit 50 is 100 km, this involves 9 segments:
     - From exit 41 to 42
     - 42 to 43
     - 43 to 44
     - 44 to 45
     - 45 to 46
     - 46 to 47
     - 47 to 48
     - 48 to 49
     - 49 to 50
   - The total of these 9 segments is 100 km.

3. **Constraints on each segment:**
   - Each segment (distance between consecutive exits) is at least 6 km.
   - This means the minimal total distance from exit 41 to exit 50, if all segments are at their minimum, is:
     $$
     9 \times 6 = 54 \text{ km}
     $$
   - But the actual total is 100 km, which is greater than 54 km, so there is some "extra" distance to distribute among these segments.

4. **Distributing the extra distance:**
   - The total "extra" beyond the minimum is:
     $$
     100 - 54 = 46 \text{ km}
     $$
   - To maximize the distance between exit 47 and exit 48, we want to maximize the length of that specific segment.
   - To do that, we should minimize the distances of all other segments, since the total sum is fixed at 100 km.

Figure 10: **Full-Trace Concrete Example (1/7):** Chain-of-Thought Response (1/3).

**[QUESTION]**
Exits on a highway are numbered consecutively from 1 to 50. The distance from exit 41 to exit 50 is 100 km. If each exit is at least 6 km from the next exit, what is the longest possible distance, in kilometers, between exit 47 and exit 48?

**[GROUND TRUTH]**
52

**[Chain-of-Thought] (2/3)**

5. **Minimizing other segments:**

  - For segments other than 47-48, assign the minimum of 6 km each.

  - The segments before exit 47 are:

  - 41-42

  - 42-43

  - 43-44

  - 44-45

  - 45-46

  - There are 5 such segments, each at 6 km:

  \[

  $5 \times 6 = 30 \text{ km}$

  \]

  - The remaining segments are:

  - 47-48

  - 48-49

  - 49-50

Figure 11: **Full-Trace Concrete Example (2/7):** Chain-of-Thought Response (2/3).

**[QUESTION]**
Exits on a highway are numbered consecutively from 1 to 50. The distance from exit 41 to exit 50 is 100 km. If each exit is at least 6 km from the next exit, what is the longest possible distance, in kilometers, between exit 47 and exit 48?

**[GROUND TRUTH]**
52

**[Chain-of-Thought] (2/3)**

6. **Calculating the maximum for segment 47-48:**
   - The total distance from exit 41 to 50 is 100 km.
   - The distances of the first 5 segments (41-42 to 45-46) sum to 30 km.
   - The remaining distance for the last 4 segments (including 47-48) is:
   \[
   100 - 30 = 70 \text{ km}
   \]
   - Since 48-49 and 49-50 are also at least 6 km each, we need to consider their minimal contributions:
     - 48-49: at least 6 km
     - 49-50: at least 6 km
   - So, the minimal total for these two segments is:
   \[
   2 \times 6 = 12 \text{ km}
   \]
   - Therefore, the maximum possible length for 47-48 is when the other two segments (48-49 and 49-50) are at their minimums:
   \[
   \text{Distance of 47-48} = 70 - 12 = 58 \text{ km}
   \]
   - This is because the total for the last 4 segments is 70 km, allocated as:
     - 47-48: as large as possible
     - 48-49: 6 km (minimum)
     - 49-50: 6 km (minimum)
   - The remaining 70 - 6 - 6 = 58 km can be assigned to 47-48.

7. **Conclusion:**
   - The maximum possible distance between exit 47 and exit 48 is 58 km.

**Final step:**

- Since this calculation is based on assigning the minimal distances to other segments (except 47-48), and respecting the minimum segment length of 6 km, this setup is feasible.

**Therefore, the longest possible distance between exit 47 and exit 48 is 58 km.**

<answer>58</answer>

Figure 12: **Full-Trace Concrete Example (3/7):** Chain-of-Thought Response (3/3).

**[QUESTION]**
Exits on a highway are numbered consecutively from 1 to 50. The distance from exit 41 to exit 50 is 100 km. If each exit is at least 6 km from the next exit, what is the longest possible distance, in kilometers, between exit 47 and exit 48?

**[GROUND TRUTH]**
52

**[Self-Refine]**

<evaluation>In the original solution, the key mistake was in the calculation of the maximum possible distance between exit 47 and exit 48. The initial approach correctly identified that to maximize this gap, the distances from exit 41 to 47 should be minimized (each at 6 km), and the distances from exit 48 to 50 should also be minimized (each at 6 km). The remaining segment, from exit 47 to 48, would then be as large as possible, given the total constraint of 100 km from exit 41 to 50.

However, the original calculation failed to account for the scaling necessary to ensure the total distance from exit 41 to exit 50 is exactly 100 km. It also overlooked the fact that the positions were initially considered in an unscaled coordinate system, leading to an overestimation of the maximum gap.

The key correction is to recognize that the maximum unscaled gap between exits 47 and 48 is 52 km, obtained by assigning minimal distances (6 km) to the other segments. Then, since the total from exit 41 to 50 is 100 km, the actual maximum gap is scaled proportionally:

$$
\text{Maximum unscaled gap} = 52 \text{ km}
$$
$$
\text{Scaling factor} = \frac{100}{106}
$$

Multiplying these gives the true maximum:

$$
52 \times \frac{100}{106} = \frac{5200}{106} = \frac{2600}{53}
$$

Thus, the corrected maximum possible distance between exit 47 and exit 48 is $\frac{2600}{53}$ km, approximately 49.06 km.

The initial answer of 58 km was overestimated because it did not properly incorporate the scaling factor to match the total distance constraint.</evaluation>

<answer>2600/53</answer>

Figure 13: **Full-Trace Concrete Example (4/7):** Self-Refine.

**[QUESTION]**
Exits on a highway are numbered consecutively from 1 to 50. The distance from exit 41 to exit 50 is 100 km. If each exit is at least 6 km from the next exit, what is the longest possible distance, in kilometers, between exit 47 and exit 48?

**[GROUND TRUTH]**
52

**[Socratic Self-Refine (Ours)] (1/3)**

```
[
  {
    "question": "How many segments are there between exit 41 and exit 50?",
    "answer": 9,
    "reference set": [
      "There are 9 segments between exit 41 and exit 50.",
      "There are 9 segments between exit 41 and exit 50.",
      "9",
      "NA",
      "NA"
    ],
    "confidence": 4
  },
  {
    "question": "What is the minimal total distance if each of these 9 segments is exactly 6 km?",
    "answer": 54,
    " reference set ": [
      "54 km",
      "54 km",
      "54 km",
      "54 km",
      "54 km"
    ],
    "confidence": 5
  },
  {
    "question": "What is the total extra distance beyond the minimal total distance of 54 km, given that the actual distance from exit 41 to exit 50 is 100 km?",
    "answer": 46,
    "reference set ": [
      "100 km - 54 km = 46 km",
      "100 km - 54 km = 46 km",
      "100 km - 54 km = 46 km",
      "100 km - 54 km = 46 km",
      "100 km - 54 km = 46 km"
    ],
    "confidence": 5
  },
```

Figure 14: **Full-Trace Concrete Example (5/7):** Socratic Self-Refine (SSR, Ours) (1/3).

**[QUESTION]**
Exits on a highway are numbered consecutively from 1 to 50. The distance from exit 41 to exit 50 is 100 km. If each exit is at least 6 km from the next exit, what is the longest possible distance, in kilometers, between exit 47 and exit 48?

**[GROUND TRUTH]**
52

**[Socratic Self-Refine (Ours)] (2/3)**

```
{
  "question": "How many segments are there between exit 41 and exit 50, and which segments are they?",
  "answer": "Between exits 41-42, 42-43, 43-44, 44-45, 45-46, 46-47, 47-48, 48-49, 49-50",
  "reference set ": [
    "There are 9 segments between exit 41 and exit 50, corresponding to the intervals between each pair of
consecutive exits: from exit 41 to 42, 42 to 43, 43 to 44, 44 to 45, 45 to 46, 46 to 47, 47 to 48, 48 to 49, and 49 to 50.",
    "There are 9 segments between exit 41 and exit 50, corresponding to the intervals between each pair of
consecutive exits from 41 to 50. These segments are:\n1. Between exit 41 and exit 42\n2. Between exit 42 and exit
43\n3. Between exit 43 and exit 44\n4. Between exit 44 and exit 45\n5. Between exit 45 and exit 46\n6. Between exit
46 and exit 47\n7. Between exit 47 and exit 48\n8. Between exit 48 and exit 49\n9. Between exit 49 and exit 50",
    "There are 9 segments between exit 41 and exit 50, corresponding to the intervals between each consecutive pair
of exits from 41 to 50. Specifically, these segments are: \n1. between exit 41 and exit 42 \n2. between exit 42 and
exit 43 \n3. between exit 43 and exit 44 \n4. between exit 44 and exit 45 \n5. between exit 45 and exit 46 \n6.
between exit 46 and exit 47 \n7. between exit 47 and exit 48 \n8. between exit 48 and exit 49 \n9. between exit 49
and exit 50",
    "There are 9 segments between exit 41 and exit 50, corresponding to the intervals between each pair of
consecutive exits: from exit 41 to 42, 42 to 43, 43 to 44, 44 to 45, 45 to 46, 46 to 47, 47 to 48, 48 to 49, and 49 to 50.",
    "There are 9 segments between exit 41 and exit 50, corresponding to the intervals between each pair of
consecutive exits: from exit 41 to 42, 42 to 43, 43 to 44, 44 to 45, 45 to 46, 46 to 47, 47 to 48, 48 to 49, and 49 to 50."
  ],
  "confidence": 4
},
{
  "question": "If all segments except the one between exits 47 and 48 are set to their minimal length of 6 km, what is
the total length of these other segments?",
  "answer": 8,
  "reference set ": [
    "48 km",
    "48 km",
    "48 km",
    "48 km",
    "48 km"
  ],
  "confidence": 0
},
```

Figure 15: **Full-Trace Concrete Example (6/7):** Socratic Self-Refine (SSR, Ours) (2/3).

**[QUESTION]**
Exits on a highway are numbered consecutively from 1 to 50. The distance from exit 41 to exit 50 is 100 km. If each exit is at least 6 km from the next exit, what is the longest possible distance, in kilometers, between exit 47 and exit 48?

**[GROUND TRUTH]**
52

**[Socratic Self-Refine (Ours)] (3/3)**

```
{
  "question": "What is the maximum possible length of the segment between exits 47 and 48, given the total distance
of 100 km and the minimal lengths of other segments?",
  "answer": 52,
  "reference set ": [
   "38",
   "46 km",
   "52 km",
   "52 km",
   "52"
  ],
  "confidence": 4
 }
]
```

<evaluation>\nUpon re-examination, I realize that I miscounted the number of segments between exits 41 and 50. There are 9 segments connecting these 10 exits, not 8. Specifically, the segments are:\n\n41-42, 42-43, 43-44, 44-45, 45-46, 46-47, 47-48, 48-49, 49-50.\n\nIn the previous reasoning, I mistakenly considered only 8 segments when fixing distances at 6 km, excluding the segment between 47 and 48. To maximize the distance between exits 47 and 48, I should set all other segments to their minimal length of 6 km, which is 8 segments (excluding the one between 47 and 48). But actually, there are 9 segments total, and only one of them is between 47 and 48.\n\nTherefore, the correct approach is:\n\n- Fix all other segments (except 47-48) at 6 km.\n- Sum these fixed segments:\n  \n  There are 8 other segments besides 47-48 (from 41-42 up to 46-47, and from 48-49 up to 49-50), totaling 8 segments.\n- Sum of these fixed segments: \\(8 \\times 6 = 48\\) km.\n\n- Total distance from 41 to 50 is 100 km; thus, the length of the segment between 47 and 48 can be:\n\n \\[\n 100 - 48 = 52 \\text{ km}\n \\]\n\nThis confirms that the maximum distance between exit 47 and exit 48 is **52 km**.\n\nThe initial miscount was in the number of segments fixed at minimal length, but after correction, the reasoning aligns with the previous conclusion.\n\n</evaluation>\n\nNow, let's proceed with the step-by-step reasoning again, explicitly addressing the count of segments:\n\n---\n\n### Step 1: Count the total segments between exits 41 and 50\n\n- The exits are numbered 41, 42, 43, 44, 45, 46, 47, 48, 49, 50.\n- The segments are:\n\n 41-42, 42-43, 43-44, 44-45, 45-46, 46-47, 47-48, 48-49, 49-50.\n- Total of **9 segments**.\n\n---\n\n### Step 2: Understand the goal\n\n- The total distance from exit 41 to exit 50 is 100 km.\n- Each segment is at least 6 km.\n- To maximize the distance between exit 47 and exit 48, set all other segments to their minimum length (6 km), except the segment between 47 and 48, which we want to maximize.\n\n---\n\n### Step 3: Fix other segments at minimal length\n\n- The segments fixed at minimum length (6 km):\n\n 41-42, 42-43, 43-44, 44-45, 45-46, 48-49, 49-50.\n- Count how many segments are fixed at 6 km:\n\n From 41-42 to 46-47: 6 segments (41-42, 42-43, 43-44, 44-45, 45-46, 46-47).\n From 48-49 to 49-50: 2 segments.\n- Total fixed segments: **8**.\n\n- Sum of fixed segments: \\(8 \\times 6 = 48\\) km.\n\n---\n\n### Step 4: Calculate the maximum possible distance for the segment 47-48\n\n- Total distance from 41 to 50: 100 km.\n- Sum of fixed segments: 48 km.\n- Remaining distance for the segment between 47 and 48:\n\n \\[\n 100 - 48 = 52 \\text{ km}\n \\]\n\n- This is feasible because the segment length is at least 6 km and can be extended up to 52 km.\n\n---\n\n### **Final conclusion:**\n\nThe maximum possible distance between exit 47 and exit 48 is **52 km**.\n\n---\n\n### Final answer:\n\n<answer>52</answer>"

Figure 16: **Full-Trace Concrete Example (7/7):** Socratic Self-Refine (SSR, Ours) (3/3).

