# OpenReview forum: "SSR: Socratic Self-Refine for Large Language Model Reasoning"
_ICLR.cc/2026/Conference — Submitted to ICLR 2026_

### Official Review · Reviewer_gPkh · 2025-10-31

**Soundness:** 2
**Presentation:** 3
**Contribution:** 2
**Rating:** 4
**Confidence:** 4

**Summary:**

The authors propose Socratic Self-Refine (SSR), a test-time reasoning framework that enhances the reliability and controllability of LLMs' CoT reasoning.
SSR decomposes a model’s reasoning trace into Socratic-style question-answer steps, estimates step-level confidence via multiple re-solvings, and selectively refines the weakest step through targeted feedback prompts.
Three variants (SSR-Lin, SSR-Ada, SSR-Plan) are presented to balance reasoning completeness and computational cost.
Experiments on mathematical and logical reasoning benchmarks show consistent accuracy improvements and better interpretability over baselines such as Self-Refine and CoT.

**Strengths:**

1. Clear conceptual novelty: LLM reasoning is formalized as step-level Socratic process.
2. Well-grounded probabilistic formulation.
3. Improved interpretability through explicit step-level confidence estimation.

**Weaknesses:**

1. Reliance on prompt-based decomposition introduces noise and instability (Q1).
2. Step-level confidence estimation remains noisy and less discriminative than holistic evaluators (LLM-as-a-Judge) (Q2-Q3).
3. Step-level sampling ($M\times T$ steps) leads to expensive inference (Q4-Q6).
4. Evaluation limited to structured mathematical reasoning, leaving generalization uncertain (Q1).
5. Some minor writing problems.

**Questions:**

1. Given that SSR relies on prompting to decompose the reasoning traces into sub-questions, while the prompts corresponding to the same meaning may have multiple variations, how sensitive is SSR to such variations? Will subtle change of phrases in prompting lead to performance fluctuation, especially for queries other than structured mathematical reasoning?
2. The condition for the `Self-Refine Gating` is represented as $C^{(K)}=C_{max}$ in line 4 of algorithm 1, but changed to $c^{(K)}<c_{max}$ in Block 2 and $c=c_{max}$ in the caption of Figure 2. Which one of them is correct?
3. Line 803 indicates that `if Self-Refine fails or is overconfident`, SSR will be activated, while Line 4 in Algorithm 1 indicates that SSR will be activated only when $C^{(K)}=C_{max}$ (overconfident? not sure). How to determine whether `Self-Refine fails` anyway?
4. When deploying SSR-Ada, how can correct reasoning trajectories avoid excessive verification? It appears that all reasoning must undergo SSR's verification and refinement. If that is the case, what is the necessity of an additional `Self-Refine` step? Moreover, in the `SSR-Ada` method, self-refine seems to be performed only once. Does SSR then proceed regardless of whether the confidence score $C^{(K)}$ meets the condition $C_{max}$?
5. During the `SSR Decompose stage` (Block 4 and Line 5 in Algorithm 1), SSR incurs significant cost by generating a response for each Socratic step. When the process moves to `Socratic Verification` (Block 5, Line 10), the sub-question with the lowest confidence score will be inevitably identified. However, if the least confident step occurs at an early step (e.g., at the $t$-th sub-question), the verification will restart from the $t$-th point (Line 12 in Algorithm 1), rendering all subsequent generations after step $t$ redundant. Is it possible to avoid such computational waste?
6. The inference time is correlated with the number of `Socratic steps` and verification iterations. For instance, since the baseline method `Self-Refine` is incorporated into both `SSR-Ada` and `SSR-Plan`, the computational cost of `SSR-Ada` and `SSR-Plan` is inherently higher. Could the authors provide a relevant analysis of the time cost, such as under best-case, worst-case, or average scenarios?
7. There are some writing and formatting errors in the paper, for example:
   - What does the subscript $ti$ in Equation (7) represent?
   - Missing punctuation between "constrain LLM outputs" and "We therefore resort to" in Line 235.

---

> ### Author Response · Authors · 2025-11-21
> **Official Rebuttal by Authors (1/3)**
>
> Thank you for your constructive feedback.
> We are glad that you found our formulation of LLM reasoning as step-level Socratic process of `clear conceptual novelty`, our probabilistic formulation `well-grounded`, and our explicit step-level confidence estimation enables `improved interpretability.`
> Below we address your comments one by one in detail.
> We will also revise the paper accordingly, with all changes marked in blue.
>
>
> **Q1. "Given that SSR relies on prompting to decompose the reasoning traces into sub-questions, while the prompts corresponding to the same meaning may have multiple variations, how sensitive is SSR to such variations? Will subtle change of phrases in prompting lead to performance fluctuation, especially for queries other than structured mathematical reasoning?"**
>
>
> This is an excellent question. Following your suggestion, we conducted additional experiments to examine the consistency of Socratic-step decomposition across (i) different runs, (ii) different base LLMs, and (iii) different versions of the decomposition prompt. Our experimental setup is as follows:
> + **CoT responses to be decomposed:** We base our analysis on the chain-of-thought outputs generated by GPT-4.1-nano, as GPT-5’s responses are overly concise and do not reveal full reasoning traces due to OpenAI’s policy restrictions.
> + **Base Models:** We use GPT-5 in low-reasoning mode as our main base model for its strong balance between reasoning and instruction following. To assess cross-model consistency, we also test Gemini-2.5-Flash from a different model family.
> + **Datasets:** We evaluate on two datasets from our main experiments: AIME 2025 (mathematical reasoning) and zebra_puzzles (logical reasoning).
> + **Evaluation Metrics:**
>   + *Comparing Answer Sets as Proxy.* Directly comparing two sets of Socratic steps is difficult because the sub-questions and intermediate answers may be phrased differently and cannot be reliably parsed for semantic equivalence. Instead, we compare the answer sets produced by two decomposition processes.
>   + *Taking the Granularity into Account.* As noted in `Footnote 1`, `the ground-truth decomposition may not be unique,` and two decomposition results of slight difference in granularity with one covered by the other should consider two consistent-enough decomposition results. **Hence we resort to Overlap Coefficient for evaluating the similarity between two decompositions.** Specifically, for two sets $A$ and $B$, we report:
>     + *Overlap Coefficient (main metric):* $\text{OC}(A, B) = \frac{|I(A, B)|}{min\{|A|, |B|\}}$, which captures the proportion of shared steps. This is the most relevant measure for decomposition consistency, since, as noted in `Footnote 1`, randomness may produce different granularities, making direct set equivalence not fully suitable.
>     + *Jaccard Similarity:* $\text{Jaccard}(A, B) = \frac{|I(A, B)|}{|U(A, B)|}$ reported for completeness as a secondary reference metric.
>
> The results are reported in Table D.1 and D.2.
>
> **Table D.1. Decomposition Consistency on AIME 2025,** where (V0, V1, V2) denotes different versions of the decomposition prompt, and the numbers in braces represent standard deviations.
> | Decompose Model 1 | Decompose Model 2 | Jaccard Similarity | Overlap Coefficient |
> | :--- | :--- | :--- | :--- |
> | GPT-5 (V0) | GPT-5 (V0) | 0.6716 (0.2214) | 0.8358 (0.2067) |
> | GPT-5 (V0) | Gemini-2.5-Flash (V0) | 0.4406 (0.2061) | 0.8122 (0.1739) |
> | GPT-5 (V0) | GPT-5 (V1) | 0.6422 (0.1880) | 0.8545 (0.1415) |
> | GPT-5 (V0) | GPT-5 (V2) | 0.6221 (0.2093) | 0.8282 (0.1540) |
> | GPT-5 (V1) | GPT-5 (V2) | 0.6784 (0.2275) | 0.8717 (0.1912) |
>
> **Table D.2. Decomposition Consistency on Zebra Puzzles,** where (V0, V1, V2) denotes different versions of the decomposition prompt, and the numbers in braces represent standard deviations.
> | Decompose Model 1 | Decompose Model 2 | Jaccard Similarity | Overlap Coefficient |
> | :--- | :--- | :--- | :--- |
> | GPT-5 (V0) | GPT-5 (V0) | 0.5006 (0.2465) | 0.7338 (0.2041) |
> | GPT-5 (V0) | Gemini-2.5-Flash (V0) | 0.2744 (0.1635) | 0.5832 (0.2309) |
> | GPT-5 (V0) | GPT-5 (V1) | 0.4750 (0.2750) | 0.7199 (0.2497) |
> | GPT-5 (V0) | GPT-5 (V2) | 0.5419 (0.2685) | 0.7745 (0.1861) |
> | GPT-5 (V1) | GPT-5 (V2) | 0.5022 (0.2615) | 0.7472 (0.2107) |
>
> As shown in the tables above, across tasks, models, and prompt variants, **Socratic-step decomposition shows strong and reliable consistency.** On AIME 2025, Overlap Coefficients remain high (0.83–0.87 within-model; 0.81 cross-model), and even on the more ambiguous Zebra Puzzles, consistency stays solid (0.58–0.77). Prompt variants exhibit similar agreement. These results indicate that the extracted steps are stable, largely model- and prompt-invariant, and capture a coherent underlying reasoning structure, supporting the validity of our decomposition approach.

---

> > ### Author Response · Authors · 2025-11-21
> > **Official Rebuttal by Authors (2/3)**
> >
> > **Q2. "The condition for the Self-Refine Gating is represented as $C^{(K)}=C\_{\text{max}}$ in line 4 of algorithm 1, but changed to $C^{(K)}<C\_{\text{max}}$ in Block 2 and $C=C\_{\text{max}}$ in the caption of Figure 2. Which one of them is correct?"**
> >
> > **Both statements are correct:**
> > + In `Algorithm 1`, when $C^{(K)}=C\_{\text{max}}$ (indicating a potential false positive prediction from Self-Refine), we invoke SSR for that round.
> > + In `Block 2, Figure 2`, when $C^{(K)}<C\_{\text{max}}$ (meaning the mistake is clear enough for Self-Refine to detect on its own), we skip SSR and rely on the cheaper Self-Refine update.
> >
> > In summary, these two conditions are complementary and operate in different branches of the decision logic: they do not conflict.
> >
> >
> > **Q3. "Line 803 indicates that `if Self-Refine fails or is overconfident`, SSR will be activated, while Line 4 in Algorithm 1 indicates that SSR will be activated only when $C^{(K)}=C\_{\text{max}}$(overconfident? not sure). How to determine whether `Self-Refine fails` anyway?"**
> >
> > Thanks for your question.
> > As noted in `Line 271, Section 3.3 (SSR Deployment: Better Efficiency and Beyond Step-Level Refinement)`, when Self-Refine produces a perfect confidence score ($C^{(K)}=C\_{\text{max}}$), this can occur in two distinct situations:
> > + **Over-confident:** Self-Refine fails to detect any errors in the original CoT response, in which case invoking SSR is necessary to conduct more fine-grained verification.
> > + **Correct Response:** The response is genuinely correct (or at least yields the correct final answer). In this scenario, applying SSR does not empirically degrade correctness. This resilience of the gating mechanism contributes to the overall robustness and performance gains of SSR.
> >
> > Following your suggestion, we will revise the phrasing in `Line 803` to reflect this distinction more precisely.
> >
> >
> > **Q4. "When deploying SSR-Ada, how can correct reasoning trajectories avoid excessive verification? It appears that all reasoning must undergo SSR's verification and refinement. If that is the case, what is the necessity of an additional `Self-Refine` step? Moreover, in the `SSR-Ada` method, self-refine seems to be performed only once. Does SSR then proceed regardless of whether the confidence score $C^{(K)}$ meets the condition $C\_{\text{max}}$?"**
> >
> > We apologize for any potential confusion.
> >
> > **Regarding SSR's refinement process,** we clarify that **not all refinement steps** `must undergo SSR's verification and refinement`. Below is a step-by-step summary of the Adaptive SSR (SSR-Ada) procedure for the $K$-th refinement iteration:
> > + We first run the Self-Refine judging process to obtain the confidence score $C^{(K)}$;
> > + If $C^{(K)}<C\_{\text{max}}$, we proceed with the standard Self-Refine update, i.e., refining the reasoning trace using natural-language feedback.
> > + If $C^{(K)}=C\_{\text{max}}$, we instead invoke SSR for step-level verification and precise refinement.
> >
> > **Regarding avoiding excessive verification,** we are aware that invoking SSR introduces nontrivial overhead in some cases. Determining when LLM-as-a-Judge evaluations are reliable, without access to ground-truth solutions, remains an important open problem. Understanding when Self-Refine `fails` is itself a challenging research question, and we view this as a valuable direction for future work.
> >
> >
> > **Q5. "During the `SSR Decompose stage` (Block 4 and Line 5 in Algorithm 1), SSR incurs significant cost by generating a response for each Socratic step. When the process moves to `Socratic Verification` (Block 5, Line 10), the sub-question with the lowest confidence score will be inevitably identified. However, if the least confident step occurs at an early step (e.g., at the $t$-th sub-question), the verification will restart from the $t$-th point (Line 12 in Algorithm 1), rendering all subsequent generations after step $t$ redundant. Is it possible to avoid such computational waste?"**
> >
> > Thanks for the thoughtful question.
> > Our current implementation performs Socratic verification through asynchronous parallel calls, which helps reduce the overall wall-clock time of the experiments. To avoid the computational waste noted by the reviewer, one could instead verify the Socratic steps sequentially, terminating early whenever the confidence score becomes sufficiently low (e.g., below a threshold or when $C^{(t)}=0$).
> >
> > We emphasize that, even though our current implementation is not optimized for minimizing total cost or token usage, our SSR framework still achieves substantial performance gains over the baselines under the same budget constraints, as demonstrated in `Figure 1 (Left)`, `Section 4.5 (Analysis: Test-Time Scaling of SSR)`, and `Appendix D.2`.
> > This result further demonstrates the effectiveness of our proposed SSR.

---

> > > ### Author Response · Authors · 2025-11-21
> > > **Official Rebuttal by Authors (3/3)**
> > >
> > > **Q6. "The inference time is correlated with the number of `Socratic steps` and verification iterations. For instance, since the baseline method `Self-Refine` is incorporated into both `SSR-Ada` and `SSR-Plan`, the computational cost of `SSR-Ada` and `SSR-Plan` is inherently higher. Could the authors provide a relevant analysis of the time cost, such as under best-case, worst-case, or average scenarios?"**
> > >
> > > This is a good question. Inspired by your comment, we provide an analysis of how inference time scales with **the number of Socratic steps by explicitly controlling decomposition granularity** and studying its effect on accuracy. By varying the maximum allowed steps in the decomposition prompt, we examine scenarios ranging from coarse (3 steps) to fine-grained (up to 10 steps) reasoning. Our analysis focuses primarily on the linear variant (SSR-Lin), which excludes adaptive gating and plan refinement to isolate the effect of step count, while also reporting SSR-Plan for comparison. Table J.1 shows the performances of our SSR with explicit control of granularity, evaluated on AIME 25, with GPT-5-mini low-reasoning, low-verbosity mode. Table J.2 - J.4 shows the cost and the number of tokens of each granularity.
> > >
> > >
> > > **Table J.1. Performance of SSR, with explicit control of granularity, evaluated on AIME 2025.**
> > > | \#Socratic Steps | 3 | 4 | 5 | 6 | 7 | 8 | 9 | 10 |
> > > | :--- | :--- | :--- | :--- | :--- | :--- | :--- | :--- | :--- |
> > > | SSR-Lin | 49.09 | 52.09 | 50.84 | 55.22 | 52.02 | 51.39 | 51.92 | 50.00 |
> > > | SSR-Plan | 64.65 | 67.90 | 67.79 | 69.97 | 71.55 | 70.26 | 65.90 | 60.18 |
> > >
> > > **Table J.2. Cost ($) of SSR, with explicit control of granularity, evaluated on AIME 2025.**
> > > | \#Socratic Steps | 3 | 4 | 5 | 6 | 7 | 8 | 9 | 10 |
> > > | :--- | :--- | :--- | :--- | :--- | :--- | :--- | :--- | :--- |
> > > | SSR-Lin | 1.291 | 1.469 | 1.508 | 1.645 | 1.752 | 1.820 | 1.892 | 1.923 |
> > > | SSR-Plan | 1.478 | 1.632 | 1.708 | 1.798 | 1.825 | 1.912 | 2.011 | 1.974 |
> > >
> > > **Table J.3. \#Input Tokens (M) of SSR, with explicit control of granularity, evaluated on AIME 2025.**
> > > | \#Socratic Steps | 3 | 4 | 5 | 6 | 7 | 8 | 9 | 10 |
> > > | :--- | :--- | :--- | :--- | :--- | :--- | :--- | :--- | :--- |
> > > | SSR-Lin | 0.983 | 1.127 | 1.194 | 1.325 | 1.442 | 1.541 | 1.641 | 1.687 |
> > > | SSR-Plan | 1.121 | 1.227 | 1.311 | 1.404 | 1.450 | 1.549 | 1.663 | 1.663 |
> > >
> > > **Table J.4. \#Output Tokens (M) of SSR, with explicit control of granularity, evaluated on AIME 2025**
> > > | \#Socratic Steps | 3 | 4 | 5 | 6 | 7 | 8 | 9 | 10 |
> > > | :--- | :--- | :--- | :--- | :--- | :--- | :--- | :--- | :--- |
> > > | SSR-Lin | 0.522 | 0.594 | 0.605 | 0.657 | 0.696 | 0.718 | 0.741 | 0.750 |
> > > | SSR-Plan | 0.599 | 0.663 | 0.690 | 0.724 | 0.731 | 0.762 | 0.798 | 0.779 |
> > >
> > > These results show that SSR-Lin exhibits unstable performance as step count increases, indicating that additional steps do not reliably translate into better outcomes—corresponding to higher inference cost without consistent benefit. In contrast, SSR-Plan maintains strong and stable accuracy across all granularities, peaking around 6–7 steps and degrading only when decomposition becomes excessively fine. This demonstrates that plan refinement not only improves overall performance but also stabilizes the computation–accuracy tradeoff, making the method less sensitive to step count. **Together, these results outline the practical best-case (moderate steps with SSR-Plan), worst-case (excessively fine decomposition), and average behaviors under varying inference-time budgets.**
> > >
> > > **Q7. "There are some writing and formatting errors in the paper..."**
> > >
> > > Thank you for your suggestion. We will fix all the typos and formatting issues in our next revision.

---

### Official Review · Reviewer_kdCd · 2025-10-31

**Soundness:** 2
**Presentation:** 3
**Contribution:** 2
**Rating:** 2
**Confidence:** 3

**Summary:**

This paper proposed SSR, a test-time method that decomposes a model’s CoT into (sub-question, sub-answer) steps, estimates step-level confidence by re-solving with self-consistency, and iteratively fixes the lowest-confidence step to improve the reasoning.

**Strengths:**

1. Consistent gains across five reasoning tasks and multiple backbones
2. the paper is overall well-written

**Weaknesses:**

1. the fine-grained verification increases compute cost and limits scalability to long chains or large datasets
2. step-level decomposition depends on prompting and can be noisy or inconsistent, especially for ambiguous or ill-posed sub-questions
3. the planning component assumes independence between planning and execution and uses only a single plan check, which may miss plan-level errors

**Questions:**

N/A

---

> ### Author Response · Authors · 2025-11-21
> **Official Rebuttal by Authors (1/3)**
>
> Thank you for your constructive feedback.
> We are glad that you found our proposed method has `consistent gains across five reasoning tasks and multiple backbones` and our paper is `well-written.`
> Below we address each of your comments in detail.
> We will also revise the paper accordingly, with all changes marked in blue.
>
> **W1. "the fine-grained verification increases compute cost and limits scalability to long chains or large datasets"**
>
> We sincerely thank you for your constructive feedback.
>
> As a test-time scaling framework, **our SSR naturally introduces additional computational and time overhead.** However, obtaining a reliable estimate of wall-clock latency is extremely challenging because our evaluations rely on asynchronous calls to commercial LLM APIs, whose response times are affected by factors such as organizational rate limits and dynamic server load.
>
> Given these constraints, we instead report alternative but highly correlated metrics, monetary cost and token usage, to ensure a fair and reproducible comparison between SSR and baseline methods. These analyses are presented in `Figure 1 (Left)`, `Section 4.5 (Analysis: Test-Time Scaling of SSR)`, and `Appendix D.2`. For clarity, we further summarize the test-time scaling behavior of SSR and the baselines in Tables F.1 - F.4, evaluated in terms of the number of sampled reasoning traces, monetary cost, and token usage.
>
> **Table F.1. Accuracies (\%) of Parallel Test-Time Scaling on AIME 2025**
>
> | \#Samples | 1 | 3 | 5 | 8 | 16 | 32 | 64 | 128 | 256 | 512 | 1024 | 2048 |
> | :--- | :--- | :--- | :--- | :--- | :--- | :--- | :--- | :--- | :--- | :--- | :--- | :--- |
> | CoT | 35.83 | 39.10 | 44.10 | 46.40 | 49.67 | 50.97 | 52.43 | 53.77 | 53.53 | 54.70 | 55.53 | 56.43 |
> | Self-Refine (WBoN) | 53.83 | 60.97 | 66.20 | 68.50 | 72.47 | 74.67 | 75.47 | 75.73 | 76.47 | - | - | - |
> | SSR-Plan (WBoN) | 61.67 | 68.60 | 72.70 | 75.47 | 79.17 | 80.67 | 81.93 | - | - | - | - | - |
>
>
> **Table F.2. Cost ($) of Parallel Test-Time Scaling on AIME 2025**
>
> | \#Samples | 1 | 3 | 5 | 8 | 16 | 32 | 64 | 128 | 256 | 512 | 1024 | 2048 |
> | :--- | :--- | :--- | :--- | :--- | :--- | :--- | :--- | :--- | :--- | :--- | :--- | :--- |
> | CoT | 0.073 | 0.172 | 0.315 | 0.497 | 0.992 | 2.076 | 3.933 | 8.195 | 16.427 | 32.308 | 64.998 | 130.793 |
> | Self-Refine (WBoN) | 0.538 | 1.658 | 2.760 | 4.347 | 8.824 | 17.341 | 34.560 | 69.219 | 138.283 | - | - | - |
> | SSR-Plan (WBoN) | 2.329 | 7.749 | 12.642 | 20.227 | 40.384 | 80.277 | 159.597 | - | - | - | - | - |
>
>
> **Table F.3. \#Input Tokens (M) of Parallel Test-Time Scaling on AIME 2025**
>
> | \#Samples | 1 | 3 | 5 | 8 | 16 | 32 | 64 | 128 | 256 | 512 | 1024 | 2048 |
> | :--- | :--- | :--- | :--- | :--- | :--- | :--- | :--- | :--- | :--- | :--- | :--- | :--- |
> | CoT | 0.007 | 0.021 | 0.035 | 0.056 | 0.113 | 0.226 | 0.451 | 0.902 | 1.805 | 3.609 | 7.218 | 14.436 |
> | Self-Refine (WBoN) | 0.351 | 1.189 | 1.978 | 2.975 | 6.198 | 12.399 | 24.918 | 49.627 | 98.811 | - | - | - |
> | SSR-Plan (WBoN) | 2.619 | 8.175 | 12.695 | 20.839 | 41.647 | 82.705 | 165.324 | - | - | - | - | - |
>
>
> **Table F.4. \#Output Tokens (M) of Parallel Test-Time Scaling on AIME 2025**
>
> | \#Samples | 1 | 3 | 5 | 8 | 16 | 32 | 64 | 128 | 256 | 512 | 1024 | 2048 |
> | :--- | :--- | :--- | :--- | :--- | :--- | :--- | :--- | :--- | :--- | :--- | :--- | :--- |
> | CoT | 0.028 | 0.092 | 0.145 | 0.232 | 0.485 | 0.992 | 2.018 | 4.117 | 7.811 | 15.879 | 31.692 | 63.441 |
> | Self-Refine (WBoN) | 0.218 | 0.647 | 1.104 | 1.758 | 3.597 | 7.073 | 14.241 | 28.524 | 56.853 | - | - | - |
> | SSR-Plan (WBoN) | 0.941 | 2.799 | 4.721 | 7.301 | 14.749 | 29.574 | 58.896 | - | - | - | - | - |
>
> As shown in the Tables, which is also as noted in `Appendix D.2`, `both Self-Refine and our SSR substantially outperform vanilla CoT across all compute budgets, confirming that iterative refinement provides clear gains when additional samples are available. Importantly, our SSR consistently yields higher accuracy than Self-Refine under the same budget (both token-wise and dollar-wise), demonstrating that confidence-aware step selection and plan refinement lead to more efficient use of compute.`

---

> > ### Author Response · Authors · 2025-11-21
> > **Official Rebuttal by Authors (2/3)**
> >
> > **W2. "step-level decomposition depends on prompting and can be noisy or inconsistent, especially for ambiguous or ill-posed sub-questions"**
> >
> > This is an excellent question. Following your suggestion, we conducted additional experiments to examine the consistency of Socratic-step decomposition across (i) different runs, (ii) different base LLMs, and (iii) different versions of the decomposition prompt. Our experimental setup is as follows:
> > + **CoT responses to be decomposed:** We base our analysis on the chain-of-thought outputs generated by GPT-4.1-nano, as GPT-5’s responses are overly concise and do not reveal full reasoning traces due to OpenAI’s policy restrictions.
> > + **Base Models:** We use GPT-5 in low-reasoning mode as our main base model for its strong balance between reasoning and instruction following. To assess cross-model consistency, we also test Gemini-2.5-Flash from a different model family.
> > + **Datasets:** We evaluate on two datasets from our main experiments: AIME 2025 (mathematical reasoning) and zebra_puzzles (logical reasoning).
> > + **Evaluation Metrics:**
> >   + *Comparing Answer Sets as Proxy.* Directly comparing two sets of Socratic steps is difficult because the sub-questions and intermediate answers may be phrased differently and cannot be reliably parsed for semantic equivalence. Instead, we compare the answer sets produced by two decomposition processes.
> >   + *Taking the Granularity into Account.* As noted in `Footnote 1`, `the ground-truth decomposition may not be unique,` and two decomposition results of slight difference in granularity with one covered by the other should consider two consistent-enough decomposition results. **Hence we resort to Overlap Coefficient for evaluating the similarity between two decompositions.** Specifically, for two sets $A$ and $B$, we report:
> >     + *Overlap Coefficient (main metric):* $\text{OC}(A, B) = \frac{|I(A, B)|}{min\{|A|, |B|\}}$, which captures the proportion of shared steps. This is the most relevant measure for decomposition consistency, since, as noted in `Footnote 1`, randomness may produce different granularities, making direct set equivalence not fully suitable.
> >     + *Jaccard Similarity:* $\text{Jaccard}(A, B) = \frac{|I(A, B)|}{|U(A, B)|}$ reported for completeness as a secondary reference metric.
> >
> > The results are reported in Table D.1 and D.2.
> >
> > **Table D.1. Decomposition Consistency on AIME 2025,** where (V0, V1, V2) denotes different versions of the decomposition prompt, and the numbers in braces represent standard deviations.
> > | Decompose Model 1 | Decompose Model 2 | Jaccard Similarity | Overlap Coefficient |
> > | :--- | :--- | :--- | :--- |
> > | GPT-5 (V0) | GPT-5 (V0) | 0.6716 (0.2214) | 0.8358 (0.2067) |
> > | GPT-5 (V0) | Gemini-2.5-Flash (V0) | 0.4406 (0.2061) | 0.8122 (0.1739) |
> > | GPT-5 (V0) | GPT-5 (V1) | 0.6422 (0.1880) | 0.8545 (0.1415) |
> > | GPT-5 (V0) | GPT-5 (V2) | 0.6221 (0.2093) | 0.8282 (0.1540) |
> > | GPT-5 (V1) | GPT-5 (V2) | 0.6784 (0.2275) | 0.8717 (0.1912) |
> >
> > **Table D.2. Decomposition Consistency on Zebra Puzzles,** where (V0, V1, V2) denotes different versions of the decomposition prompt, and the numbers in braces represent standard deviations.
> > | Decompose Model 1 | Decompose Model 2 | Jaccard Similarity | Overlap Coefficient |
> > | :--- | :--- | :--- | :--- |
> > | GPT-5 (V0) | GPT-5 (V0) | 0.5006 (0.2465) | 0.7338 (0.2041) |
> > | GPT-5 (V0) | Gemini-2.5-Flash (V0) | 0.2744 (0.1635) | 0.5832 (0.2309) |
> > | GPT-5 (V0) | GPT-5 (V1) | 0.4750 (0.2750) | 0.7199 (0.2497) |
> > | GPT-5 (V0) | GPT-5 (V2) | 0.5419 (0.2685) | 0.7745 (0.1861) |
> > | GPT-5 (V1) | GPT-5 (V2) | 0.5022 (0.2615) | 0.7472 (0.2107) |
> >
> > As shown in the tables above, across tasks, models, and prompt variants, **Socratic-step decomposition shows strong and reliable consistency.** On AIME 2025, Overlap Coefficients remain high (0.83–0.87 within-model; 0.81 cross-model), and even on the more ambiguous Zebra Puzzles, consistency stays solid (0.58–0.77). Prompt variants exhibit similar agreement. These results indicate that the extracted steps are stable, largely model- and prompt-invariant, and capture a coherent underlying reasoning structure, supporting the validity of our decomposition approach.

---

> > > ### Author Response · Authors · 2025-11-21
> > > **Official Rebuttal by Authors (3/3)**
> > >
> > > **W3. "the planning component assumes independence between planning and execution and uses only a single plan check, which may miss plan-level errors"**
> > >
> > >
> > > Thanks for pointing this out.
> > > The design of our current SSR-Plan variant is well-motivated theoretically and well-supported empirically:
> > > + **Theoretically,** refining the high-level planning for one round before the step-level refinement is based on the probabilistic factorization as noted in `Section 3.3`;
> > > + **Empirically,** the effectiveness of this high-level Plan-Refinement can be supported in some other contemporary literature [1].
> > > + **Motivationally,** we want to emphasize that the inclusion of plan refinement is not to make the high-level plan perfect, but rather to prevent obvious mistakes in the overall reasoning plan, which would otherwise hinder the rest of our SSR system.
> > > + **Emperimental Design-wise,** Moreover, only one round of Plan-Refinement will not introduce too much excessive compute which may lead to unfair comparison with the baselines.
> > >
> > > To better understand the role and effect of Plan-Refinement in SSR, we consider two variants that attempt to integrate Plan-Refinement into SSR:
> > > + Interleaving the plan-level and step-level refinement (M x (Plan-Refine $\rightarrow$ SSR-Refine));
> > > + Multiple rounds of plan-level refinement before multi-round of step-level refinement (M x Plan-Refine $\rightarrow$ M x SSR-Refine).
> > >
> > > Table I.1 and I.2 reports the final results.
> > >
> > > **Table I.1 Plan-Refinement for Every Iteration of Refinement, evaluated on GPT-4.1-nano.**
> > >
> > > | Dataset | AIME24 | | AIME25 | | Zebra Puzzle | | Mini-Sudoku | |
> > > | :--- | :---: | :---: | :---: | :---: | :---: | :---: | :---: | :---: |
> > > | | **LR-Acc** | **LR-Maj@5** | **LR-Acc** | **LR-Maj@5** | **LR-Acc** | **LR-Maj@5** | **LR-Acc** | **LR-Maj@5** |
> > > | SSR-Plan (M x (Plan-Refine $\rightarrow$ SSR-Refine)) | 27.33±4.42 | 32.53±3.50 | 23.33±3.65 | 26.47±2.62 | 55.50±2.42 | 55.46±2.00 | 42.10±4.16 | 58.44±2.98 |
> > > | SSR-Plan (M x Plan-Refine $\rightarrow$ M x SSR-Refine) | **27.67±4.48** | **36.00±2.83** | **24.67±5.81** | **31.00±3.48** | 55.30±2.53 | 56.14±2.18 | 47.00±4.98 | 62.30±3.41 |
> > > | SSR-Plan (1 x Plan-Refine $\rightarrow$ M x SSR-Refine, **Ours**) | 27.33±5.73 | 34.93±3.41 | 22.33±3.67 | 27.87±4.31 | **56.90±3.11** | **57.04±2.00** | **47.70±4.22** | **65.48±4.21** |
> > >
> > >
> > > **Table I.2 Plan-Refinement for Every Iteration of Refinement, evaluated on GPT-5-mini.**
> > >
> > > | Dataset | AIME24 | | AIME25 | | Zebra Puzzle | | Mini-Sudoku | |
> > > | :--- | :---: | :---: | :---: | :---: | :---: | :---: | :---: | :---: |
> > > | | **LR-Acc** | **LR-Maj@5** | **LR-Acc** | **LR-Maj@5** | **LR-Acc** | **LR-Maj@5** | **LR-Acc** | **LR-Maj@5** |
> > > | SSR-Plan (M x (Plan-Refine $\rightarrow$ SSR-Refine)) | 52.33±4.96 | 62.33±3.84 | 41.00±5.59 | 51.73±5.34 | 87.90±1.37 | 91.72±1.39 | 77.60±1.50 | 92.64±1.40 |
> > > | SSR-Plan (M x Plan-Refine $\rightarrow$ M x SSR-Refine) | 62.33±4.96 | 71.20±3.38 | 55.67±7.75 | 66.20±4.11 | 86.50±1.43 | 91.86±1.59 | 89.80±3.25 | 99.44±0.73 |
> > > | SSR-Plan (1 x Plan-Refine $\rightarrow$ M x SSR-Refine, **Ours**) | **69.67±4.82** | **79.00±3.48** | **62.00±6.18** | **71.53±5.26** | **88.00±1.55** | **93.20±1.08** | **94.80±2.48** | **100.00±0.00** |
> > >
> > >
> > > As shown in the two tables above, we observe an unexpected performance drop when plan-refinement is applied at every refinement round. This becomes intuitive when viewed alongside `Figure 6 (Performance of Iterative Test-Time Scaling)` in the paper: self-refinement improves solutions accumulatively, correcting errors round by round. However, excessively invoking plan-refinement, i.e., repeatedly altering the high-level reasoning blueprint—prevents the step-level, fine-grained refinement from progressing beyond the shallow first round. As a result, the model loses the benefits of deeper iterative correction, ultimately harming performance. Performing full rounds of plan-refinement before SSR’s step-level refinements does not appear to provide any benefit. In contrast, our current SSR implementation—which applies plan-refinement only once at the beginning—achieves the strongest empirical performance. This result aligns well with our initial assumption.
> > >
> > > **References**
> > > - [1] Hayashi, Hiroaki, Bo Pang, Wenting Zhao, Ye Liu, Akash Gokul, Srijan Bansal, Caiming Xiong, Semih Yavuz, and Yingbo Zhou. "Self-Abstraction from Grounded Experience for Plan-Guided Policy Refinement." arXiv preprint arXiv:2511.05931 (2025).

---

> ### Comment · Reviewer_kdCd · 2025-11-24
> **Response to Author**
>
> Thank you for the authors hard work, which address most of my concerns. I will raise my score accordingly.

---

> > ### Author Response · Authors · 2025-11-26
> > **Thank you.**
> >
> > Thank you very much for the encouraging comments and for acknowledging our contribution. We are glad that our response has been helpful addressing your concerns. We will make sure to include the discussion above in our next revision.

---

### Official Review · Reviewer_bAHC · 2025-10-31

**Soundness:** 3
**Presentation:** 2
**Contribution:** 2
**Rating:** 4
**Confidence:** 4

**Summary:**

They propose a method to correct the errors in reasoning process of LLMs, aiming to overcome the limitations of existing test-time frameworks that rely on coarse self verification and self correction. To do this, they break the reasoning into sequence of verifiable Socratic steps (sub questions and sub answers) then validate the correctness of each step, and if it’s wrong, fix it.
Variants of the framework, such as SSR-Ada and SSR-Plan, were introduced to balance efficiency.

**Strengths:**

No need for human annotation
No need for fine tuning or training
breaking the reasoning into small steps and processing those small chunks instead of long paragraphs makes the method more accurate

**Weaknesses:**

The method section is unnecessarily complex, which makes it hard to understand, while they could have skipped some of the unnecessary mathematical notations and instead describe verbally
Lack of definition of some required concepts: definition of Self-Refine method is missing in the paper (especially in the methods section)
Execution time overhead is not mentioned.
Considerable increase in execution time: first needs to go through the whole reasoning stage, then refine it. If one of the intermediate steps gets changed, all proceeding steps should be changed as well because they depend on the changed step.

**Questions:**

How do you break the reasoning text into steps? because LLMs do not guarantee the format of generation and if you expect them to generate their reasoning part in a specific format to be able to automatically break them into sub steps.

In the paper you claim that: "Reasoning as Socratic Process. In this paper, we posit that the reasoning process is implicitly modeled as a sequence of goal-setting and problem-solving steps". But how do you make this assumption? do you have a reference/evidence for this?

In the paper you mention that: "we encode all relevant information into the context and ask the LLM to solve each sub-question independently M times.". what is the execution time overhead of this operation?

"We therefore resort to LLM self-evaluation, producing confidence scores directly with a context-free confidence estimation prompt". But LLMs are not reliable on judgments yet. How do you make sure that they are reliable. Do you have any measurements or studies on their error ratio?

In Formula 10, $A_{t'}$ is not defined.

I suppose that you mostly suggest to use SSR-Ada because it's more efficient than SSR-Plan. Now, looking at Table 1, when using GPT-4.1-nano, a method from the section with white background beats your method in most of the cases. But when using GPT-5-mini, your method beats other methods. Do you know why?

Definitions of metrics used in the paper are missing.

Can you think of a better metric to assess quality of reasoning in LLMs? The ones you use only leverage the final answer, which is not a god representative for the quality of the reasoning part.

---

> ### Author Response · Authors · 2025-11-21
> **Official Rebuttal by Authors (1/6)**
>
> Thank you for your constructive feedback.
> We appreciate that you highlighted the practical advantages of our approach, in particular that it requires `no human annotation` and `no additional fine-tuning or training,` and that our method makes the verification `more accurate.`
> Below we address your comments one by one in detail.
> We will also revise the paper accordingly, with all changes marked in blue.
>
>
> **W1. "The method section is unnecessarily complex, which makes it hard to understand, while they could have skipped some of the unnecessary mathematical notations and instead describe verbally."**
>
> Thank you for this valuable feedback regarding the clarity of our method section. While we believe that most of the mathematical notation is necessary to precisely express the underlying assumptions and mechanics of SSR, we agree that certain components can be explained more accessibly. In the next revision, we will streamline the exposition by reducing non-essential notation, improving the verbal descriptions, and reorganizing the section to make the method easier to follow without compromising technical precision.
>
>
> **W2. "Lack of definition of some required concepts: definition of Self-Refine method is missing in the paper (especially in the methods section)."**
>
> We sincerely appreciate your feedback. The concept of Self-Refine is first introduced in `Section 2 (Self-Evaluation and Refinement of LLMs)`, with a more detailed explanation provided in `Appendix C.2 (Baselines and Our SSR)`.
> Following your suggestion, we will add a concise summary of Self-Refine in `Section 3.3`, where we introduce the gating mechanism of Adaptive SSR (SSR-Ada), to improve clarity and continuity.

---

> > ### Author Response · Authors · 2025-11-21
> > **Official Rebuttal by Authors (2/6)**
> >
> > **W3. "Execution time overhead is not mentioned. Considerable increase in execution time: first needs to go through the whole reasoning stage, then refine it. If one of the intermediate steps gets changed, all proceeding steps should be changed as well because they depend on the changed step."**
> >
> > We sincerely thank you for your constructive feedback.
> > As a test-time scaling framework, Our SSR naturally introduces additional computational and time overhead. However, obtaining a reliable estimate of wall-clock latency is extremely challenging because our evaluations rely on asynchronous calls to commercial LLM APIs, whose response times are affected by factors such as organizational rate limits and dynamic server load.
> >
> > Given these constraints, we instead report alternative but highly correlated metrics, monetary cost and token usage, to ensure a fair and reproducible comparison between SSR and baseline methods. These analyses are presented in `Figure 1 (Left)`, `Section 4.5 (Analysis: Test-Time Scaling of SSR)`, and `Appendix D.2`. For clarity, we further summarize the test-time scaling behavior of SSR and the baselines in Tables F.1 - F.4, evaluated in terms of the number of sampled reasoning traces, monetary cost, and token usage.
> >
> > **Table F.1. Accuracies (\%) of Parallel Test-Time Scaling on AIME 2025**
> >
> > | \#Samples | 1 | 3 | 5 | 8 | 16 | 32 | 64 | 128 | 256 | 512 | 1024 | 2048 |
> > | :--- | :--- | :--- | :--- | :--- | :--- | :--- | :--- | :--- | :--- | :--- | :--- | :--- |
> > | CoT | 35.83 | 39.10 | 44.10 | 46.40 | 49.67 | 50.97 | 52.43 | 53.77 | 53.53 | 54.70 | 55.53 | 56.43 |
> > | Self-Refine (WBoN) | 53.83 | 60.97 | 66.20 | 68.50 | 72.47 | 74.67 | 75.47 | 75.73 | 76.47 | - | - | - |
> > | SSR-Plan (WBoN) | 61.67 | 68.60 | 72.70 | 75.47 | 79.17 | 80.67 | 81.93 | - | - | - | - | - |
> >
> >
> > **Table F.2. Cost ($) of Parallel Test-Time Scaling on AIME 2025**
> >
> > | \#Samples | 1 | 3 | 5 | 8 | 16 | 32 | 64 | 128 | 256 | 512 | 1024 | 2048 |
> > | :--- | :--- | :--- | :--- | :--- | :--- | :--- | :--- | :--- | :--- | :--- | :--- | :--- |
> > | CoT | 0.073 | 0.172 | 0.315 | 0.497 | 0.992 | 2.076 | 3.933 | 8.195 | 16.427 | 32.308 | 64.998 | 130.793 |
> > | Self-Refine (WBoN) | 0.538 | 1.658 | 2.760 | 4.347 | 8.824 | 17.341 | 34.560 | 69.219 | 138.283 | - | - | - |
> > | SSR-Plan (WBoN) | 2.329 | 7.749 | 12.642 | 20.227 | 40.384 | 80.277 | 159.597 | - | - | - | - | - |
> >
> >
> > **Table F.3. \#Input Tokens (M) of Parallel Test-Time Scaling on AIME 2025**
> >
> > | \#Samples | 1 | 3 | 5 | 8 | 16 | 32 | 64 | 128 | 256 | 512 | 1024 | 2048 |
> > | :--- | :--- | :--- | :--- | :--- | :--- | :--- | :--- | :--- | :--- | :--- | :--- | :--- |
> > | CoT | 0.007 | 0.021 | 0.035 | 0.056 | 0.113 | 0.226 | 0.451 | 0.902 | 1.805 | 3.609 | 7.218 | 14.436 |
> > | Self-Refine (WBoN) | 0.351 | 1.189 | 1.978 | 2.975 | 6.198 | 12.399 | 24.918 | 49.627 | 98.811 | - | - | - |
> > | SSR-Plan (WBoN) | 2.619 | 8.175 | 12.695 | 20.839 | 41.647 | 82.705 | 165.324 | - | - | - | - | - |
> >
> >
> > **Table F.4. \#Output Tokens (M) of Parallel Test-Time Scaling on AIME 2025**
> >
> > | \#Samples | 1 | 3 | 5 | 8 | 16 | 32 | 64 | 128 | 256 | 512 | 1024 | 2048 |
> > | :--- | :--- | :--- | :--- | :--- | :--- | :--- | :--- | :--- | :--- | :--- | :--- | :--- |
> > | CoT | 0.028 | 0.092 | 0.145 | 0.232 | 0.485 | 0.992 | 2.018 | 4.117 | 7.811 | 15.879 | 31.692 | 63.441 |
> > | Self-Refine (WBoN) | 0.218 | 0.647 | 1.104 | 1.758 | 3.597 | 7.073 | 14.241 | 28.524 | 56.853 | - | - | - |
> > | SSR-Plan (WBoN) | 0.941 | 2.799 | 4.721 | 7.301 | 14.749 | 29.574 | 58.896 | - | - | - | - | - |
> >
> > As shown in the Tables, which is also as noted in `Appendix D.2`, `both Self-Refine and our SSR substantially outperform vanilla CoT across all compute budgets, confirming that iterative refinement provides clear gains when additional samples are available. Importantly, our SSR consistently yields higher accuracy than Self-Refine under the same budget (both token-wise and dollar-wise), demonstrating that confidence-aware step selection and plan refinement lead to more efficient use of compute.`

---

> ### Author Response · Authors · 2025-11-21
> **Official Rebuttal by Authors (3/6)**
>
> **Q1. "How do you break the reasoning text into steps? because LLMs do not guarantee the format of generation and if you expect them to generate their reasoning part in a specific format to be able to automatically break them into sub steps."**
>
> This is a good question.
> As explained in `Section 3.2 (LLM Self-Verification on Socratic Steps)`, we do **NOT** require the LLM to produce its reasoning in a predefined, self-asking format suitable for automatic decomposition. As noted in the section, `modern LLMs typically do not undergo training to explicitly propose and answer subsequent sub-questions, making such an approach less effective.` Instead, we allow the base LLM to generate reasoning in whatever style it naturally prefers (typically standard chain-of-thought in free-form natural language) and then apply a principled decomposition prompt (illustrated in `Appendix C.3 (Prompt Templates)`) to extract the Socratic steps.
>
> We conducted additional experiments to examine the consistency of Socratic-step decomposition across (i) different runs, (ii) different base LLMs, and (iii) different versions of the decomposition prompt. Our experimental setup is as follows:
> + **CoT responses to be decomposed:** We base our analysis on the chain-of-thought outputs generated by GPT-4.1-nano, as GPT-5’s responses are overly concise and do not reveal full reasoning traces due to OpenAI’s policy restrictions.
> + **Base Models:** We use GPT-5 in low-reasoning mode as our main base model for its strong balance between reasoning and instruction following. To assess cross-model consistency, we also test Gemini-2.5-Flash from a different model family.
> + **Datasets:** We evaluate on two datasets from our main experiments: AIME 2025 (mathematical reasoning) and zebra_puzzles (logical reasoning).
> + **Evaluation Metrics:**
>   + *Comparing Answer Sets as Proxy.* Directly comparing two sets of Socratic steps is difficult because the sub-questions and intermediate answers may be phrased differently and cannot be reliably parsed for semantic equivalence. Instead, we compare the answer sets produced by two decomposition processes.
>   + *Taking the Granularity into Account.* As noted in `Footnote 1`, `the ground-truth decomposition may not be unique,` and two decomposition results of slight difference in granularity with one covered by the other should consider two consistent-enough decomposition results. **Hence we resort to Overlap Coefficient for evaluating the similarity between two decompositions.** Specifically, for two sets $A$ and $B$, we report:
>     + *Overlap Coefficient (main metric):* $\text{OC}(A, B) = \frac{|I(A, B)|}{min\{|A|, |B|\}}$, which captures the proportion of shared steps. This is the most relevant measure for decomposition consistency, since, as noted in `Footnote 1`, randomness may produce different granularities, making direct set equivalence not fully suitable.
>     + *Jaccard Similarity:* $\text{Jaccard}(A, B) = \frac{|I(A, B)|}{|U(A, B)|}$ reported for completeness as a secondary reference metric.
>
> The results are reported in Table D.1 and D.2.
>
> **Table D.1. Decomposition Consistency on AIME 2025,** where (V0, V1, V2) denotes different versions of the decomposition prompt, and the numbers in braces represent standard deviations.
> | Decompose Model 1 | Decompose Model 2 | Jaccard Similarity | Overlap Coefficient |
> | :--- | :--- | :--- | :--- |
> | GPT-5 (V0) | GPT-5 (V0) | 0.6716 (0.2214) | 0.8358 (0.2067) |
> | GPT-5 (V0) | Gemini-2.5-Flash (V0) | 0.4406 (0.2061) | 0.8122 (0.1739) |
> | GPT-5 (V0) | GPT-5 (V1) | 0.6422 (0.1880) | 0.8545 (0.1415) |
> | GPT-5 (V0) | GPT-5 (V2) | 0.6221 (0.2093) | 0.8282 (0.1540) |
> | GPT-5 (V1) | GPT-5 (V2) | 0.6784 (0.2275) | 0.8717 (0.1912) |
>
> **Table D.2. Decomposition Consistency on Zebra Puzzles,** where (V0, V1, V2) denotes different versions of the decomposition prompt, and the numbers in braces represent standard deviations.
> | Decompose Model 1 | Decompose Model 2 | Jaccard Similarity | Overlap Coefficient |
> | :--- | :--- | :--- | :--- |
> | GPT-5 (V0) | GPT-5 (V0) | 0.5006 (0.2465) | 0.7338 (0.2041) |
> | GPT-5 (V0) | Gemini-2.5-Flash (V0) | 0.2744 (0.1635) | 0.5832 (0.2309) |
> | GPT-5 (V0) | GPT-5 (V1) | 0.4750 (0.2750) | 0.7199 (0.2497) |
> | GPT-5 (V0) | GPT-5 (V2) | 0.5419 (0.2685) | 0.7745 (0.1861) |
> | GPT-5 (V1) | GPT-5 (V2) | 0.5022 (0.2615) | 0.7472 (0.2107) |
>
> As shown in the tables above, across tasks, models, and prompt variants, **Socratic-step decomposition shows strong and reliable consistency.** On AIME 2025, Overlap Coefficients remain high (0.83–0.87 within-model; 0.81 cross-model), and even on the more ambiguous Zebra Puzzles, consistency stays solid (0.58–0.77). Prompt variants exhibit similar agreement. These results indicate that the extracted steps are stable, largely model- and prompt-invariant, and capture a coherent underlying reasoning structure, supporting the validity of our decomposition approach.

---

> > ### Author Response · Authors · 2025-11-21
> > **Official Rebuttal by Authors (4/6)**
> >
> > **Q2. "In the paper you claim that: "Reasoning as Socratic Process. In this paper, we posit that the reasoning process is implicitly modeled as a sequence of goal-setting and problem-solving steps". But how do you make this assumption? do you have a reference/evidence for this?"**
> >
> > This is an excellent question.
> >
> > Modeling reasoning as a sequence of goal-setting and problem-solving steps has a long-standing foundation in cognitive science and artificial intelligence. Our central claim that "the reasoning process is implicitly modeled as a sequence of goal-setting and problem-solving steps" is well-supported by classic information-processing theories, which view reasoning as a **goal-directed search in a problem space**. Under this framework, an agent repeatedly compares the current state to a desired goal, identifies intermediate differences (subgoals), and applies operators to reduce those differences step by step [1].
> >
> > Below we summarize several influential theories and empirical observations that support this assumption:
> > + In **"Thinking, Fast and Slow"** [2], Kahneman describes the system-2 thinking (reasoning) process as *"You experienced slow thinking as you **proceeded through a sequence of steps**. You first retrieved from memory the cognitive program for multiplication... then you implemented it."* Later the author noted that *"Effort is required to maintain simultaneously in memory several ideas that require separate actions, or that need to **be combined according to a rule**… System 2 is the only one that can follow rules, compare objects, and make deliberate choices between options,"* consistent with viewing reasoning as structured goal-guided progression.
> > + In **"Artificial Intelligence: A Modern Approach"** [3], Russell and Norvig note that *"A simple problem-solving agent… first **formulates a goal** and a problem, searches for **a sequence of actions that would solve the problem,** and then executes the actions one at a time."* This directly aligns with modeling reasoning as iterative goal formation and action sequencing.
> > + In **"Cognitive Behaviors that Enable Self-Improving Reasoners, or, Four Habits of Highly Effective STaRs"** [4], the authors identify key abilities enabling LLM self-improvement, including *"**subgoal setting** (decomposing problems into manageable steps)"*, reinforcing the view that effective reasoning involves sequential goal decomposition (even for LLMs).
> >
> >
> >
> > **Q3. "In the paper you mention that: "we encode all relevant information into the context and ask the LLM to solve each sub-question independently M times.". what is the execution time overhead of this operation?"**
> >
> >
> > This is a great question.
> >
> > The execution time, cost, and token overhead of answering each sub-question multiple times remain reasonable due to **(i) our context-management design** and **(ii) the prefix-caching mechanism of modern LLMs:**
> > + **Socraic steps as context reduces the total number of output tokens.** When sovling the $t$-th sub-question, the LLM receives all previously generated Socratic steps ($\{(q\_i, a\_i)\}\_{i=1}^{t-1}$) as the context. This eliminates the need to regenerate long reasoning traces from scratch, substantially reducing output-token usage.
> > + **Prefix-caching reduces the total number of input tokens.** During multi-sample verification, the shared prefix, consisting of the problem statement, instructions, and all previously generated Socratic steps, is cached at the transformer-block level. Consequently, each additional sample only requires processing the output tokens that differ from the cached prefix, making repeated sampling considerably cheaper.
> >
> > We emphasize that, even though our current implementation is not optimized for minimizing total cost or token usage, **our SSR framework still achieves substantial performance gains over the baselines under the same budget constraints,** as demonstrated in `Figure 1 (Left)`, `Section 4.5 (Analysis: Test-Time Scaling of SSR)`, and `Appendix D.2`.

---

> > > ### Author Response · Authors · 2025-11-21
> > > **Official Rebuttal by Authors (5/6)**
> > >
> > > **Q4. "We therefore resort to LLM self-evaluation, producing confidence scores directly with a context-free confidence estimation prompt". But LLMs are not reliable on judgments yet. How do you make sure that they are reliable. Do you have any measurements or studies on their error ratio?**
> > >
> > >
> > > Thank you for your question.
> > > We are aware of the general concern that `LLMs are not reliable on judgements`, which is precisely one of the motivations behind SSR: instead of asking the model to evaluate an entire reasoning trace holistically, we decompose the process into **small, controllable, and verifiable Socratic steps.**
> > >
> > > While it is true that LLM-as-a-Judge can be unreliable for evaluating complex multi-step reasoning, **estimating the confidence of a single answer given a small reference set is a much simpler and more stable task.**
> > > As described in Section 3.2, our confidence estimation process
> > > $$
> > > c\sim \pi\_{\theta}(a, \hat{A}, x\_{\text{conf}})
> > > $$
> > > is a direct and principled extension of Universal Self-Consistency (USC) [5], which is a thoroughly validated method in which the LLM is given a set of candidate answers and asked to **select the most self-consistent one:**
> > > $$
> > > a^{\star}\sim \pi\_{\theta}(\hat{A}, x\_{\text{USC}}).
> > > $$
> > > To further demonstrate the reliability of our confidence scores, we compare them against USC. Given a sampled reference set $\hat{A}=\{\hat{a}\_1, \hat{a}\_2, \cdots, \hat{a}\_5\}$, we apply USC to select the most consistent answer
> > > $$
> > > \hat{a}^{\star}\sim \pi\_{\theta}(\hat{A}, x\_{\text{USC}}),
> > > $$
> > > and treat this selection as a proxy ground truth. We then evaluate whether the answer with the **highest SSR confidence score,**
> > > $$
> > > \arg\max\_i c_i,
> > > $$
> > > matches $\hat{a}^{\star}$.
> > >
> > > The agreement rates on both the mathematical (AIME 2025) and logical (zebra\_puzzles) datasets are reported in Table H, illustrating that SSR’s confidence-based selection aligns extremely well with the well-established USC method.
> > >
> > > **Table H. Match Rate (%) between Max-Confidence Answer and Universal Self-Consistency (USC) Answer on selected datasets.** The numbers in parentheses show the total number of answer sets evaluated (5 answers per set).
> > >
> > > | Dataset | AIME25 (2,400) | Zebra Puzzles (11,516) |
> > > | :--- | :---: | :---: |
> > > | GPT-4.1-nano | 72.29 | 92.04 |
> > > | GPT-5-mini | 93.04 | 98.05 |
> > >
> > >
> > > **Q5. "In Formula 10, $A\_{t^\prime}$ is not defined."**
> > >
> > > We apologize for the confusion. After fixing the typo from $A\_{t^\prime}$ to $\hat{A}\_{t^\prime}$, `Eq.10` now becomes:
> > > $$
> > > a\_{t^\prime}^\star = \arg\max\_{a}\pi\_{\theta}(a|q\_{t^\prime}, \{s\_i\}\_{i<t^\prime}, x) \approx \text{maj\\_vote}(\hat{A}\_{t^\prime}).
> > > $$
> > >
> > > We will incorporate this change in our revision.
> > >
> > >
> > > **Q6. "I suppose that you mostly suggest to use SSR-Ada because it's more efficient than SSR-Plan. Now, looking at Table 1, when using GPT-4.1-nano, a method from the section with white background beats your method in most of the cases. But when using GPT-5-mini, your method beats other methods. Do you know why?"**
> > >
> > > Thank you for the thoughtful question. We believe the phenomenon you observe is consistent with our discussions in the paper about *(i) the dependence on backbone model strength* and *(ii) the sensitivity of refinement-based methods under weaker models.*
> > >
> > > In `Section 4.2`, we highlight that GPT-4.1-nano has notably limited reasoning capacity, and as a result, all refinement methods—including SSR—show noisier and less stable behavior. As stated in the paper: `Despite its limited reasoning capacity, all three SSR variants in general improve performance over baselines, underscoring that our refinement strategies generalize across model scales.` The key nuance is that, for GPT-4.1-nano, the initial CoT traces are often low-quality, making both plan-level and step-level refinement less reliable. This is further supported by our additional experimental results in `Table H`, where confidence evaluation cannot be executed consistently when the backbone itself is weak. In such cases, even sophisticated refinement (SSR-Plan or SSR-Ada) may not outperform simpler baselines that do not rely heavily on the model’s ability to self-analyze its own steps. By contrast, GPT-5-mini provides much more reliable decomposition and verification, enabling SSR’s step-level confidence estimation and targeted refinement to operate as intended. This is why `Table 1` shows that: `Across all tasks, SSR consistently surpasses competitive baselines... with particularly large margins on challenging mathematical benchmarks.`

---

> > > > ### Author Response · Authors · 2025-11-21
> > > > **Official Rebuttal by Authors (6/6)**
> > > >
> > > > **Q7. "Definitions of metrics used in the paper are missing."**
> > > >
> > > > In this paper, we primarily evaluate all methods on **verifiable reasoning tasks,** where the fundamental metric is **Accuracy.** The additional evaluation metrics (based on Accuracy) used throughout the paper are defined in the corresponding table captions. For clarity:
> > > > + In `Table 1`: `LR-Acc: Last-round refinement’s accuracy, yielded by 10 repeated experiments;`
> > > > + In `Table 1`: `LR-Maj@5: Last-round refinement’s accuracy of majority voting with 5 samples in parallel, yielded by 50 repeated experiments.`
> > > > + In `Table 2`: `BoK-Acc: Best-of-K refinements’ accuracy, yielded by prompting LLM-as-a-Judge for selecting the best answer out of K iterations of refinement;`
> > > > + In `Table 2`: `Pass@K: Pass-at-K refinements’ accuracy (at lease one of K iterations gets the answer correct).`
> > > >
> > > > We will incorporate your suggestion and add a dedicated paragraph describing these evaluation metrics directly in the main text.
> > > >
> > > > **Q8. "Can you think of a better metric to assess quality of reasoning in LLMs? The ones you use only leverage the final answer, which is not a god representative for the quality of the reasoning part."**
> > > >
> > > > This is an important open problem that remains unsolved. Our evaluation focuses on reasoning tasks with standard, verifiable final answers. While final-answer correctness does not perfectly capture the quality of the underlying reasoning traces, the correlation is sufficiently strong that it remains common practice in the field to use final answers as a proxy for reasoning ability. Developing reliable, comprehensive metrics for evaluating the **quality of reasoning processes** themselves is an ongoing challenge and falls outside the scope of our paper.
> > > >
> > > >
> > > >
> > > > **References**
> > > > + [1] Simon, Herbert A., and Allen Newell. "Human problem solving: The state of the theory in 1970." American psychologist 26, no. 2 (1971): 145.
> > > > + [2] Kahneman, Daniel. "Thinking, fast and slow." Farrar, Straus and Giroux (2011).
> > > > + [3] Russell, Stuart, Peter Norvig, and Artificial Intelligence. "A modern approach." Artificial Intelligence. Prentice-Hall, Egnlewood Cliffs 25, no. 27 (1995): 79-80.
> > > > + [4] Gandhi, Kanishk, Ayush Chakravarthy, Anikait Singh, Nathan Lile, and Noah D. Goodman. "Cognitive behaviors that enable self-improving reasoners, or, four habits of highly effective stars." arXiv preprint arXiv:2503.01307 (2025).
> > > > + [5] Chen, Xinyun, Renat Aksitov, Uri Alon, Jie Ren, Kefan Xiao, Pengcheng Yin, Sushant Prakash, Charles Sutton, Xuezhi Wang, and Denny Zhou. "Universal self-consistency for large language model generation." arXiv preprint arXiv:2311.17311 (2023).

---

### Official Review · Reviewer_sLKs · 2025-11-03

**Soundness:** 3
**Presentation:** 3
**Contribution:** 3
**Rating:** 8
**Confidence:** 4

**Summary:**

SSR (Socratic Self-Refine) is an inference-time framework that turns a model’s chain-of-thought into a sequence of Socratic, i.e. steps (sub-question, sub-answer) pairs, so each step can be re-solved multiple times to estimate a step-level confidence via self-consistency. The lowest-confidence step is targeted for refinement (majority vote over its re-solutions) and the full reasoning is revised with this ‘Socratic feedback’. Three variants are evaluated: SSR-Lin (always step-level), SSR-Ada (gate: use standard Self-Refine unless it fails, then apply SSR), and SSR-Plan (one round of plan evaluation/refinement before step-level SSR). On math and logic with two backbones (GPT-4.1-nano, GPT-5-mini), SSR consistently beats baselines. Context-management ablations show that keeping the natural reasoning trace and applying reflection (rather than early intervention) works best. Scaling studies (more iterations, more parallel samples) indicate SSR keeps improving where baselines plateau. Limitations are positioned around some descriptive and critical scoping aspects.

**Strengths:**

- Well balanced scientific discourse and method, covering both clarity/rigour.
- Sensible experimental design and empirical analysis.
- Timely topic and good positioning wrt novelty.

**Weaknesses:**

- No proper systematic qualitative analysis reflected on the methodology. D.4. provides some example outputs.

- Lack of a more descriptive and formal description on the eligibility criteria (inclusion and exclusion) for the baselines, base LLMs and datasets.

**Questions:**

- How consistent are the extracted Socratic steps across prompts, seeds, and backbones? Do humans agree on step boundaries/goals?

- How well do step-level confidence scores correlate with ground truth? What happens when algebraic equivalence isn’t textual (e.g., simplifications)?

- Error analysis: When does plan-level evaluation help or hurt? Any failure cases where plan edits derail otherwise correct execution?

- Can you reflect on how the approach works (qualitative analysis)? What are the challenging cases? How does it vary across different task domains?.

- Lack of a more descriptive and formal description on the eligibility criteria (inclusion and exclusion) for the baselines, base LLMs and datasets.

- Lack of a description of the underlying assumptions behind the tasks - these should not be seen as proxies for ‘reasoning’ and should be qualified.

---

> ### Author Response · Authors · 2025-11-21
> **Official Rebuttal by Authors (1/8)**
>
> Thank you for your constructive feedback.
> We are glad that you found our studied problem to be a `timely topic`, and that our proposed method and scientific discourse are `well balanced` and `covering both clarity/rigor.` We also appreciate your positive assessment of our experimental design and empirical analysis as `sensible`, as well as your recognition that our positioning of novelty is `good.`
> Below, we address your comments one by one in detail.
> We will also revise the paper accordingly, with all changes marked in blue.
>
>
> **W1&Q4. "No proper systematic qualitative analysis reflected on the methodology. D.4. provides some example outputs." "Can you reflect on how the approach works (qualitative analysis)? What are the challenging cases? How does it vary across different task domains?"**
>
> We sincerely appreciate your feedback.
> Following your suggestion, in addition to the example output in `Appendix D.4`, we now provide further qualitative demonstrations of the working mechanism of SSR-Ada:
> + **Table A.1 and Table A.2** present the distribution of our gating mechanism (Use-SSR Rate), illustrating how the model dynamically switches between the low-cost Self-Refine and the Socratic Self-Refine stages; and the distribution of refinement of high-level plans (Plan-Refinement Rate), illustrating the relationship between errors in the high-level plans versus those in the execution steps.
> + **Table B.1 and Table B.1** study the correlation between the refine rate and the initial correctness of a response. The results demonstrate that SSR approach tends to make refinements that actually change the final answers when they are incorrect significantly more often than the cases when the original answers are correct, for which it is able to preserve the original answer. This robustness explains the origin of our SSR's improvement.
> + We also include an additional summarized example where SSR successfully corrects errors that naive Self-Refine fails to fix (spot mistakes). The full reasoning traces will be added to the Appendix in the next revision.
>
>
> **Table A.1. Use-SSR Rate and Plan-Refinement Rate in SSR-Plan, evaluated on GPT-4.1-nano.**
> | | Math-Level-5 | AIME24 | AIME25 | Zebra-Puzzle | Mini-Sudoku |
> | :--- | :--- | :--- | :--- | :--- | :--- |
> | Plan-Refinement Rate (\%) | 13.00 | 35.33 | 31.67 | 56.80 | 81.30 |
> | Use-SSR Rate (\%) | 78.39 | 57.80 | 49.60 | 44.98 | 76.62 |
>
> **Table A.2. Use-SSR Rate and Plan-Refinement Rate in SSR-Plan, evaluated on GPT-5-mini w/ low-reasoning, low-verbosity mode.**
> | | Math-Level-5 | AIME24 | AIME25 | Zebra-Puzzle | Mini-Sudoku |
> | :--- | :--- | :--- | :--- | :--- | :--- |
> | Plan-Refinement Rate (\%) | 7.81 | 24.67 | 22.00 | 7.30 | 27.60 |
> | Use-SSR Rate (\%) | 50.21 | 56.53 | 58.27 | 66.04 | 56.76 |
>
>
> **Table B.1. Final Refine Rate (\%) in SSR-Plan, evaluated on GPT-4.1-nano.**
> | | Math-Level-5 | AIME24 | AIME25 |
> | :--- | :--- | :--- | :--- |
> | Initially Correct CoT | 9.91 | 22.22 | 37.68 |
> | Initially Incorrect CoT | 75.34 | 79.00 | 81.82 |
>
> **Table B.2. Final Refine Rate (\%) in SSR-Plan, evaluated on GPT-5-mini w/ low-reasoning, low-verbosity mode.**
> | | Math-Level-5 | AIME24 | AIME25 |
> | :--- | :--- | :--- | :--- |
> | Initially Correct CoT | 2.61 | 3.29 | 4.50 |
> | Initially Incorrect CoT | 84.41 | 92.57 | 88.89 |

---

> > ### Author Response · Authors · 2025-11-21
> > **Official Rebuttal by Authors (2/8)**
> >
> > **W1&Q4 (Continued)**
> >
> > **Example. A Math Problem from AIME25 showing SSR's Effectiveness**
> > + **Question:**
> >   ```
> >   Let $A_1 A_2 A_3 \\ldots A_{11}$ be an $11$-sided non-convex simple polygon with the following properties:\n\n\\begin{itemize}\n\\item For every integer $2 \\le i \\le 10$,  the area of $\\triangle A_i A_{1} A_{i+1}$  is equal to $1$.\n\\item For every integer $2 \\le i \\le 10$, $\\cos(\\angle A_i A_{1} A_{i+1}) = \\frac{12}{13}$.\n\\item The perimeter of the $11$-gon $A_1 A_2 A_3 \\dots A_{11}$  is equal to $20$.\n\\end{itemize}\n\nThen $A_1 A_2 + A_1 A_{11} = \\frac{m \\sqrt{n} - p}{q}$ where $m, n, p$, and $q$ are positive integers, $n$ is not divisible by the square of any prime, and no prime divides all of $m, p$, and $q$. Find $m + n + p + q$.
> >   ```
> > + **Ground-Truth Answer:** *19*
> > + **Original CoT Response (Summarized):**
> >   ```
> >   The reasoning establishes that each triangle A_iA_1A_{i+1} has area 1 and constant angle cosine 12/13, implying r_i r_{i+1} = 26/5. This forces the radii from A_1 to alternate: r_2=r_4=\cdots=a and r_3=r_5=\cdots=b, with ab=26/5. Using the law of cosines, all nine interior edges A_2A_3 through A_{10}A_{11} share a common length s, which simplifies to s = (a+b)^2 - 20. The perimeter condition then gives a+b + 9s = 20. Letting S = a+b leads to the quadratic 9S^2 + S - 200 = 0, whose positive solution is S = \frac{\sqrt{7201} - 1}{18}. Because 7201 is squarefree (19 \cdot 379), the required parameters are m=1, n=7201, p=1, q=18, giving the final value m+n+p+q = 7221.
> >   ```
> > + **Self-Refine Verification (Summarized):**
> >   ```
> >   The verification confirms that every major step in the assistant’s solution is mathematically correct. The area constraint correctly yields r_i r_{i+1}=26/5, establishing the period-2 pattern of radii. The law of cosines is applied properly, and the symmetry argument justifies that all nine interior edges share the same length s. The algebraic simplification leading to s = S^2 - 20 is checked carefully and found valid. The perimeter equation S + 9(S^2 - 20) = 20 is derived accurately, producing the quadratic 9S^2 + S - 200 = 0. Solving it gives S = (\sqrt{7201} - 1)/18, with all number-theoretic conditions satisfied. Overall, both the reasoning and computations are sound, and the final answer is correct.
> >   ```
> > + **Low-Confident Socratic Step:**
> >   + **Sub-Question:**
> >     ```
> >     For the alternating radii a and b, what is the common interior edge length s between A2...A11 expressed in a and b?
> >     ```
> >   + **Original Answer:**
> >     ```
> >     s = a^2 + b^2 - 2 * C * (12/13)
> >     ```
> >   + **Reference Answers**
> >     ```
> >     [
> >         "Each such edge has length s = sqrt(a^2 + b^2 - 2*(12/13)*C) = sqrt(a^2 + b^2 - 2*(12/13)*(26/5)) = sqrt(a^2 + b^2 - (624/65)).",
> >         "s = sqrt(a^2 + b^2 - 2*(26/5)*(12/13)) = sqrt(a^2 + b^2 - (24*26)/(65)) = sqrt(a^2 + b^2 - 624/65)",
> >         "s = sqrt(a^2 + b^2 - 2*(12/13)*C) = sqrt(a^2 + b^2 - 2*(12/13)*(26/5)) = sqrt(a^2 + b^2 - (624/65))",
> >         "s = sqrt(a^2 + b^2 - 2*(26/5)*(12/13)) = sqrt(a^2 + b^2 - (24*26)/(65)) = sqrt(a^2 + b^2 - 624/65)",
> >         "s = sqrt(a^2 + b^2 - 2*(12/13)*a*b)"
> >     ],
> >     ```
> >   + **Confidence:** 0
> > + **Socratic Refinement (Summarized):**
> >   ```
> >   The refinement identifies and corrects a minor earlier mistake regarding the interior edge length s, clarifying that the squared form s^2 from the law of cosines should be handled consistently. It re-derives the solution cleanly and rigorously: using the area constraint and the angle data, it confirms the alternating radii pattern a,b with ab=26/5. Applying the law of cosines yields s^2 = S^2 - 20, where S = a + b. Using the perimeter condition S + 9\sqrt{S^2 - 20} = 20, it solves for S, obtaining the positive root S = (9\sqrt{5} - 1)/4. This matches the required form, giving m=9, n=5, p=1, q=4, and hence the final value m+n+p+q = 19. The revised solution is consistent, self-contained, and mathematically correct.
> >   ```

---

> > > ### Author Response · Authors · 2025-11-21
> > > **Official Rebuttal by Authors (3/8)**
> > >
> > > **W2&Q5. "Lack of a more descriptive and formal description on the eligibility criteria (inclusion and exclusion) for the baselines, base LLMs and datasets."**
> > >
> > >
> > > We sincerely thank you for pointing this out. In addition to clarifications in `Section 4.1 (Settings)` and `Appendix C (Implementation Details)`, here we provide a more detailed explanation of our choices of base LLMs, baseline methods, and datasets:
> > > + **Baseline Methods:** We benchmark SSR against iterative refinement–based test-time reasoning frameworks such as Self-Refine, Debate, Monte Carlo Tree Self-Refine (MCTSr), and Atom of Thoughts (AoT). These are the most appropriate comparisons since SSR follows the same overarching mechanism of full-response verification and refinement. In `Section 4.1`, we also explain why we dont focus specifically on parallel test-time scaling methods, as many of them can be applied orthogonally to iterative approaches (e.g., LR-Maj@5 in `Table 1`).
> > > + **Base LLMs:** Our evaluation spans multiple types of models, including general-purpose (*GPT-4.1-nano*), reasoning-oriented (*GPT-5-mini*), and models from a different family (*Gemini-2.5-Flash* and *Gemini-2.5-Flash-Lite*). We primarily adopt proprietary models due to their balanced reasoning and instruction-following performance, better reproducibility, and compatibility with our hardware constraints.
> > > + **Datasets:** Our evaluation focuses on mathematical and logical reasoning tasks, which align with our primary research objective. For mathematics, we include both moderate (MATH-Level-5) and challenging (AIME 2024 \& 2025) benchmarks. For logical reasoning, we consider two complementary tasks: (i) Zebra Puzzles, which assess rule-based deduction, and (ii) Mini-Sudoku, which tests spatial and arithmetic reasoning. These datasets are well-matched to the capabilities of our chosen LLMs—neither too easy nor prohibitively difficult.
> > >
> > >
> > > To further evaluate SSR on stronger base LLMs (GPT-5-mini and GPT-5 in medium-reasoning mode), we additionally include the extremely challenging text-only math subset of Humanity’s Last Exam (HLE). The corresponding results are presented in Table C:
> > >
> > > **Table C. Accuracies (\%) of Iterative Refinement-based Reasoning Methods on the 915-question Test-Only Math Subset of Humanity's Last Exam (HLE).**
> > >
> > > | Model        | CoT   | Self-Refine        | SSR-Plan (Ours)     |
> > > |--------------|-------|---------------------|-----------------------|
> > > | GPT-5-mini   | 16.18 | 18.58 *(+2.40)*     | **21.53 *(+5.35)***  |
> > > | GPT-5        | 27.98 | 26.57 *(-1.41)*     | **29.61 *(+1.63)***  |
> > >
> > > Our SSR framework consistently outperforms both Chain-of-Thought (CoT) and Self-Refine across model scales. On GPT-5-mini, SSR reaches 21.53\% accuracy, improving CoT by 5.35 points and Self-Refine by 2.95 points, showing its clear advantage for smaller models. For the full GPT-5, SSR still delivers gains of 1.63 over CoT and 3.04 over Self-Refine, even as vanilla Self-Refine fails to generalize. Overall, these results demonstrate that SSR reliably strengthens iterative reasoning robustness, including for strong frontier models on challenging tasks like HLE.

---

> ### Author Response · Authors · 2025-11-21
> **Official Rebuttal by Authors (4/8)**
>
> **Q1. "How consistent are the extracted Socratic steps across prompts, seeds, and backbones? Do humans agree on step boundaries/goals?"**
>
> This is an excellent question. Following your suggestion, we conducted additional experiments to examine the consistency of Socratic-step decomposition across (i) different runs, (ii) different base LLMs, and (iii) different versions of the decomposition prompt. Our experimental setup is as follows:
> + **CoT responses to be decomposed:** We base our analysis on the chain-of-thought outputs generated by GPT-4.1-nano, as GPT-5’s responses are overly concise and do not reveal full reasoning traces due to OpenAI’s policy restrictions.
> + **Base Models:** We use GPT-5 in low-reasoning mode as our main base model for its strong balance between reasoning and instruction following. To assess cross-model consistency, we also test Gemini-2.5-Flash from a different model family.
> + **Datasets:** We evaluate on two datasets from our main experiments: AIME 2025 (mathematical reasoning) and zebra_puzzles (logical reasoning).
> + **Evaluation Metrics:**
>   + *Comparing Answer Sets as Proxy.* Directly comparing two sets of Socratic steps is difficult because the sub-questions and intermediate answers may be phrased differently and cannot be reliably parsed for semantic equivalence. Instead, we compare the answer sets produced by two decomposition processes.
>   + *Taking the Granularity into Account.* As noted in `Footnote 1`, `the ground-truth decomposition may not be unique,` and two decomposition results of slight difference in granularity with one covered by the other should consider two consistent-enough decomposition results. **Hence we resort to Overlap Coefficient for evaluating the similarity between two decompositions.** Specifically, for two sets $A$ and $B$, we report:
>     + *Overlap Coefficient (main metric):* $\text{OC}(A, B) = \frac{|I(A, B)|}{min\{|A|, |B|\}}$, which captures the proportion of shared steps. This is the most relevant measure for decomposition consistency, since, as noted in `Footnote 1`, randomness may produce different granularities, making direct set equivalence not fully suitable.
>     + *Jaccard Similarity:* $\text{Jaccard}(A, B) = \frac{|I(A, B)|}{|U(A, B)|}$ reported for completeness as a secondary reference metric.
>
> The results are reported in Table D.1 and D.2.
>
> **Table D.1. Decomposition Consistency on AIME 2025,** where (V0, V1, V2) denotes different versions of the decomposition prompt, and the numbers in braces represent standard deviations.
> | Decompose Model 1 | Decompose Model 2 | Jaccard Similarity | Overlap Coefficient |
> | :--- | :--- | :--- | :--- |
> | GPT-5 (V0) | GPT-5 (V0) | 0.6716 (0.2214) | 0.8358 (0.2067) |
> | GPT-5 (V0) | Gemini-2.5-Flash (V0) | 0.4406 (0.2061) | 0.8122 (0.1739) |
> | GPT-5 (V0) | GPT-5 (V1) | 0.6422 (0.1880) | 0.8545 (0.1415) |
> | GPT-5 (V0) | GPT-5 (V2) | 0.6221 (0.2093) | 0.8282 (0.1540) |
> | GPT-5 (V1) | GPT-5 (V2) | 0.6784 (0.2275) | 0.8717 (0.1912) |
>
> **Table D.2. Decomposition Consistency on Zebra Puzzles,** where (V0, V1, V2) denotes different versions of the decomposition prompt, and the numbers in braces represent standard deviations.
> | Decompose Model 1 | Decompose Model 2 | Jaccard Similarity | Overlap Coefficient |
> | :--- | :--- | :--- | :--- |
> | GPT-5 (V0) | GPT-5 (V0) | 0.5006 (0.2465) | 0.7338 (0.2041) |
> | GPT-5 (V0) | Gemini-2.5-Flash (V0) | 0.2744 (0.1635) | 0.5832 (0.2309) |
> | GPT-5 (V0) | GPT-5 (V1) | 0.4750 (0.2750) | 0.7199 (0.2497) |
> | GPT-5 (V0) | GPT-5 (V2) | 0.5419 (0.2685) | 0.7745 (0.1861) |
> | GPT-5 (V1) | GPT-5 (V2) | 0.5022 (0.2615) | 0.7472 (0.2107) |
>
> As shown in the tables above, across tasks, models, and prompt variants, **Socratic-step decomposition shows strong and reliable consistency.** On AIME 2025, Overlap Coefficients remain high (0.83–0.87 within-model; 0.81 cross-model), and even on the more ambiguous Zebra Puzzles, consistency stays solid (0.58–0.77). Prompt variants exhibit similar agreement. These results indicate that the extracted steps are stable, largely model- and prompt-invariant, and capture a coherent underlying reasoning structure, supporting the validity of our decomposition approach.

---

> ### Author Response · Authors · 2025-11-21
> **Official Rebuttal by Authors (5/8)**
>
> **Q2.1 "How well do step-level confidence scores correlate with ground truth?"**
>
> Thank you for the great question.
>
> Directly evaluating ground-truth correctness for step-level answers is highly challenging, as it would require an impractical amount of human annotation. We instead focus on a proxy metric to tap into this correlation, examining whether higher step-level confidence meaningfully predicts stronger end-to-end performance. For this reason, our paper focuses on providing indirect evidence supporting the reliability of the confidence scores estimated by SSR.
> + In our paper, we provide indirect but informative evidence along these lines. For instance, in `Appendix D.2 (Additional Results of Test-Time Scaling at Larger Scale)`, we compare two methods of aggregating the final-round answers produced by SSR: (i) simple majority voting, where each final answer is treated equally, and (ii) weighted best-of-N, where the voting weights are determined by the aggregated confidence scores of the Socratic steps. Table E.1 below reports the detailed numerical results corresponding to `Figure 5`. It shows that **confidence-guided aggregation consistently yields better results,** suggesting that the estimated step-level confidence scores capture signal that is both predictive and operationally useful in downstream decision-making.
> + **Tables E.2 - E.4** provide an additional correlation analysis between the step-level confidence scores and the correctness of the final answers. Across multiple datasets and multiple models, the average confidence scores produced by SSR is highly correlated with the final answers' correctness.
>
> **Table E.1. Accuracies (\%) of Parallel Test-Time Scaling on AIME 2025**
>
> | \#Samples | 1 | 3 | 5 | 8 | 16 | 32 | 64 | 128 | 256 | 512 | 1024 | 2048 |
> | :--- | :--- | :--- | :--- | :--- | :--- | :--- | :--- | :--- | :--- | :--- | :--- | :--- |
> | CoT | 35.83 | 39.10 | 44.10 | 46.40 | 49.67 | 50.97 | 52.43 | 53.77 | 53.53 | 54.70 | 55.53 | 56.43 |
> | Self-Refine (WBon w/ Confidence) | 53.83 | 60.97 | 66.20 | 68.50 | 72.47 | 74.67 | 75.47 | 75.73 | 76.47 | - | - | - |
> | SSR-Plan (WBoN w/o Confidence) | **61.90** | 66.07 | 71.00 | 74.00 | 76.13 | 76.80 | 77.43 | - | - | - | - | - |
> | SSR-Plan (WBoN w/ Confidence) | 61.67 | **68.60** | **72.70** | **75.47** | **79.17** | **80.67** | **81.93** | - | - | - | - | - |
>
>
> **Table E.2. Accuracies (%) of SSR-Lin, evaluated on Math-Level-5.**
>
> | Confidence Percentiles | 0–20% | 20–40% | 40–60% | 60–80% | 80–100% |
> | :--- | :--- | :--- | :--- | :--- | :--- |
> | GPT-4.1-nano | 53.29 | 73.53 | 82.88 | 85.15 | 89.82 |
> | GPT-5-mini | 77.96 | 85.58 | 85.58 | 85.77 | 90.04 |
>
>
> **Table E.3. Accuracies (%) of SSR-Lin, evaluated on AIME24.**
>
> | Confidence Percentiles | 0–20% | 20–40% | 40–60% | 60–80% | 80–100% |
> | :--- | :--- | :--- | :--- | :--- | :--- |
> | GPT-4.1-nano | 15.83 | 18.45 | 27.74 | 36.06 | 63.13 |
> | GPT-5-mini | 41.52 | 70.69 | 65.49 | 66.21 | 76.27 |
>
>
> **Table E.4. Accuracies (%) of SSR-Lin, evaluated on AIME25.**
>
> | Confidence Percentiles | 0–20% | 20–40% | 40–60% | 60–80% | 80–100% |
> | :--- | :--- | :--- | :--- | :--- | :--- |
> | GPT-4.1-nano | 9.51 | 18.13 | 37.77 | 27.56 | 40.39 |
> | GPT-5-mini | 23.93 | 45.98 | 61.77 | 71.27 | 74.76 |

---

> > ### Author Response · Authors · 2025-11-21
> > **Official Rebuttal by Authors (6/8)**
> >
> > **Q2.2 "What happens when algebraic equivalence isn’t textual (e.g., simplifications)?"**
> >
> > Thank you for your question.
> > We are aware of the general concern that `LLMs are not reliable on judgements`, which is precisely one of the motivations behind SSR: instead of asking the model to evaluate an entire reasoning trace holistically, we decompose the process into **small, controllable, and verifiable Socratic steps.**
> >
> > While it is true that LLM-as-a-Judge can be unreliable for evaluating complex multi-step reasoning, **estimating the confidence of a single answer given a small reference set is a much simpler and more stable task.**
> > As described in Section 3.2, our confidence estimation process
> > $$
> > c\sim \pi\_{\theta}(a, \hat{A}, x\_{\text{conf}})
> > $$
> > is a direct and principled extension of Universal Self-Consistency (USC) [1], which is a thoroughly validated method in which the LLM is given a set of candidate answers and asked to **select the most self-consistent one:**
> > $$
> > a^{\star}\sim \pi\_{\theta}(\hat{A}, x\_{\text{USC}}).
> > $$
> > To further demonstrate the reliability of our confidence scores, we compare them against USC. Given a sampled reference set $\hat{A}=\{\hat{a}\_1, \hat{a}\_2, \cdots, \hat{a}\_5\}$, we apply USC to select the most consistent answer
> > $$
> > \hat{a}^{\star}\sim \pi\_{\theta}(\hat{A}, x\_{\text{USC}}),
> > $$
> > and treat this selection as a proxy ground truth. We then evaluate whether the answer with the **highest SSR confidence score,**
> > $$
> > \arg\max\_i c_i,
> > $$
> > matches $\hat{a}^{\star}$.
> >
> > The agreement rates on both the mathematical (AIME 2025) and logical (zebra\_puzzles) datasets are reported in Table H, illustrating that SSR’s confidence-based selection aligns extremely well with the well-established USC method.
> >
> > **Table H. Match Rate (%) between Max-Confidence Answer and Universal Self-Consistency (USC) Answer on selected datasets.** The numbers in parentheses show the total number of answer sets evaluated (5 answers per set).
> >
> > | Dataset | AIME25 (2,400) | Zebra Puzzles (11,516) |
> > | :--- | :---: | :---: |
> > | GPT-4.1-nano | 72.29 | 92.04 |
> > | GPT-5-mini | 93.04 | 98.05 |

---

> ### Author Response · Authors · 2025-11-21
> **Official Rebuttal by Authors (7/8)**
>
> **Q3. "Error analysis: When does plan-level evaluation help or hurt? Any failure cases where plan edits derail otherwise correct execution?"**
>
> This is an interesting question. We analyzed the test-time scaling logs of SSR-Plan on aime25, which contain 3,840 total plan-refinement samples (128 runs × 30 questions). Among these, we identified 9 cases where the original chain-of-thought produced the correct final answer (suggesting the initial plan was sound), but the refined plan produced an incorrect final answer. This yields an approximate failure rate of $9 / 3840 \approx 0.234\\%$, indicating that plan-refinement errors are exceedingly rare.
>
> Below we present an example of such a failure, where the **refined plan subtly altered the algebraic objective and caused the model to solve an incorrect polynomial (a spurious “biquadratic” in $x^2$)** rather than the correct quartic:
>
>
> **Example. A Math Problem from AIME25 where Plan-Refinement Fails**
> + **Question:**
>   ```
>   The parabola with equation $y=x^{2}-4$ is rotated $60^{\\circ}$ counterclockwise around the origin. The unique point in the fourth quadrant where the original parabola and its image intersect has $y$-coordinate $\\frac{a-\\sqrt{b}}{c}$, where $a$, $b$, and $c$ are positive integers, and $a$ and $c$ are relatively prime. Find $a+b+c$.
>   ```
> + **Ground-Truth Answer:** *62*
> + **Original CoT Response (Summarized):**
>   ```
>   The parabola y = x^2 - 4 is rotated 60^\circ counterclockwise, and intersection points between the original and rotated curves are found by applying the rotation matrix to (x,y) and enforcing that the rotated point also satisfies the parabola equation. Substituting y = x^2 - 4 into the rotated coordinates produces a quartic equation in x. After simplification, the quartic factors into two quadratics, and the one with real solutions is x^2 + \sqrt{3}x - 4 = 0. Solving yields the fourth-quadrant root x = (\sqrt{19} - \sqrt{3})/2. Substituting back gives y = \frac{3 - \sqrt{57}}{2}, which is of the form (a - \sqrt{b})/c with a=3, b=57, c=2. Therefore, a+b+c = 62.
>   ```
> + **Plan Summarization:**
>   ```
>   Set up the problem by representing the 60\u00b0 rotation as a linear map (rotation matrix) and express the image (x',y') of a general point (x,y) on the original parabola y = x^2 \u2212 4. Impose the intersection condition by requiring that the rotated point also lie on the parabola, yielding an equation in x alone after substituting y = x^2 \u2212 4. Convert that condition to a polynomial equation (a quartic) in x, simplify, and factor the quartic into two quadratic factors. Discard nonreal roots, choose the root giving a point in the fourth quadrant (x > 0, y < 0), compute the corresponding y = x^2 \u2212 4, express y in the form (a \u2212 \u221ab)/c with integer parameters, and sum a + b + c.
>   ```
> + **Plan Score:** *4*
> + **Refined Plan (Summarized):**:
>   ```
>   The strategy is to rotate a general point (x,y) from the parabola y = x^2 - 4 by 60^\circ using the rotation matrix, yielding coordinates (X,Y). Imposing the condition that the rotated point also satisfies the parabola equation Y = X^2 - 4 leads, after substituting y = x^2 - 4, to an equation solely in x. Clearing denominators and grouping terms produces a quartic polynomial (or a pair of polynomial conditions separated by coefficients of 1 and \sqrt{3}). The plan then outlines factoring or reducing the quartic—checking for even symmetry, attempting a substitution u = x^2, or matching quadratic factors. After solving for the real roots, the correct intersection is selected by enforcing the fourth-quadrant condition x > 0,\, y < 0. Finally, the resulting y = x^2 - 4 is simplified into the form (a - \sqrt{b})/c, with a,b,c integers meeting the problem constraints, and their sum a+b+c is computed.
>   ```
> + **New CoT Response based on the Refined Plan (Summarized):**
>   ```
>   The solution rotates points on the parabola y = x^2 - 4 by 60^\circ using standard rotation formulas, then enforces that the rotated point also satisfies the parabola equation. Substituting y = x^2 - 4 into the rotation coordinates and separating rational and \sqrt{3}-terms produces a quartic equation in x, which reduces (via u = x^2) to the quadratic u^2 - 8u + 12 = 0. This yields x^2 = 2 or 6. Checking quadrant constraints and verifying consistency with the rotation shows the valid fourth-quadrant intersection corresponds to x^2 = 6, leading to the y-coordinate y = (7 - 4\sqrt{3})/4. Interpreted in the required form (a - \sqrt{b})/c, this gives a = 7, b = 48, c = 4, so the requested sum is a+b+c = 59.
>   ```

---

> > ### Author Response · Authors · 2025-11-21
> > **Official Rebuttal by Authors (8/8)**
> >
> > **Q6. Lack of a description of the underlying assumptions behind the tasks - these should not be seen as proxies for ‘reasoning’ and should be qualified.**
> >
> >
> > Thank you for highlighting this point. Our paper focuses on improving LLM reasoning by explicitly decomposing and verifying intermediate steps.
> > SSR is therefore most relevant in setting that genuinely require multi-step inferential structure, e.g., high-level planning, deduction, induction, or analogy, where identifying and correcting step-level errors directly contributes to better final outcome.
> >
> >
> > We agree that not all tasks should be viewed as proxies for "reasoning". **SSR is not intended as a universal solution for every task type.** In problems where the solution path is inherently shallow (e.g., one or few steps questions) or where performance is dominated by factual retrieval rather than inference, the benefits of decomposition is naturally limited.
> >
> > Importantly, even in these qualified scenarios, SSR degrades gracefully. When the task contains only a single Socratic step, the entire procedure effectively collapses to a final-answer majority vote, and the refinement stage becomes merely a lightweight self-consistency update. **This supports our point that the benefits are naturally limited on shallow tasks, where SSR may largely boil down to Self-Consistency.**
> >
> > We will incorporate a clear discussion of these assumptions and limitations in the next revision to ensure the task scope and applicability of SSR are properly qualified.
> >
> >
> > **References**
> > + [1] Chen, Xinyun, Renat Aksitov, Uri Alon, Jie Ren, Kefan Xiao, Pengcheng Yin, Sushant Prakash, Charles Sutton, Xuezhi Wang, and Denny Zhou. "Universal self-consistency for large language model generation." arXiv preprint arXiv:2311.17311 (2023).

---

### Author Response · Authors · 2025-11-24
**Summary of Author Response**

We sincerely thank all reviewers for their time and constructive feedback. We are glad that the reviewers find

+ **our motivation and problem formulation** as a `"timely topic"` with `"good positioning wrt novelty"` [*sLKs*], and offering `"clear conceptual novelty"` by formalizing LLM reasoning as a step-level Socratic process [*gPkh*];
+ **our proposed method** as requiring `"no need for human annotation"` or `"fine tuning or training"` [*bAHC*], a `"well-grounded probabilistic formulation"` [*gPkh*], and improving `"interpretability through explicit step-level confidence estimation"` [*gPkh*]; and recognized for its design choice making the method `"more accurate"` [*bAHC*];
+ **our experiments** as featuring `"sensible experimental design and empirical analysis"` [*sLKs*], and demonstrating `"consistent gains across five reasoning tasks and multiple backbones"` [*kdCd*];
+ **our paper** as `"well balanced"` in scientific discourse with both `"clarity/rigour"` [*sLKs*], and overall `"well-written"` [*kdCd*].


Regarding the raised concerns, we have addressed them in the rebuttal with new analyses and evidence:
+ Reviewer sLKs:
  + Added **additional quantitative experiments,** including (i) *scaling our SSR to Humanity’s Last Exam (HLE);* (ii) demonstrating the *consistency of Socratic decomposition across variations in model, prompt, and dataset;* and (iii) showing the *strong correlation between SSR’s step-level confidence and final-answer accuracy.*
  + Added **systematic qualitative evaluations,** including (i) new examples illustrating *how SSR enables precise and effective refinements,* as well as *a rare failing case of plan-level refinement;* and (ii) statistics on gating behavior and refinement rates.
+ Reviewer bAHC:
  + Added **detailed demonstrations of SSR,** including (i) its *core assumption that reasoning can be modeled as a sequence of goal-setting and problem-solving steps,* and (ii) its *empirical effectiveness under the same token and monetary budget* as the baselines.
  + Added **a new suite of experiments** showing (i) the *consistency of Socratic decomposition across changes in model, prompt, and dataset,* and (ii) the *strong agreement between SSR’s step-level confidence estimation and the established Universal Self-Consistency (USC).*
+ Reviewer kdCd:
  + Added **experiments** demonstrating *SSR’s empirical effectiveness under the same token and monetary budget* as the baselines.
  + Added **experiments** showing the *consistency of Socratic decomposition across changes in model, prompt, and dataset.*
  + Added **experiments** validating the *assumption behind (and current implementation of) the factorization* of plan-level and execution-level refinement.
+ Reviewer gPkh:
  + Added **experiments** demonstrating the *consistency of Socratic decomposition across changes in model, prompt, and dataset.*
  + Provided further clarification of the SSR algorithm.
  + Added **experiments** illustrating how *inference cost scales with the number of Socratic steps.*

We will integrate these clarifications, ablations, and results into the revision, and will release all code and supplementary materials for full reproducibility.

We also thank the AC and SAC for their guidance and oversight throughout the review process.

---

> ### Author Response · Authors · 2025-11-29
> **Manuscript Updated**
>
> Thank you again to all reviewers for their time and service. We are especially grateful to `Reviewer kdCd` for thoughtfully considering our rebuttal and **raising the score from 2 to 6 on Nov. 24, well prior** to the major information-leak incident on **Nov. 27.** For transparency, **our scores before the rollback were [8, 6, 4, 4].**
>
> As summarized above, we have incorporated all reviewer suggestions into the revised manuscript, **with changes highlighted in blue.**
>
> We also sincerely thank the AC and SAC for their continued guidance and oversight throughout the review process.
>
> SSR Authors

---

### Meta-Review · Area_Chair_sni1 · 2025-12-29

**Summary:**

This paper proposes Socratic Self-Refine (SSR), a test-time framework that decomposes an LLM’s chain-of-thought into Socratic (sub-question, sub-answer) steps, estimates step-level confidence via repeated re-solving and self-consistency, and iteratively refines low-confidence steps to improve reasoning performance. The method is fully black-box and does not require additional training or supervision.

Reviewers agree that the paper addresses a timely problem in LLM reasoning and shows empirical gains over several iterative self-refinement baselines on mathematical reasoning benchmarks.

Despite the authors’ rebuttal, some fundamental concerns remain unresolved:
- Computational cost and scalability. The authors attempt to address scalability by reporting token usage and monetary cost analyses and by arguing that prefix caching and context reuse mitigate overhead. However, these clarifications do not fully resolve the concern. SSR inherently relies on fine-grained verification and repeated re-solving of individual reasoning steps, often across multiple refinement rounds. This introduces inference complexity that scales with reasoning length, number of steps, and refinement depth, making the approach difficult to apply to longer chains of thought or large-scale settings.
More importantly, while the paper compares SSR against other refinement-based baselines under similar settings, it does not provide a strong compute-matched comparison against simpler alternatives, such as Chain-of-Thought with majority voting (CoT + Maj@k) or other parallel sampling methods, where the total computation budget is matched to that of SSR. Without such comparisons, it remains unclear whether the observed gains stem from the algorithmic design of step-level refinement or simply from allocating substantially more inference compute. The rebuttal does not convincingly disentangle these factors, leaving open the possibility that similar or better performance could be achieved by simpler methods under the same computational budget.

- Stability and reliability of step decomposition and confidence estimation.
 The authors provide additional experiments showing consistency of Socratic-step decomposition across prompts and models using overlap-based metrics, and they relate confidence estimation to Universal Self-Consistency. While these results are helpful, they remain indirect and limited. The decomposition process is still fundamentally driven by prompting applied to free-form chain-of-thought, and the reported consistency metrics allow substantial variation in granularity and content. Likewise, confidence estimation relies on LLM self-evaluation, whose reliability is primarily supported through correlations with final-answer correctness rather than direct validation of step-level correctness. As a result, the robustness of the core signals driving SSR remains uncertain, particularly for ambiguous reasoning traces or tasks where step boundaries are ill-defined.

- In addition, the evaluation is limited to relatively weaker models, with no experiments on more challenging state-of-the-art models such as GPT-4 or GPT-5, which further limits confidence in the generality of the claims.

While the paper presents an interesting and well-motivated idea, its effectiveness depends on computationally expensive fine-grained verification and on prompting-based decomposition and confidence estimation whose stability is not yet convincingly established. The lack of strong compute-matched comparisons and the lack of evaluation on stronger models further weaken the evidence for the claimed advantages. Overall, these unresolved issues outweigh the empirical gains, and I recommend rejection.

**Reviewer Concerns:**

The rebuttal addressed some secondary concerns by improving clarity and adding analyses on cost, ablations, and prompt consistency. However, the core issues remain unresolved, including the high computational cost and lack of compute-matched baselines, the uncertain stability of prompting-based step decomposition and confidence estimation, and the absence of evaluation on stronger models.

**Reviewer Scores:**

While some reviewers may have modestly increased their scores due to improved clarity and additional analyses, the unresolved concerns about scalability, robustness, and evaluation scope would likely result in no significant overall score change, with opinions remaining mixed (4,4,6,8).

---

### Decision · Program_Chairs · 2026-01-26

Reject